# Let's Think in Two Steps: Mitigating Agreement Bias in MLLMs with Self-Grounded Verification

**Moises Andrade   Joonhyuk Cha   Brandon Ho   Vriksha Srihari   Karmesh Yadav   Zsolt Kira**

Georgia Institute of Technology

{mandrade, jcha, bho36, vriksha.srihari, kyadav32, zkira}@gatech.edu

## Abstract

Verifiers—functions assigning rewards to agent behavior—have been key to AI progress in domains such as math, code, and games. However, extending these gains to domains without clear-cut success criteria (e.g., computer use) remains a challenge: while humans can recognize desired outcomes, translating this intuition into scalable rules is nontrivial. Multimodal LLMs (MLLMs) emerge as a promising solution, given vast world knowledge, human-preference alignment, and reasoning capabilities. We evaluate MLLMs as verifiers across web navigation, computer use, and robotics, spanning 13+ model families, 28+ evaluation templates, curated trajectories from diverse agents and of varying lengths, and distinct verifier applications. We identify a critical limitation: a strong tendency for MLLMs to over-validate agent behavior—a phenomenon we term agreement bias. This bias is pervasive across models, resilient to test-time scaling, and can harm methods relying on MLLM evaluations, such as filtered behavior cloning and self-improvement. We provide guidance on the design and evaluation of MLLM verifiers, and introduce Self-Grounded Verification (SGV), a lightweight method that harnesses MLLMs' own sampling mechanisms by modulating (un)conditional generation to better leverage their knowledge, alignment, and reasoning. SGV operates in two steps: first, the MLLM is elicited to generate broad priors about desired behavior, independent of the data under evaluation. Then, conditioned on self-generated priors, it reasons over and evaluates a candidate trajectory. Our methods yield gains across models and environments, improving failure detection by up to 25 pp and accuracy by 14 pp, with benefits extending to downstream applications. In self-improvement and online supervision, SGV boosts task completion of a GUI specialist in OSWorld, a diffusion policy in robomimic, and a ReAct agent in VisualWebArena—setting a new state of the art, surpassing the previous best by 20pp. Finally, we release an updated version of VisualWebArena featuring strong agent baselines, more human-aligned evaluators, high-fidelity environment parallelism, runtime speedups exceeding 10×, and VisualWebArena-Lite, a 1/3-scale subset with comparable evaluation fidelity. Our code, models, and data are publicly available at our project page.

## 1 Introduction

Several breakthroughs in artificial intelligence can be viewed through the lens of search guided by verifiers—functions assigning rewards to agent behavior aligned with desired criteria. Notable examples include seminal work in Go (Campbell et al., 2002) and Chess (Silver et al., 2017), where search and learning are guided by 0/1 rewards tied to the game's final outcome, and recent advancements in large reasoning models (LRMs), leveraging formal verifiers in code and math (Shao et al., 2024b).

However, while domains such as math, code, and games benefit from relatively well-defined criteria to evaluate agent behavior, this clarity diminishes in open-ended settings. Evaluation in such scenarios often requires nuanced criteria and reasoning over possibly long sequences of multimodal inputs. For example, consider evaluating the trajectory in Fig. 1 (top) produced by a digital agent asked to "add the least expensive opaque phone case to a shopping cart". Should the agent sort products by price?

Perform an advanced search? What is "opaque enough"? Although humans can often recognize satisfactory outcomes, formalizing this intuition into precise and scalable rules remains a challenge.

Multimodal Large Language Models (MLLMs) emerge as a promising solution to bridge this gap. With vast world knowledge, human-preference alignment, and large context windows, MLLMs hold the potential to serve as general-purpose verifiers, capable of handling inputs and producing rewards in multiple modalities. In this work, we probe this potential through a comprehensive study of MLLM-as-verifiers on open-ended tasks, spanning diverse environments, agent architectures, state-of-the-art MLLMs and LRMs, test-time scaling techniques, verifier designs, and evaluation metrics. We consider web, desktop, and robotic environments—VisualWebArena (Koh et al., 2024a), OSWorld (Xie et al., 2024), and robomimic (Mandlekar et al., 2021)—covering roughly 1,300 tasks across varied domains, where verification demands nuanced criteria and multimodal reasoning.

We identify a systematic and critical limitation: a strong tendency for MLLMs to over-validate agent behavior, a phenomenon we call *agreement bias*. As shown in Fig. 1 (middle), an MLLM verifier validates flawed behavior and even generates chains-of-thought (CoT) to rationalize incorrect judgments. This bias limits MLLMs' ability to fulfill a core function of a verifier—identifying flawed behavior and providing feedback to improve performance—posing risks to methods that rely on MLLM-based evaluations. In particular, it can limit MLLMs' effectiveness as data curators for fine-tuning and self-improvement pipelines (Trabucco et al., 2025a; Pan et al., 2024a; Wang et al., 2024; Yu et al., 2025; Shinn et al., 2023); as providers of rewards for training, search, and agent steering (Pan et al., 2024b; Koh et al., 2024b; Sun et al., 2025; Bai et al., 2024); and as judges (Chen et al., 2024; Lee et al., 2024; Zheng et al., 2023) and monitors (Baker et al., 2025) of agent behavior, where failure detection is essential for balanced assessments and to prevent harmful outcomes.

Notably, such failures occur despite MLLMs exhibiting human-aligned priors about desired behavior, suggesting a bottleneck in knowledge extraction—potentially rooted in fundamental limitations in pretraining (Allen-Zhu and Li, 2024b;a) and RLHF (Sharma et al., 2025)—that remains unresolved by major test-time scaling techniques and training for reasoning. To address this, we propose Self-Grounded Verification (SGV), a simple yet effective method that harnesses MLLMs' own sampling mechanisms by modulating (un)conditional generation to better leverage their knowledge, alignment, and reasoning (Fig. 1, bottom). SGV operates in two steps: first, an MLLM is elicited to produce broad priors about desired behavior, conditioned only on the necessary context to frame the task. Then, conditioned on self-generated priors, the model reasons over and evaluates a candidate trajectory.

To assess the benefits and limitations of MLLM verifiers, as well as the effectiveness of SGV, we evaluate performance across three representative settings: offline evaluation of agent performance and two downstream applications—online supervision and self-improvement via Reflexion(Shinn et al., 2023). Our findings and contributions can be summarized as follows:

- Agreement bias is pervasive across MLLMs and LRMs, and is reflected in multiple quantitative metrics, including failure detection rates as low as 50% and output distributions skewed toward favorable judgments. This bias persists across 28+ prompt/scoring templates, techniques to mitigate biases in LLM judgments, and test-time scaling. It holds for both weak and strong agents built from models distinct from the verifier; increases with the capability gap between agents and verifiers; and is exacerbated when verification is treated as a binary success-or-failure classification.

- We elucidate how several applications rely on MLLMs acting as verifiers, and are sensitive to the quality of their evaluations. We show that agreement bias can impair the use of MLLMs for benchmarking agent performance, behavior cloning, self-improvement, and online supervision.

- Notably, these findings apply to a verifier that shows high aggregate metrics (e.g., over 90% recall) and state-of-the-art results in concurrent reward benchmarks. We discuss key factors for evaluating verifiers, including the importance of fine-grained and downstream metrics, challenges arising from leniency-strictness trade-offs, and confounders such as environment bugs and agent strength.

- We evaluate several design choices for building and evaluating MLLM-based verifiers, including the impact of Likert and score-based scales, trajectory length, agent strength, tools, model calibration, techniques to mitigate biases in LLM judges, and periodic vs. outcome-based verification.

- Our methods yield gains of up to 25 pp in failure identification, 14 pp in accuracy, and more human-aligned evaluations across models and environments, with benefits to downstream applications.

- In self-improvement, our stronger SGV-based verifier yields gains of up to 10 pp (24% relative) on VisualWebArena. In online supervision, it encourages agents to backtrack and avoid greedy

strategies, yielding gains of 9 pp (20%) for a ReAct agent on VisualWebArena, 5 pp (22%) for the GUI-Specialist UI-TARS on OSWorld, and 8 pp (33%) for a diffusion policy on robomimic's tool-hang task. Our agents set a **new state-of-the-art on VisualWebArena**, outperforming the previous best by 20 pp while incurring lower token overhead and only utilizing native web actions.

• As a byproduct of our work, we release an updated version of VisualWebArena featuring more human-aligned evaluators, high-fidelity environment parallelism, runtime speedups exceeding 10×, and VisualWebArena-Lite—a 1/3-scale subset with comparable evaluation fidelity.

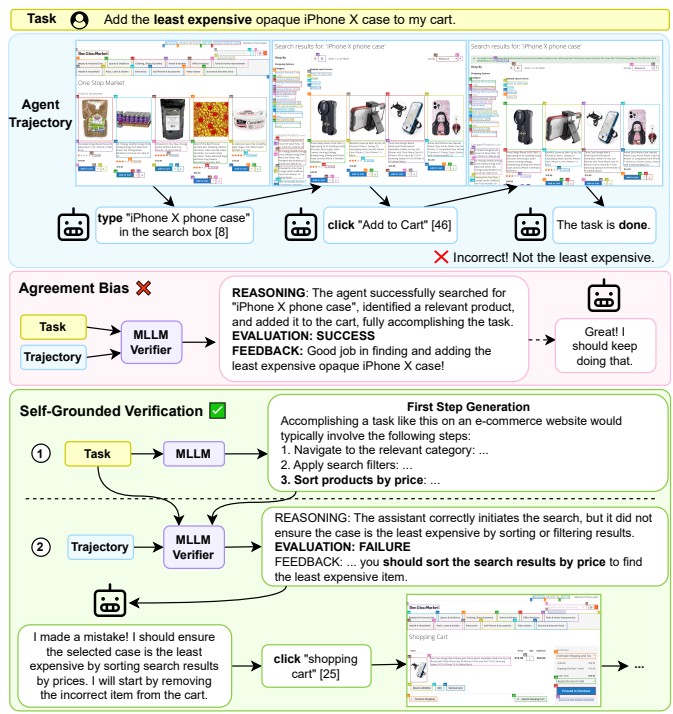

Figure 1: **Top**: example of a web task and corresponding agent trajectory. **Middle**: an MLLM verifier validates and reinforces flawed behavior, generating reasoning to rationalize its incorrect judgments. **Bottom**: SGV improves verification and enables the agent to backtrack.

## 2 RELATED WORK

**MLLMs as Evaluators.** (M)LLMs have been employed as evaluators of model outputs in various scenarios under various names—(M)LLMs as judges, critics, reward models, value functions. This work focuses on multimodal and environment-interaction scenarios. MLLMs have been used to score and filter agent trajectories for subsequent use in finetuning (Trabucco et al., 2025b; Pan et al., 2024a; Ma et al., 2024), and test-time refinements such as to induce prompts, reflections, and tools (Wang et al., 2024; Yu et al., 2025; Shinn et al., 2023; Sarch et al., 2025). They have also served as a source of real-time feedback by producing natural language "critiques" (Pan et al., 2024b), scores to rank action proposals in search (Koh et al., 2024c; Yang et al., 2025), and rewards for training (Sun et al., 2025).

**AI Agents.** There is growing interest in building AI agents to act in various environments, including the web (Koh et al., 2024a), mobile phones (Li et al., 2024), and computers (Liu et al., 2023). In this field, the work most closely related to ours is Pan et al. (2024a), which uses a GPT-4V-based evaluator, prompted with benchmark-specific rubrics, to evaluate trajectories for Reflexion (Shinn et al., 2023) and behavior cloning. Following works employ a similar evaluator to guide tree search (Yu et al., 2024; Koh et al., 2024b), filter trajectories to generate text-based memories or tools (Wang et al., 2024; Yu et al., 2024; Sarch et al., 2025) to boost agent performance in (Visual)WebArena (Zhou et al., 2023; Koh et al., 2024a), and for RL training in simpler environments (Bai et al., 2024).

**Test-time scaling** (TTS) is a major paradigm to improve model performance without increasing parameters (Snell et al., 2024; Welleck et al., 2024). Early work (Wei et al., 2022; Kojima et al., 2022) shows that prompting LLMs to generate "chains-of-thought" yields substantial gains in reasoning-

oriented tasks. This idea has been extended to several settings, including multimodal (Zhang et al., 2023) and environment-interaction (Zawalski et al., 2024; Yao et al., 2023b). Orthogonal approaches scale test-time compute via sampling and search (Yao et al., 2023a; Hao et al., 2023), where multiple generations are selected through heuristics (Wang et al., 2022) or verifiers (Shao et al., 2024a; Lightman et al., 2023). Recent work leverages sampling, RL, and formal verifiers to train (M)LLMs that autonomously generate reasoning traces (DeepSeek-AI, 2025; OpenAI, 2024; Kimi Team, 2025; Gemini Team, 2025). Extending these methods to open-ended problems requires flexible and multimodal verification, for which MLLMs offer an appealing solution.

Compared to this body of work, we **(1)** Unify applications of MLLM evaluators based on their use of multimodal rewards derived from MLLM verifiers. **(2)** Evaluate MLLM verifiers on *multimodal* settings across a range of models, benchmarks, agents, TTS techniques, evaluation templates, and applications. **(3)** Dissect MLLM verifier performance over several fine-grained metrics, offering guidance on how to measure the quality of their evaluations and artifacts derived from them. **(4)** Identify agreement bias, show its resilience to TTS techniques, and the risks it poses for downstream applications. **(5)** Discuss several design choices for building MLLM verifiers and introduce SGV, a simple yet effective method that can be easily integrated into pipelines involving MLLM verifiers. **(6)** Propose methods to improve MLLM verification with benefits extending to downstream applications such as online supervision and self-improvement, achieving a new state of the art on VisualWebArena.

## 3 PROBLEM SETUP

### 3.1 PRELIMINARIES AND DEFINITIONS

**Agents and Multimodal Verifiers.** Our goal is to study functions that approximate human judgment of agent behavior in interactive environments. Specifically, an agent is tasked to complete a task $q \in \mathcal{Q}$ through a series of actions $a_t$. Given a history of environment states $s_{r:t} \equiv [s_r, ..., s_t]$, the agent executes an action sampled from a policy $\pi$: $a_t = \pi(s_{r:t}, q)$, leading to a new state $s_{t+1}$. The repetition of this process yields a **trajectory** $\tau_{r:t} \equiv (s_r, a_r, \ldots, s_t, a_t)$ that, along with the task $q$, serves as the basis for evaluating agent performance. To evaluate trajectory-task pairs $(q, \tau_{r:t})$, humans process and produce information in multiple modalities, motivating our definition of a verifier and the scope of tasks considered. We define a **multimodal verifier** as a function $r : \mathcal{T} \times \mathcal{Q} \rightarrow \mathbb{R} \times (\mathcal{V} \cup \{\emptyset\})$, $r \in \mathcal{R}$, that maps trajectory-task pairs to rewards consisting of a real-valued score and optional outputs in other modalities from $\mathcal{V}$. These multimodal aspects are reflected in our settings. A task $q$ can be represented as images, text, or both, states $s_t$ as screenshots or DOM trees, actions $a_t$ as text (e.g. <type opaque phone>) or images from URLs, and verifier outputs as 0/1 scalars or natural language.

**Oracle Verifiers.** In the benchmarks we study, instantiations of $r$ are obtained via human-written scripts that have privileged access to task and environment states, which we refer to as **oracle** verifiers. They produce $0/1$ rewards reflecting task completion, which we treat as aligned with human judgments. These functions, however, are not perfect, and we discuss limitations in Secs. 3.2 and 4.

**Applications of Verifiers.** To characterize the potential benefits and risks of MLLM verifiers, we unify the view of several applications by how they use $r$ and are therefore affected by its quality, categorizing them into *offline* and *online*. The *offline* setting covers cases where $r$ is applied to trajectories post hoc. A canonical example is agent performance evaluation, where $r$ maps trajectories to scores reflecting task completion. Composite applications include methods that use $r$ to improve the agent's subsequent performance, such as filtering successful trajectories for finetuning (Pan et al., 2024a; Ma et al., 2024), as well as Reflexion-style refinement pipelines, where $r$ filters trajectories to induce prompts—e.g., memories, reflections (Shinn et al., 2023; Sarch et al., 2025; Yu et al., 2025), and tools (Wang et al., 2024; 2023)—that are included in the agent's context in subsequent executions. The *online* setting covers cases where $r$ influences policy distribution *during* execution. Applications include online supervision, where $r$ maps trajectories to scalars or text rewards to steer agents toward task completion and possibly update policy parameters (Bai et al., 2024; Wu et al., 2025), as well as using $r$ to rank state-action samples to guide (tree) search (Koh et al., 2024b; Zhou et al., 2024).

### 3.2 MLLM VERIFIERS, AGREEMENT BIAS, AND SELF-GROUNDED VERIFICATION

Human and scripted evaluation for open-ended tasks is hard to scale, motivating the use of MLLMs as an alternative. The usual approach to obtain $r$ with an MLLM is to prompt it with the task $q$,

trajectory $\tau_{r:t}$, and context $C$ that can include evaluation rubrics and instructions for reasoning steps:

$$r_{MLLM}(q, \tau_t, C) = h\big(\textstyle\prod_{i=1}^{n} P(y_i \mid y_{<i}, q, \tau_t, C)\big)$$

where $y = y_1, \ldots, y_n$ are MLLM outputs, and $h$ is a function mapping them to rewards—e.g., a regex that maps model completions to 0/1 scores or natural language feedback.

However, this approach is subject to a failure mode we term **agreement bias** (Figs. 7, 8 and 10): a tendency to over-validate agent behavior, judging flawed trajectories as aligned with the task $q$ despite the use of carefully crafted instructions $C$, established test-time scaling techniques, and decoding algorithms. This bias degrades the quality of MLLM-based rewards, $r_{MLLM}$, and can negatively impact several applications that rely directly or indirectly on them (Sec. 3.1).

Notably, this bias occurs despite MLLMs exhibiting strong, human-aligned priors on desired behavior—e.g., first-step generations in Figs. 1 and 7—that are not properly leveraged during verification, an issue that is unresolved by techniques eliciting intermediate reasoning steps. These observations align with results showing that the way a transformer encodes knowledge within its parameters during pretraining can hinder its extraction depending on how information is presented (Allen-Zhu and Li, 2024a;b), and that models may conflate truthfulness with human rater satisfaction due to inherent limitations of RLHF (Sharma et al., 2025). However, addressing these issues by modifying pretraining corpora, training recipes, or retraining large models remains both unclear and costly, highlighting the need for alternative solutions.

Motivated by this, we propose **Self-Grounded Verification (SGV)** (Fig. 1 (middle)), a simple yet effective method that substantially improves the performance of MLLM verifiers. SGV operates in two steps: first, the MLLM is elicited to extract broad priors $\hat{k}_q$ associated with successful completion of the task $q$, conditioned only on the data needed to frame the task:

$$\hat{k}_q = g\big(\textstyle\prod_{i=1}^{n} P(y_i \mid y_{<i}, s_{0:t}, C, q)\big)$$

where $s_{0:t}$ are states needed to define the task (e.g., an initial screenshot) and $g$ is a function to select a completion. In the second step, the MLLM evaluates the trajectory conditioned on the first step priors:

$$r_{\text{SGV}}(\tau_t, C, q) = h\big(\textstyle\prod_{i=1}^{n} P(y_i \mid y_{<i}, q, \tau_t, C, \hat{\boldsymbol{k}}_q)\big)$$

Intuitively, by modulating (un)conditional generation, SGV harnesses MLLMs' own sampling mechanisms to enable more effective use of their knowledge, alignment, and reasoning capabilities. Conditioning only on essential information in the first step encourages the model to explore its probability distribution freely, extracting knowledge pertinent to the task at hand and independent of the data under evaluation. In the second step, the MLLM evaluates a candidate trajectory by sampling from a conditional distribution induced by its own priors, for which we expect more balanced distributions and more accurate verification. We hypothesize that MLLMs, given their extensive world knowledge, can generally produce human-aligned priors on desired behavior that can serve as impartial references for grounding the verification, leading to more truthful and accurate verification.

### 3.3 EVALUATING MLLM VERIFIERS

This section outlines several factors considered for a reliable assessment of MLLM verifiers and for empirically demonstrating agreement bias and validating our hypotheses.

**Environment Diversity and Multimodality.** We consider benchmarks that require nuanced verification, multimodal reasoning, and collectively span a diverse range of tasks. VisualWebArena (VWA) (Koh et al., 2024a) emulates a web browser and spans 910 tasks, many of which combine text and image instructions (e.g., *"Buy this product"* + *<image>*). OSWorld (Xie et al., 2024) emulates a computer and comprises 369 tasks involving widely used desktop applications across both single- and multi-application workflows. Finally, robomimic (Mandlekar et al., 2021) provides long-horizon robot manipulation tasks, and we focus on *tool hang*, the most challenging among them, which consists of two subtasks: (1) inserting an L-shaped pencil into a base and (2) hanging a wrench on it. Sec. G.1 provides illustrations of representative tasks and trajectories.

**Trajectory Annotations.** Similar to prior work (Pan et al., 2024a; Xu et al., 2024; Huang et al., 2023), we use benchmark–provided oracle verifiers as proxies for human judgment due to the high cost of large-scale annotation. However, we observed several issues with the oracles in VWA, an issue also

noted in concurrent work (Men et al., 2025) comparing rule-based evaluation with human annotations. To establish a reliable reference, we corrected non-ambiguous issues in the VWA oracles—e.g., string parsing bugs, mismatches between task intents and oracle requirements, and incorrect annotations. To validate the effectiveness and impartiality of these changes, we evaluated the revised oracles on external labeled trajectories from (Men et al., 2025), observing near-perfect agreement with human judgments (Tab. 9). For details about these and other refinements to (Visual)WebArena, see Sec. F.

**Agents and Trajectory Quality.** Weak agents and buggy environments lead to trajectories that are trivial to verify and can artificially inflate verifier performance. For instance, some trajectories in (Men et al., 2025) exhibit long action loops, as well as "page not found" errors due to bugs in the browsergym (de Chezelles et al., 2025) suite. Moreover, incorporating trajectories generated by diverse methods is crucial to ensure the generalization of our findings. Therefore, we fixed several bugs in the (Visual)WebArena environments and considered agents built from different methods. To generate trajectories in VWA, we build a strong ReAct agent (Yao et al., 2023b) that yields a balanced ratio of successes and failures. For OSWorld, we employ the GUI-Specialist UI-TARS-1.5 (Qin et al., 2025), the best-performing agent on the benchmark at the time of this work. For robomimic, we train a diffusion policy on the expert demonstrations collected by (Zawalski et al., 2024).

**Choice of Verifier Applications.** MLLMs have been used to approximate $r$ in several of the applications discussed in Sec. 3.1. While exploring this whole range is infeasible, we focus on three representative cases that can inform general applicability: offline evaluation of agent trajectories, self-refinement via Reflexion, and online supervision. These applications: (1) introduce fewer confounding factors; (2) are of direct practical interest and often serve as building blocks in larger pipelines; and (3) yield informative signals for broader applicability. For instance, if $r_{\text{MLLM}}$ produces many false positives in trajectory evaluation, finetuning on those trajectories is likely to be affected. Similarly, if $r_{\text{MLLM}}$ yields weak rewards for online supervision or Reflexion, it is likely to be suboptimal for online training and more complex Reflexion-style self-refinement pipelines.

**Baselines and Ablations.** To probe limitations of MLLM verifiers and set strong baselines, we build verifiers with several methods, including chain-of-thought (CoT) (Wei et al., 2022) and set-of-marks (SoM) prompting (Yang et al., 2023), majority voting (Wang et al., 2022), and reasoning models. For VWA, we also consider the approach from Pan et al. (2024a), where we additionally provide benchmark-specific rubrics for evaluation. MLLMs are given full trajectories represented as sequences of screenshot-action pairs and asked to assign a ternary Likert label: `SUCCESS`, `PARTIAL SUCCESS`, or `FAILURE`, mapped to `[1, 0, 0]` to align with oracle scores. For reference, Tab. 10 shows that our **baseline MLLM verifier** is already strong, achieving **state-of-the-art performance on AgentRewardBench**. **SGV further improves** upon that, surpassing even the original VisualWebArena oracles. Sec. E ablates on all such choices, including model family and size, prompt/scoring templates, and SGV design and prior generation mechanism.

**Quantitative Metrics.** Verifiers should negatively (positively) evaluate trajectories marked as failures (successes) by humans or proxies to them. Evaluations consistently biased toward either side are likely undesirable. To capture the degree of alignment in MLLM responses, we evaluate (1) bias, (2) distance skewness, and (3) true positive and true negative rates:

$$\text{bias} = \frac{1}{n}\sum_i \mathbb{E}[d_i], \; \text{dSkew} = 1 - \frac{\sum_{ij}\|d_i - d_j\|}{\sum_{ij}\|d_i + d_j\|}, \; \text{T?R}(c) = \frac{\sum_i \mathbf{1}(\hat{r}_i = c \wedge r_i^* = c)}{\sum_i \mathbf{1}(r_i^* = c)} \approx \hat{P}(\hat{r}_i = c \mid r_i^* = c), \; c \in \{0, 1\}$$

where $\hat{r}_i$ is the MLLM verifier reward, $r_i^*$ is the human or oracle reward, and $d_i = \hat{r}_i - r_i^*$.

Ultimately, a verifier should also be evaluated by its downstream impact: if its feedback is effective, agent performance should improve. Therefore, **for downstream applications**, we use **task completion rates (SR)** of base agents with and without verifier interventions as the primary metric.

The following highlights key aspects regarding quantitative analysis of verifiers: **(1)** $bias$ and $dSkew$, also adopted by Xu et al. (2024), are summary statistics of the distribution of MLLM responses. Positive values indicate rewards systematically higher than those given by humans, whereas values near zero indicate closer alignment. **(2)** TNR measures how often MLLMs identify failures among trajectories marked as failures by humans, and is therefore an empirical estimate of the probability of classifying a trajectory as flawed when it truly is. Moreover, we use low TNR and high false-positive rate interchangeably, since $TNR = 1 - FalsePositiveRate$ (analogous for TPR). **(3)** While we report multiple metrics for robustness, we emphasize the practical importance of statistics such as TPR and TNR, which directly relate to the core function of verifiers: identifying flawed behavior and providing

feedback to improve agent performance. Low values indicate not only misalignment but also risks to downstream applications. **(4)** Accuracy (ACC) is included as an *auxiliary* metric for interpretation and comparison, as it provides a summary of TPR-TNR trade-offs: $ACC = (1 - \text{SR}) \cdot TNR + \text{SR} \cdot TPR$.

## 4 OFFLINE EVALUATION OF AGENT TRAJECTORIES

In this section we evaluate MLLM verification performance for about 1,300 trajectories generated in VisualWebArena (VWA) and OSWorld. Trajectories are based on `Gemini-2.5-Flash` in VWA and `UI-Tars-1.5` in OSWorld, with success rates of 47% and 22%, respectively. Tabs. 1 and 2 report performance across a range of models and test-time scaling techniques. Secs. A, D.1 and E provide breakdowns by environment and trajectory length, as well as other ablations.

Table 1: Verification of digital agent trajectories. (a) MLLMs tend to over-validate agent behavior, exhibiting positively skewed rewards, a high number of false positives (1-TNR), and a low probability of flagging failures (TNR as low as 50%). (b) SGV improves all metrics across all models and benchmarks.

| Model | (a) No SGV | | | | | (b) SGV | | | | | (b) - (a) | | |
|---|---|---|---|---|---|---|---|---|---|---|---|---|---|
| | Acc | TPR | TNR | Bias | dSkew | Acc | TPR | TNR | Bias | dSkew | Acc ↑ | Bias ↓ | dSkew ↓ |
| Gemini 2.0 | 61 | 96 | 42 | 36 | 34 | 69 | 94 | 55 | 27 | 23 | +9 | -9 | -11 |
| Gemini 2.5 Lite | 55 | 96 | 34 | 41 | 39 | 65 | 90 | 51 | 29 | 25 | +9 | -11 | -13 |
| Gemini 2.5 Flash | 68 | 94 | 55 | 27 | 24 | 80 | 88 | 76 | 12 | 8 | +12 | -15 | -16 |
| Gemini-2.5-Flash (T) | 74 | 92 | 64 | 21 | 18 | 82 | 89 | 78 | 10 | 6 | +8 | -11 | -13 |
| Qwen3-32b | 69 | 92 | 57 | 25 | 21 | 76 | 88 | 71 | 14 | 11 | +7 | -11 | -10 |
| Qwen3-235b-a22b | 65 | 93 | 51 | 28 | 25 | 76 | 89 | 70 | 15 | 12 | +10 | -14 | -13 |
| Qwen3-235b-a22b (T) | 66 | 92 | 53 | 27 | 24 | 77 | 91 | 71 | 15 | 12 | +11 | -12 | -12 |
| GPT-4.1 Mini | 60 | 96 | 40 | 37 | 35 | 74 | 92 | 65 | 17 | 14 | +14 | -20 | -21 |
| GPT-4.1 | 74 | 90 | 64 | 19 | 15 | 81 | 87 | 78 | 10 | 6 | +7 | -9 | -9 |
| GPT-o1 (T) | 70 | 80 | 62 | 22 | 16 | 78 | 83 | 73 | 13 | 7 | +7 | -9 | -8 |
| GPT-o4 (T) | 78 | 88 | 71 | 11 | 7 | 84 | 86 | 82 | 6 | 2 | +6 | -6 | -5 |
| GPT-5-Nano (T) | 72 | 84 | 65 | 16 | 11 | 76 | 82 | 73 | 10 | 6 | +4 | -6 | -5 |
| GPT-5 (T) | 81 | 86 | 78 | 8 | 4 | 86 | 85 | 87 | 2 | 1 | +5 | -6 | -3 |
| Llama-4-Maverick-17B-128E | 60 | 92 | 44 | 33 | 29 | 65 | 89 | 54 | 25 | 22 | +5 | -7 | -8 |

*(T) = Thinking enabled with the maximum thinking budget for the corresponding model.

**Agreement Bias.** MLLMs display a strong tendency to over-validate agent behavior. This manifests as a high number of false positives (1-TNR), responses tilted toward favorable evaluations (strictly positive bias and skewness), and a low probability of flagging failures (TNR as low as 50%). As shown in Tab. 2, this pattern persists despite techniques such as CoT and SoM (rows 3 and 4), majority voting (row 5), and the inclusion of task-specific evaluation criteria (row 6). Figs. 7, 9 and 10 illustrate a reason behind this pattern: MLLMs generate reasoning that rationalizes flawed trajectories and their wrong judgments. Sampling fails to mitigate this issue and may even exacerbate it, as higher temperatures can increase the likelihood of hallucinations and spurious correlations.

Table 2: Major test-time scaling techniques fail to mitigate agreement bias, whereas SGV improves all metrics.

| # | Method | VisualWebArena | | | | | OSWorld | | | | |
|---|---|---|---|---|---|---|---|---|---|---|---|
| | | Acc | TPR | TNR | Bias | dSkew | Acc | TPR | TNR | Bias | dSkew |
| 1 | No CoT | 64 | 91 | 44 | 28 | 24 | 68 | 97 | 60 | 30 | 30 |
| 2 | CoT | 65 | 90 | 47 | 26 | 21 | 71 | 97 | 63 | 28 | 27 |
| 3 | CoT (binary) | 59 | 90 | 36 | 34 | 29 | 69 | 99 | 61 | 30 | 30 |
| 4 | CoT (no SoM) | 64 | 91 | 44 | 28 | 24 | 71 | 99 | 64 | 28 | 28 |
| 5 | CoT (M) | 67 | 92 | 48 | 27 | 23 | 69 | 97 | 62 | 30 | 29 |
| 6 | Pan et al. (2024a) | 66 | 83 | 53 | 21 | 14 | – | – | – | – | – |
| 7 | Thinking | 70 | 90 | 55 | 23 | 18 | 78 | 95 | 73 | 20 | 18 |
| 8 | Thinking (M) | 70 | 91 | 55 | 24 | 20 | 76 | 95 | 71 | 22 | 20 |
| 9 | SGV | 76 | 84 | 71 | 11 | 5 | 83 | 92 | 81 | 13 | 11 |
| 10 | SGV (T) | 77 | 86 | 70 | 12 | 6 | 87 | 91 | 86 | 8 | 5 |

*(M) = majority voting; (T) = thinking enabled.
Oracle success rates are 47% in VisualWebArena and 22% in OSWorld.

Table 3: **(A):** Distribution of MLLM evaluations averaged over 28 scoring templates. **(B):** MLLM–oracle agreement rates over multiple samples stratified by trajectory status.

| (A) | S | PS | F/PF | U |
|---|---|---|---|---|
| Oracle | 57 | – | 43 | – |
| No SGV (binary) | 73 | – | 25 | 2 |
| No SGV | 70 | 6 | 23 | 1 |
| SGV | 56 | 4 | 39 | 1 |

| (B) | VisualWebArena | | | OSWorld | | |
|---|---|---|---|---|---|---|
| | All | F | S | All | F | S |
| No SGV | 65 | 48 | 91 | 71 | 56 | 93 |
| No SGV (T) | 68 | 55 | 90 | 79 | 65 | 93 |
| SGV | 75 | 70 | 86 | 85 | 77 | 89 |
| # Samples | 7,280 | | | 2,784 | | |

*(P)F/S denotes trajectories labeled as (Partial) Failure/Success, U as Uncertain.

Tab. 3 further contextualizes results through the distribution of MLLM responses. The top panel shows distributions averaged across 28+ evaluation templates that incorporate commonly used bias-mitigation strategies, such as criteria-order randomization and label rephrasing (see Sec. E.2

for details and all histograms). The bottom panel reports oracle–MLLM agreement rates over 10,000+ samples stratified by success and failure subsets, measuring the probability of sampling a correct verification from the MLLM output distribution. Tab. 3 (top) reveals that **(i)** MLLMs tend to concentrate responses in high score regions; **(ii)** binary scales (e.g., success/failure as adopted in (Pan et al., 2024a) and subsequent work) tend to amplify this bias (e.g., 72% SR vs. 57% expected); and **(iii)** increasing label granularity (as in our ternary Likert-scale) can partially mitigate this issue. Tab. 3 (bottom) shows that over-validation is reflected in the model's output distribution: on failure subsets, the probability of sampling a correct response from the MLLM is at or below chance (e.g., 48% in VWA). This imbalance explains the ineffectiveness of methods like majority voting (that rely on the mode of the distribution), model calibration (Sec. E.3.1), and opens opportunities for test- and training-time methods that account for it during sampling (see Sec. E.3.2 for a proof of concept).

**Cross-Model Variations and Verification Difficulty.** Agreement bias arises even when verifiers and agents are built from different model families, sizes, and methods—in Tab. 1, models such as GPT-4.1, and Qwen3 belong to distinct families and are stronger than the models used to build the agents. Likewise, it occurs for trajectories produced by weak and strong agents, as evidenced by the suboptimal performance in both OSWorld (22% SR) and VWA (47% SR)(Tabs. 2, 6 and 7) and weaker agents in VWA (Tab. 17). It is, however, more severe when models are weaker (e.g., GPT-4.1-mini) and for trajectories generated by stronger agents. This suggests that closing the gap between agent and verifier capabilities can alleviate (though not eliminate) the issue, whereas strategies such as building verifiers with (ensembles of) models that differ from the agents are insufficient.

**Adverse Impacts.** The bias toward over-validation means that MLLM verifiers fail precisely when most needed—when agent behavior is flawed and requires correction—with consequences extending beyond imprecise evaluation of agent performance. For example, for the task *"Buy the cheapest <product> from <category> with <attribute>"*, a trajectory as in Fig. 7 where the agent searches for *"<product>"*, clicks the first result, and adds it to the cart, is deemed successful despite omitting steps such as filtering, sorting, and completing the checkout. Fig. 10 illustrates an even more extreme case: the agent produces an answer unsupported by any trajectory information, yet an MLLM verifier validates it—even though, by construction, no information exists to justify such a judgment.

**Relation to Other Biases.** Agreement bias has distinct characteristics from other biases identified in LLM literature, such as "self-bias" (Xu et al., 2024; Panickssery et al., 2024; Chen et al., 2025b)—where models favor their own text generations and for which external knowledge injection is a typical remedy—and biases attributable to positional or phrasing dependencies in evaluation templates (Zheng et al., 2023; Chen et al., 2025a; Ye et al., 2024). In our settings, agreement bias arises despite the verifier's inputs omitting text directly produced by agents, and including *multimodal* inputs augmented with *external* information derived from truthful environment data, such as screenshots augmented with Set-of-Marks, and low-level action representations enriched with DOM data. Likewise, it persists when agents and verifiers are built from different methods and models and despite interventions such as label shuffling and rephrasing, as well as access to grounding tools (Tab. 3 and Secs. E.2 and E.8).

**Effectiveness of SGV.** SGV improves verification across all metrics, models, and benchmarks, increasing TNR by up to 25 percentage points (pp), overall accuracy by up to 14pp, and promoting responses more aligned with human preferences. This is reflected in reduced bias and skewness, and in more balanced response distributions across 28+ evaluation templates (see Tab. 3(top)), as well as in the models' output distribution (Tab. 3(bottom)). These gains hold even in settings where the verifier is weaker than the generator (e.g., Gemini 2.0 and GPT-4.1-Mini; see also Sec. E.6). Moreover, SGV outperforms instructions with task-specific evaluation criteria (Tab. 2, row 5), indicating it enables models to generate completions to condition themselves that can surpass human-written rubrics, offering a more scalable alternative. Finally, additional results in Sec. E.8 show that SGV outperforms grounding via web-search tools and is robust to moderate noise in priors generated in the first step; that weaker models can produce effective priors for stronger models and models of different families; and that multiple and diverse priors in the first step can further improve performance.

Remarkably, SGV (i) enables non-reasoning models to match the performance of reasoning counterparts, and (ii) boosts the accuracy of reasoning models by up to 11pp—a perhaps surprising result, given that LRMs are explicitly trained to produce intermediate traces, and interventions can degrade performance (OpenAI, 2025). These results align with intuitions about limitations stemming from earlier phases of LLM training, raising questions about the potential benefits of augmenting

reasoning-oriented training with an SGV step, pretraining data rewriting (Allen-Zhu and Li, 2024a;b), or methods that account for imbalances in MLLMs' output distributions.

**Fine-Grained Metrics.** These results highlight the importance of reporting fine-grained metrics in works proposing MLLMs in evaluative roles and artifacts produced by them. Given the trade-offs inherent to verification, it is crucial to report statistics capturing *both* the probability of identifying correct behavior and especially, flawed behavior. Reporting only single or aggregate metrics can be misleading. As shown in Tabs. 1 and 2, verifiers can display about 97% recall and 70% accuracy, yet misclassify ~50% of failed trajectories as successes, which can severely harm downstream applications.

**Leniency-Strictness Trade-offs in Verification.** For some models, SGV can lead to a lower TPR. This pattern stems from two main causes: disagreements with lenient oracles on simplistic tasks and stricter verification. Specifically, evaluation in these digital benchmarks is based on hard-coded rules applied to a subset of states. This means that trajectories such as in Fig. 9 where an agent "Buy the cheapest <product>," by searching for "<product>," and purchasing the first result are deemed successful by oracle scripts. Influenced by agreement bias, MLLMs tend to validate such trajectories, but when SGV is applied, this behavior is rejected for lacking steps that confirm that the item is the cheapest, thereby reducing TPR. On the flip side, Fig. 15 illustrates a case where SGV only validates an otherwise correct trajectory after the agent performs extra steps to double-check item attributes.

These examples highlight the challenges of open-ended verification, as well as the potential and limits of MLLMs as alternatives. For humans, it is readily apparent that behaviors such as in Fig. 9 do not generalize. Yet, translating this intuition into precise rules is far from trivial: stricter criteria can reject valid solutions, whereas permissive ones may encourage brittle behavior. MLLMs, like humans, offer flexibility in interpretation, but this comes at the cost of formal guarantees and vulnerabilities—with agreement bias being a strong one. In the next section, we analyze the impact of these trade-offs on downstream applications and show that SGV interventions are typically non-disruptive (Fig. 15), ultimately improving task completion through more generalizable behavior.

## 5 DOWNSTREAM APPLICATIONS

In this section, we evaluate MLLM verifiers on downstream applications, focusing on their ability to boost agent performance in self-improvement and online supervision. These are applications of direct interest that, so far, have shown limited benefits and occasional performance degradation (Pan et al., 2024b; Huang et al., 2023; Shinn et al., 2023; Stechly et al., 2024; Pan et al., 2024b; Shinn et al., 2023; Huang et al., 2023; Kamoi et al., 2024; Stechly et al., 2024). Importantly, these experiments provide an assessment of verifiers' net impact, given the trade-offs discussed in Sec. 4. Below we discuss main results; for experimental details and qualitative analysis, see Secs. A.2 and C.2.

**Self-Improvement**. In this setting, after each episode: (i) a verifier evaluates the trajectory; (ii) the agent reflects on its previous attempt conditioned on the verifier's evaluation; and (iii) reflections are given to the agent in subsequent attempts to enable self-correction over time. As shown in Fig. 2, under oracle verifier supervision, task success rate (SR) increases by up to 21 pps after three iterations, confirming the feasibility and defining an upper bound for self-refinement. When supervised by the (strong) baseline MLLM verifier, the base agent's performance quickly plateaus, yielding minimal improvements, whereas SGV enables consistent progress, boosting SR by up to 10.4 pps. This pattern demonstrates how agreement bias limits the effectiveness of MLLM verifiers. Referring back to Sec. 4, a TNR close to 50% implies that the verifier fails to provide a corrective signal in about half of the cases where agent behavior is flawed, hindering its ability to self-improve.

**Online Supervision**. Similarly, agreement bias impairs the effectiveness of MLLM-derived rewards for online methods. Tab. 4 shows that the (strong) baseline verifier fails to meaningfully improve digital agent performance. In contrast, SGV boosts SR by 9 percentage points (pp) on VisualWebArena (20% relative) and by 5 pp on OSWorld (22% relative). Notably, our agent sets a new **state of the art on VisualWebArena**, surpassing the previous best by 20 pp (58% relative) while requiring substantially fewer tokens, no access to prior trajectories, and using only native web actions.

A few factors explain these results. First, SGV identifies suboptimal behavior and provides feedback that enables agents to backtrack and complete the task (e.g., Fig. 13). Without SGV, agreement bias prevents verifiers from intervening precisely when agent behavior is flawed, limiting their effectiveness (e.g., Fig. 7). Second, while SGV can lead to stricter verification, interventions are mostly

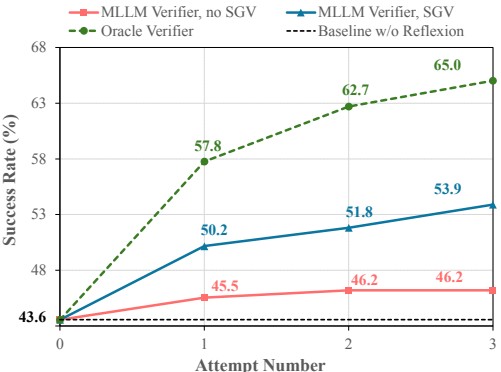

Figure 2: **Self-Improvement in VisualWebArena.** Agreement bias prevents MLLM verifiers from providing corrective feedback when agent behavior is flawed, hindering its ability to self-improve. SGV reduces agreement bias, yielding consistent gains across refinement iterations.

Table 4: Task success rate and token usage (in parenthesis) on VisualWebArena and OSWorld.

| Method | VisualWebArena | | OSW |
|---|---|---|---|
| | All | S / C / R | All |
| Search Agent Koh et al. (2024b) | 29 (4.9x) | 34 / 22 / 30 | – |
| R-MCTS Yu et al. (2024) | 34 (9.3x) | 41 / 29 / 32 | – |
| Human | 89 (–) | 88 / 91 / 87 | – |
| Base Agent | 45 (1x) | 50 / 35 / 48 | 22 |
| + Verifier, no SGV | 46 (1.5x) | 52 / 36 / 49 | 24 |
| **+ Verifier, SGV** | **54 (2.2x)** | **56 / 43 / 58** | **27** |

*Averages across 3 runs. S/C/R denote shopping, classifieds, and reddit subsets of VisualWebArena.

Table 5: Diffusion policy steering on robomimic.

| Method | FR | PSR | SR | # Replans |
|---|---|---|---|---|
| Diffusion Policy | 32 | 68 | 24 | 88 |
| + Verifier, no SGV | 22 | 78 | 16 | 89 |
| **+ Verifier, SGV** | **28** | **72** | **32** | **96** |

*FR, PSR and SR are the proportions of roll-outs with 0, 1 and 2 subtasks (out of 2) completed.

non-disruptive. For example, in Fig. 15 the verifier rejects a greedy strategy that technically satisfies the benchmark, prompting the agent to search for and confirm user-requested attributes, ultimately leading to task completion through a more robust approach. Fig. 3 quantifies this pattern: without SGV, the verifier endorses flawed behavior in 30% of the tasks, offering no signal for improvement. With SGV, 10% of tasks improve due to accurate failure detection, and among the 7.4% of tasks marked as false negatives, 6.6% remain successful, whereas only 0.8% transition to failure. Likewise, robomimic results (Tab. 5) show that the baseline MLLM verifier is able to guide the policy to partial completion (high PSR). However, influenced by agreement bias, it struggles to further guide the policy toward full completion, as evidenced by the low number of replans and SR *lower* than the baseline. In contrast, SGV triggers replans more often, improving over the baseline by 8 pp. For a categorization of factors influencing success rates and the impact of verifier design choices, see Secs. C.2 and E.4.

**Discussion**. These results illustrate the risks posed by agreement bias: MLLMs' tendency to over-validate agent behavior leads to unreliable evaluations precisely when most needed—when behavior is flawed and requires correction—limiting their effectiveness for trajectory selection and online feedback. Notably, this holds for an MLLM verifier with **strong** numbers in judging trajectories post-hoc: over 91% recall (higher than SGV) and state-of-the-art performance (precision) on AgentRewardBench. This highlights the importance of evaluating verifier performance using (i) fine-grained metrics and (ii) downstream applications, as leniency–strictness trade-offs in verification can render post-hoc metrics insufficient to characterize a verifier's practical utility.

## 6    CONCLUSION, LIMITATIONS, AND FUTURE WORK

We identify agreement bias, a critical limitation that hinders MLLMs from serving as verifiers of agent behavior. We demonstrate its adverse effects on existing applications, discuss methods for evaluating and improving MLLM-based verification, and introduce SGV, a simple yet effective method that improves verification across multiple models and environments. Although SGV mitigates agreement bias, it does not eliminate it. Our qualitative analysis (Sec. C.1) indicates that remaining failures often stem from limitations in base models' integration of visual perception and language. Future work may explore methods to enhance these capabilities, such as integrating visual experts for fine-grained perception, potentially yielding gains complementary to SGV. In parallel, compelling directions for open-ended verification include combining MLLMs with symbolic methods (Kambhampati et al., 2024) and training- or test-time strategies that account for skewness in MLLM output distributions associated with agreement bias (Sec. E.3). Finally, an important question concerns *when* to invoke a verifier and the relative value of process and outcome-based verification, particularly in digital environments where state spaces are often discrete, and actions can be irreversible or destructive (Sec. E.4).

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

## A  EXPERIMENTAL DETAILS AND PRACTICAL IMPLEMENTATION

### A.1  OFFLINE EVALUATION OF AGENT TRAJECTORIES

For the experiments reported in Tab. 1, we employ each model with its author-recommended inference parameters. For the ablations in Tab. 2 and Sec. E, we employ `gemini-2.5-flash` with zero temperature, except for the majority voting baselines, where we select the most frequent response among 8 completions for each trajectory generated with a `temperature` of 1.5, `top-p` of 0.95, and `top-k` set to 64.

For non-thinking models, all settings include a CoT instruction (see prompts in Sec. B). For thinking models we set thinking budgets to the maximum value for each model and omit CoT instructions, as they inherently produce reasoning traces and adding them can degrade performance OpenAI (2025).

Models are given interleaved images and text, in the form of screenshots produced by the environment and actions generated by the agent after they are parsed by the environment. In the case of ID-based actions, we overlay bounding boxes on the screenshots to indicate the location of each interactable element. In the case of coordinate-based actions, we overlay red markers to indicate the target of mouse actions. For UI-Tars-1.5-7b, we additionally translate agent outputs to english using `gemini-2.0` as the agent sometimes produces answers in chinese. Sec. G show examples of trajectories, and Sec. B contains details about prompt templates.

## A.2 Downstream Applications

For our results in downstream applications, we employ `gemini-2.5-flash` as the MLLM to build the verifiers, with temperature set to 0, thinking disabled, and a token limit of 8192. The prompt templates and inputs given to the verifier are the same as the ones in the offline scenario described in Sec. B.

The numbers reported are averages across three runs in the digital benchmarks, and 50 rollouts with a horizon of 700 time steps in robomimic.

**Reflexion – Implementation Details.** After each episode for a given task, the same agent that generates the trajectory is prompted to reflect on its performance, using the verifier's score as part of the context. This reflection is saved to an external memory, that is appended to the agent's context window in subsequent episodes. The prompt is almost the same as the agent's prompt, and is provided in Sec. B.6.

**Online Supervision – Verifier and Agent Implementation Details.**

For VisualWebArena experiments in Sec. 5, we use `gemini-2.5-flash` for both the ReAct agent and the verifier. For the no-SGV case, the model is given a CoT instruction as in Sec. B. The environment settings follow those in Koh et al. (2024a): we adopt Set-of-Mark prompting Yang et al. (2023) for the screenshots, with a viewport size of 1280x2048 pixels. For consistency with prior work, we use BLIP-2 Li et al. (2023) to provide text captions for images making part of the task objectives and environment observations (though we observe little to no difference in performance when captions are omitted). We set the maximum number of steps to 30, which allows the agent some room for backtracking while keeping the search budget comparable to—though still more constrained than—prior approaches that employ tree-search methods Koh et al. (2024b); Yu et al. (2024). For the OSWorld experiments, we follow the same setup and implementation as in Qin et al. (2025). We allow up to 50 steps for the agent and run UI-TARS-1.5-7B on a single RTX 3090 GPU.

For VisualWebArena, the MLLM acts as an **outcome verifier**, intervening when the agent issues a stop action. If it deems the task as incomplete, it generates natural language feedback that is appended to the agent's context window to influence subsequent generations.

For OSWorld, we experiment with two configurations: one where the MLLM intervenes only after stop actions, and another where it verifies progress every 5 steps (shown in our main results). The latter configuration was inspired by preliminary experiments showing that UI-Tars-1.5 can derail from the task objective and rapidly reach points difficult to recover from. As shown in Tab. 8, SGV improves performance in both cases, although results are slightly better in the latter.

In robomimic, a verifier monitors task progression and triggers the diffusion policy to regenerate an action sequence (or "replan") at determined steps. In the oracle scenario, the steps are determined by ground-truth subtask completions. In the MLLM verifier scenario, the MLLM receives a natural language description of the task, the first and last screenshots, and is called every 20 time steps. If it deems the trajectory off-track, a replan is triggered.

Sec. B.4 shows how the feedback is incorporated, and figures in Sec. G provide examples of evaluations and generated feedback.

## A.3 Practical Implementation

SGV introduces low compute overhead and generally improves performance without complex prompt engineering, making it simple to integrate into pipelines where MLLMs play evaluative roles. For instance, adding the first step with a single-line instruction to produce domain knowledge generally helps, with little to no change to existing prompts (see Sec. E.7). Nonetheless, we observe the following patterns that may inform general use:

**1)** Generating broader, more encompassing priors in the first step tends to be superior.

**2)** Producing priors in a single, monolithic step degrades performance, as models tend to anchor them to the data under evaluation.

**3)** Some models benefit when the second step includes an instruction to compare against their priors—something SGV inherently enables—although a standard CoT instruction typically suffices. See Sec. B for prompts and our codebase for more examples.

We also refer readers to Sec. E, which provides further guidance on MLLM verifier design, including analyses of the performance of Likert, numeric, and other prompt templates and interventions; the effects of noise, model scale, and diversity in SGV prior generation; the potential for calibrating MLLM output distributions to account for the skew introduced by agreement bias; comparisons between periodic and outcome-based verification; and the impact of trajectory length and verification difficulty on MLLM verification performance.

## B  PROMPTS

For all prompts, content enclosed in curly braces (`{...}`) is dynamically replaced at runtime with the corresponding information. Text enclosed in double angle brackets («...») provides explanatory notes for the reader and is not included in the prompts at runtime. The labels `USER:` and `ASSISTANT:` indicate role fields passed via the OpenAI API format and are not literal strings in the prompt. Text separated by a pipe (`|`) inside curly braces (`{...}`) indicates phrases that change depending on the context.

### B.1  COMMON PROMPTS

The following three prompt templates are common across all verifier and reflexion prompts, with slight variations to accommodate specific details such as image annotation mode and environment specifics.

---

**Trajectory Prompt - VisualWebArena and OSWorld**

```
USER: ## OBJECTIVE: {task objective text} {task objective images}

USER: ### STATE `t-j` SCREENSHOT: {screenshot}
ASSISTANT: {generated action}
...
USER: ### STATE `t-1` SCREENSHOT: {screenshot}
ASSISTANT: {generated action}
```

---

**Image Information Prompts - VisualWebArena**

```
<<if set-of-mark prompting is used (baseline)>>
### Webpage screenshots: These are screenshots of the webpage, with each interactable
element assigned a unique numerical id. Each bounding box and its respective id shares
the same color.

<<if set-of-marks and coordinate annotations>>
### Webpage screenshots: These are screenshots of the webpage, with each interactable
element assigned a unique numerical id. Each bounding box and its respective id shares
the same color. The colored markers, if present, indicate the destination of actions
taken at the corresponding state.

<<if coordinates are used>>
### Webpage screenshots: These are screenshots of the webpage taken at each step of the
 navigation. The colored markers, if present, indicate the destination of actions taken
 at the corresponding state.

<<if no visual annotations>>
### Webpage screenshots: These are screenshots of the webpage taken at each step of the
 navigation.
```

---

```
Image Information Prompts - OSWorld

<<if set-of-mark prompting is used>>
### Screenshots: These are visual captures of the computer screen taken at specific
states of the
navigation process, with each interactable element assigned a unique numerical id. Each
 bounding box and its respective id shares the same color.

<<if set-of-marks and coordinates annotations>>
### Screenshots: These are visual captures of the computer screen taken at specific
states of the
navigation process, with each interactable element assigned a unique numerical id. Each
 bounding box and its respective id shares the same color. The colored markers, if
present, indicate the destination of mouse actions taken at the corresponding state.

<<if coordinates annotations only (baseline)>>
### Screenshots: These are visual captures of the computer screen taken at specific
states of the
navigation process. The colored markers, if present, indicate the destination of
actions taken at the corresponding state. <<if coordinates are used>>

<<if no visual annotations>>
### Screenshots: These are visual captures of the computer screen taken at specific
states of the
navigation process.
```

## B.2 VERIFIER PROMPTS

To maximize consistency and minimize sensitivity due to prompt variations, we construct all verifier prompts from the following parts:

- The system prompt, which describes the role of the agent and environment details.
- The trajectory prompt, which is created in an interleaved user-assistant conversation format, and contains information on the task to be accomplished, screenshots of the environment states, and the actions taken by the agent being evaluated.
- The evaluation prompt, which includes (i) the criteria for the evaluation, and (ii) instructions for step-by-step reasoning. Check Secs. B.9 and E.2 for ablations to the scoring templates. Ablations to (ii) are indicated in the prompts below, and discussed in the main text (Sec. 4) and Secs. B.8 and E.

We aim to keep the prompts as consistent as possible across experiments, applying slight adjustments to accommodate environment-specific details and prompting variations. For instance, the NoCoT variant in section 4 is nearly identical to the CoT variant, differing only in the removal of the step-by-step reasoning instructions.

```
Verifier System Prompt - VisualWebArena

You are an intelligent agent tasked with supervising an assistant navigating a web
browser to accomplish a web-based task. Your job is to evaluate the assistant's work,
and provide feedback so it can progress towards the objective.

## Here's the information you'll have:
### The objective: This is the task the assistant is trying to complete.
{image_info_prompt_part}
### The execution trace: This is a sequence of webpage screenshots paired with the
assistant's actions, detailing the web navigation so far.
### General web knowledge: This is a general description of how tasks like this are
typically accomplished on the web. <<Only included for SGV variants>>

## Assistant's capabilities: To effectively analyze the assistant's work, consider the
actions it can perform. These actions fall into the following categories:
### Page Operation Actions:
```click [id]```: Click on an element with a specific id on the webpage.
```type [id] [content] [enter_after]```: Type the content into the field with id. If `
enter_after` is 0, the "Enter" key is not pressed after typing; otherwise, the "Enter"
key is automatically pressed.
```

```
```hover [id]```: Hover over an element with id.
```press [key_comb]```: Press a keyboard key or key combination (e.g., delete, ctrl+a).
```scroll [down]``` or ```scroll [up]```: Scroll the webpage up or down.

### Tab Management Actions:
```new_tab```: Open a new, empty browser tab.
```tab_focus [tab_index]```: Switch the browser's focus to a specific tab using its
index.
```close_tab```: Close the currently active tab.

### URL Navigation Actions:
```goto [url]```: Navigate to a specific URL.
```go_back```: Navigate to the previously viewed page.
```go_forward```: Navigate to the next page (if a previous 'go_back' action was
performed).

### Completion Action:
```stop [answer]```: This action is issued when the task is believed to be complete. If
 the task requires a text-based answer, the answer is provided within the brackets. If
the task is deemed impossible, this action is issued optionally with a reason why.

## To be successful, it is very important to consider the following:
1. You must not not assume the assistant's work is correct or incorrect beforehand. You
 should come up with your own opinion based on the information provided.
2. You must connect the dots between the objective and the information provided.
3. Your evaluation should be based on task completion; don't worry about efficiency for
 now.
4. Give utmost importance to the visual information provided, especially the
screenshots in the execution trace.
5. The General web knowledge can contain noise or information not relevant to your
specific context. Therefore, you should selectively use it to guide your analysis and
reasoning, making sure to **consider the context of the specific task and web
navigation given to you.** <<Only included for SGV variants>>
```

## Verifier System Prompt - OSWorld

```
You are an intelligent agent tasked with supervising an assistant utilizing a computer
to accomplish computer-based tasks. Your job is to evaluate the assistant's work, and
provide feedback so it can progress towards the objective.

## Here's the information you'll have:
### The objective: This is the task the assistant is trying to complete.
{image_info_prompt_part}
### The execution trace: This is a sequence of screenshots of the computer screen
paired with the assistant's responses, detailing the computer navigation so far.
### General Computer Knowledge: This is a general description of how tasks like this
are typically accomplished on a computer. <<Only included for SGV variants>>

## Assistant's capabilities: To effectively analyze the assistant's work, consider the
actions it can perform. These actions fall into the following categories:
### Mouse Actions:
- click(start_box='<|box_start|>(x1,y1)<|box_end|>'): Left clicks at the (x1,y1)
coordinates.
- left_double(start_box='<|box_start|>(x1,y1)<|box_end|>'): Left double click at the (
x1,y1) coordinates.
- right_single(start_box='<|box_start|>(x1,y1)<|box_end|>'): Right click at the (x1,y1)
 coordinates.
- drag(start_box='<|box_start|>(x1,y1)<|box_end|>', end_box='<|box_start|>(x3,y3)<|
box_end|>'): Drags the element at the (x1,y1) coordinates to the (x3,y3) coordinates.
- scroll(start_box='<|box_start|>(x1,y1)<|box_end|>', direction='down or up or right or
 left'): Scrolls the mouse wheel in the specified direction.

### Keyboard Actions:
- hotkey(key=''): Press a specific key.
- type(content=''): Types the string following `content`. If followed by "\n", it means
 the input is submitted.
- wait(): Pauses for 5s and takes a new screenshot to check for any changes.

### Completion Action:
- finished(content='<|content|>'): This action is issued when the task is believed to
be complete. `<content>` is optional, and is used to provide a reason why the task is
complete.

## To be successful, it is **very important** to follow the following rules:
```

```
1. You must not not assume the assistant's work is correct or incorrect beforehand. You
   should come up with your own opinion based on the information provided.
2. You must connect the dots between the objective and the information provided.
3. Give utmost importance to the visual information provided, especially the
screenshots in the execution trace.
4. Your evaluation should be based on task completion; don't worry about efficiency for
   now.
5. The General computer knowledge can contain noise or information not relevant to your
   specific context. Therefore, you should selectively use it to guide your analysis and
reasoning, making sure to **consider the context of the specific task and computer
navigation given to you.**<<Only included for SGV variants>>
```

**Full Verifier Prompt**

```
{system prompt}
{trajectory prompt}

USER:
## General web knowledge: {output from first step} <<Only included for SGV variants>>

Now please provide your response.

## Here is the evaluation criteria:<<This section is changed in ablations to evaluation
 templates, e.g., numerical scoring>>
SUCCESS: The assistant executed **all of** what's necessary to complete the objective.
The task is fully accomplished.
PARTIAL SUCCESS: The assistant executed **most of** what's necessary to complete the
objective. The task is partially accomplished.
FAILURE: The assistant executed **mostly incorrect** steps. The task is not
accomplished, and major revisions are needed.

## Provide your response as follows:
REASONING: [Comprehensive step-by-step reasoning to come up with your evaluation and
feedback] <<Only included for CoT variants>>
EVALUATION: [Your evaluation following the evaluation criteria]
FEEDBACK: [Feedback so the assistant can progress towards the objective]
```

**Pan et al. (2024a) Verifier System Prompt**

```
You are an expert in evaluating the performance of a web navigation agent. The agent is
 designed to help a human user navigate a website to complete a task. Given the user's
intent, the agent's action history, the final state of the webpage, and the agent's
response to the user, your goal is to decide whether the agent's execution is
successful or not.

There are three types of tasks:
1. Information seeking: The user wants to obtain certain information from the webpage,
such as the information of a product, reviews, map info, comparison of map routes, etc.
 The bot's response must contain the information the user wants, or explicitly state
that the information is not available. Otherwise, e.g. the bot encounters an exception
and respond with the error content, the task is considered a failure. Besides, be
careful about the sufficiency of the agent's actions. For example, when asked to list
the top-searched items in a shop, the agent should order the items by the number of
searches, and then return the top items. If the ordering action is missing, the task is
 likely to fail.
2. Site navigation: The user wants to navigate to a specific page. Carefully examine
the bot's action history and the final state of the webpage to determine whether the
bot successfully completes the task. No need to consider the bot's response.
3. Content modification: The user wants to modify the content of a webpage or
configuration. Carefully examine the bot's action history and the final state of the
webpage to determine whether the bot successfully completes the task. No need to
consider the bot's response.

*IMPORTANT*
Format your response into two lines as shown below:

Thoughts: <your thoughts and reasoning process>
Status: "success", "partial success", or "failure"
```

---

**Full Pan et al. (2024a) Verifier Prompt**

```
{system prompt}

USER:
## User Intent: {task objective text}

## Action History:
1: {generated action}
...
n: {generated action}

## Last snapshot of the webpage: {screenshot}
```

---

## B.3   SGV VERIFIER - FIRST STEP PROMPTS

For the main experiments, we use separate prompts for the retrieval step and the evaluation step.
Below, we present the prompts for the first step. As shown in Sec. B.8, similar performance can
be achieved by reusing the prompts from Sec. B.2, as long as the verification process is explicitly
divided into retrieval and evaluation phases.

---

**SGV - First Step System Prompt**

```
You are a helpful assistant with deep knowledge in {web navigation | computer-based
workflows}.
Your job is to provide a description of how tasks like the ones provided to you are
typically accomplished on {the web | computers}.

## Here's the information you'll have:
### Objective: This is an english description of the task, possibly accompanied by
images that provide more context on what must be accomplished.
### Screenshots: These are screenshots of the {webpage | computer screen}, giving you
context to further understand what must be accomplished.
```

---

**SGV - Full First Step Prompt**

```
{system prompt first-step}
## OBJECTIVE: {task objective text}

## IMAGES:
Image 0: {task objective image 1}
...
Image n: {task objective image n}
Image n+1: initial {webpage | computer} screenshot: {screenshot}

Now please provide your response:
<Description of how tasks such as this are typically accomplished on {the web |
computers}>.
```

---

## B.4   VISUALWEBARENA AGENT PROMPT

The prompt for VisualWebArena is built upon the authors' original implementation Koh et al. (2024a),
with the following changes:

- We provide the history of thoughts and actions generated by the agent in previous interactions,
  which serve as a text-based memory to prevent loops and allow backtracking.
- We refine parts of the prompt to use Markdown formatting and to make the instructions more
  precise.
- For one of the examples, we include previous thoughts and actions to match the new inputs.

---

**VisualWebArena Agent Prompt**

```
{system prompt}
{examples}

## RATIONALE AND ACTION HISTORY:
### STATE `t-j`:
- **RATIONALE**: {agent generations}
- **ACTION**: {action parsed by the environment}
...
### STATE `t-1`:
- **RATIONALE**: {agent generations}
- **ACTION**: {action parsed by the environment}

Here is the current state `t` for you to determine what to do next:
## TEXT OBSERVATION `t`: {text observation}

## URL: {webpage URL}

## OBJECTIVE: {task objective text}

## FEEDBACK: Here is your previous response at the current state `t` and **feedback**
about it. Use this to revise your response if you deem appropriate.
### Previous response: {last generated action}
### Feedback: {verifier feedback} <<Only included in rounds where feedback is provided
by the verifier>>

IMAGES:
Image 0: {task objective image 0}
...
Image n: {task objective image n}
Image n+1: current webpage screenshot at state `t`: {screenshot}
```

## B.5 OSWORLD AGENT PROMPT

---

**OSWorld Agent Prompt**

```
You are a GUI agent. You are given a task and your action history, with screenshots.
You need to perform the next action to complete the task.

## Output Format
```
Thought: ...
Action: ...
```

## Action Space

click(start_box='<|box_start|>(x1,y1)<|box_end|>')
left_double(start_box='<|box_start|>(x1,y1)<|box_end|>')
right_single(start_box='<|box_start|>(x1,y1)<|box_end|>')
drag(start_box='<|box_start|>(x1,y1)<|box_end|>', end_box='<|box_start|>(x3,y3)<|
box_end|>')
hotkey(key='')
type(content='') #If you want to submit your input, use "\n" at the end of `content`.
scroll(start_box='<|box_start|>(x1,y1)<|box_end|>', direction='down or up or right or
left')
wait() #Sleep for 5s and take a screenshot to check for any changes.
finished(content='xxx') # Use escape characters \', \", and \n in content part to
ensure we can parse the content in normal python string format.

## Note
- Use English in `Thought` part.
- Write a small plan and finally summarize your next action (with its target element)
in one sentence in `Thought` part.

## User Instruction
{task objective text}
```

## B.6 REFLEXION PROMPT

---

**Reflexion System Prompt**

```
You are an autonomous intelligent agent that was tasked with navigating a web browser
to complete web-based tasks.
Your goal now is to analyze your behavior on a previous task attempt. You must analyze
the strategy and path you took when trying to complete the task, understand the reason
why you {failed|succeded}, and devise **concise** reflections that can be helpful when
you are solving the same task in the future.

## Here's the information you'll have:
### The objective: This is the task you were trying to complete.
### The execution trace: This is a sequence of webpage screenshots paired with your
responses, detailing the web navigation from the beginning to the end of the previous
task attempt.
### Previous reflections: This is a list of previous reflections you've made about
other failed executions of this task.
### Webpage screenshots: These are screenshots of the webpage, with each interactable
element assigned a unique numerical id. Each bounding box and its respective id shares
the same color.

## To analyze your past task attempts, consider the actions that were available to you.
 These actions fall into the following categories:
### Page Operation Actions:
```click [id]```: This action clicks on an element with a specific id on the webpage.
```type [id] [content] [enter_after]```: Use this to type the content into the field
with id. If `enter_after` is 0, the "Enter" key is not pressed after typing; otherwise,
 the "Enter" key is automatically pressed.
```hover [id]```: Hover over an element with id.
```press [key_comb] [text]```: Press a keyboard key or key combination (e.g., delete,
ctrl+a) with optional text input if the key combination requires it (e.g., ```press [
ctrl+f] [some_text]```).
```scroll [down]``` or ```scroll [up]```: Scroll the webpage up or down.

### Tab Management Actions:
```new_tab```: Open a new, empty browser tab.
```tab_focus [tab_index]```: Switch the browser's focus to a specific tab using its
index.
```close_tab```: Close the currently active tab.

### URL Navigation Actions:
```goto [url]```: Navigate to a specific URL.
```go_back```: Navigate to the previously viewed page.
```go_forward```: Navigate to the next page (if a previous 'go_back' action was
performed).

### Completion Action:
```stop [answer]```: Issue this action if you believe the task is complete or
infeasible. If the objective is to find a text-based answer, provide the answer in the
bracket. If you deem the task is infeasible, provide a reason why.

## To be successful, it is **very important** to consider the following:
1. You should carefully analyze the execution trace to come up with your response.
2. You must connect the dots between the objective and the information provided.
3. The execution trace is from a previous task attempt that is **finished**. Therefore,
 you should not try to continue the task nor propose actions continuing from the
previous task attempt.
4. Prioritize key and general aspects. Your reflections should be applicable to future
attempts to this task that will be **independent** from previous task attempts and will
 start from the beginning.
5. Try to to think differently from previous task attempts and reflections. Integrate
information from previous reflections that can be pertinent for future task attempts
while thinking of possible new strategies that can lead to a successful completion of
this task.
```

---

**Reflexion Full Prompt**

```
{system prompt}
{trajectory prompt}
## Task Status: {verifier evaluation of the task attempt (e.g., FAILURE)}
## Previous Reflections: {reflections from prior executions of the task}
```

```
## Now please provide your response as follows:
POSSIBLE REASONS FOR {status}: [What are the possible reasons why the objective was {
achieved|not achieved}?] <<Note: we observe agreement bias can lead to blind agreement
with prior executions and affect reflexion generation. Similar to SGV, this section
helps mitigate this issue.>>
REFLECTION: [Reflections for future task attempts]
```

## B.7 ROBOMIMIC PROMPTS

---

**robomimic SGV Prompt - First Step**

```
You are highly skilled in robotic tasks.
Based on the task description and the image of the initial state, predict how the task
must look like at the current time stamp.

Task Description:
Pick up L-shaped pencil and insert L-shaped pencil into the hole
Pick up tool and hang tool on L-shaped pencil

Current time stamp: {n/N}
{image}
```

---

**robomimic SGV Prompt - Second Step**

```
Compare the prediction with the actual state of the system in the current time step
depicted in the image.

If the current state shows a situation that depicts failure, label it as failure.

If both tasks being completed or nearing completion was predicted and both tasks have
been completed, label it as success.

If the prediction is that the task will be in progress and the task is in progress,
label it as in progress.

{first step generation}
Current time stamp: {n/N}
{image}
```

---

**robomimic - Verifier Prompt, no SGV**

```
You are highly skilled in robotic task verification.

Based on the task description, the current time step and the the actual state of the
system in the current time step (depicted in the image), label it as 'success', '
failure' or 'in progress'.

Given the time step, if the current state is not as per expectation (with dropped tools
), label it as failure.

If the time step is towards completion, and the task has been completed, label it as
success.

Given the time step, if the task is expected to be in progress and the task is in
progress, label it as in progress.

Current time stamp: {n/N}
{image}
```

## B.8    ABLATION PROMPTS - SGV

As we show in Sec. B.8, similar performance for SGV can be achieved by re-utilizing the verifier prompts from Sec. B.2, as long as prior generation and verification are divided into two phases. Below is the prompt used in the first step. The second step prompts are identical to those in Sec. B.2.

---

**SGV Unified - Full First Step Prompt**

```
{verifier system prompt} <<same as in Appendix B.2>>

## OBJECTIVE: {task objective text}

## IMAGES:
Image 0: {task objective image 0}
...
Image n: {task objective image n}
Image n+1: initial {webpage | computer} screenshot: {screenshot}

Please first provide the following
[Description of how tasks such as this are typically accomplished on {the web |
computers}].
```

---

The following prompts are used when prior generation and verification are performed together in a single, monolithic step. The system prompts are the same as in Sec. B.2, except for the rules, which are replaced by the following.

---

**Monolithic Retrieve and Verify - System Prompt Rules**

```
## To be successful, it is very important to follow the following rules:
1. You must not not assume the assistant's work is correct or incorrect beforehand. You
 should come up with your own opinion based on the information provided.
2. You must connect the dots between the objective and the information provided.
3. Your evaluation should be based on task completion; don't worry about efficiency for
 now.
4. Give utmost importance to the visual information provided, especially the
screenshots in the execution trace.
5. You should come up with a detailed description of how tasks like this are typically
accomplished on {the web | computers}.
6. Use this General {Web Knowledge | Computer Knowledge} as a guide, but also consider
the context of the specific task given to you.
```

---

**Monolithic Retrieve and Verify - Full Prompt**

```
{system prompt}
{trajectory prompt}

Now please provide your response.

## Here is the evaluation criteria:
SUCCESS: The assistant executed **all of** what's necessary to complete the objective.
The task is fully accomplished.
PARTIAL SUCCESS: The assistant executed **most of** what's necessary to complete the
objective. The task is partially accomplished.
FAILURE: The assistant executed **mostly incorrect** steps. The task is not
accomplished, and major revisions are needed.

## Provide your response as follows:
GENERAL {WEB | COMPUTER} KNOWLEDGE: <Description of how tasks such as this are
typically accomplished on {the web | computer}>
REASONING: <Comprehensive step-by-step reasoning to come up with your evaluation and
feedback> <<Only included for CoT variants>>
EVALUATION: <Your evaluation following the evaluation criteria>
FEEDBACK: <Feedback so the assistant can progress towards the objective>
```

---

## B.9 ABLATION PROMPTS - EVALUATION CRITERIA

The following list some examples of scoring templates used in our experiments for ablations in Sec. E.2. For all the 28+ templates, please refer to our codebase. The remainder of the prompts is the same as in Sec. B.2, except for the evaluation criteria section, which is replaced by the corresponding templates below.

---

**Scoring Template Examples**

```
<<Likert Ternary Scale (baseline)>>
SUCCESS: The assistant executed **all of** what's necessary to complete the objective.
The task is fully accomplished.
PARTIAL SUCCESS: The assistant executed **most of** what's necessary to complete the
objective. The task is partially accomplished. <<removed if using a binary scale>>
FAILURE: The assistant executed **mostly incorrect** steps. The task is not
accomplished, and major revisions are needed.

<<Score-based Scale, four-way>>
Provide a single score, rating the assistant's work strictly on a scale of 1 to 10
based on the confidence of its correctness for the task.

Score 1 to 2: You are very confident that the assistant's work is completely incorrect
for the task.
Score 3 to 4: You are fairly confident that the assistant's work is incorrect, but
there is a small chance it could be partially correct.
Score 5 to 6: You are uncertain about the correctness of the assistant's work; it could
 be right or wrong.
Score 7 to 8: You are fairly confident that the assistant's work is correct, but there
is a small chance it could be partially incorrect.
Score 9 to 10: You are very confident that the assistant's work is completely correct
for the task.

<<Score-based Scale, three-way>>
Provide a single score, rating the assistant's work strictly on a scale of 1 to 10
based on the confidence of its correctness for the task.

Score 1 to 4: You are confident that the assistant's work is incorrect, but there is a
small chance it could be partially correct.
Score 5 to 6: You are uncertain about the correctness of the assistant's work; it could
 be right or wrong.
Score 7 to 10: You are confident that the assistant's work is completely correct for
the task.

<<Range-based Scale, implicit>>
Provide a single score, rating the assistant's work strictly on a scale of 1 to 10
based on the confidence of its correctness for the task.
The score can be any scalar between 1 and 10, where:
- A **low score** means you are very confident that the assistant's work is completely
incorrect for the task.
- A **high score** means you are very confident that the assistant's work is completely
 correct for the task.
- A **mid-range score** means you are uncertain about the correctness of the assistant'
s work; it could be right or wrong.

<<Range-based Scale, explicit>>
Provide a single score, rating the assistant's work strictly on a scale of 1 to 10
based on the confidence of its correctness for the task.
The score can be any scalar between 1 and 10, where:
- A score close to 1 means you are very confident that the assistant's work is
completely incorrect for the task.
- A score close to 10 means you are very confident that the assistant's work is
completely correct for the task.
- A score close to 5 means you are uncertain about the correctness of the assistant's
work; it could be right or wrong.

<<True/False Statements>>
The following are statements about the assistant's work. If a statement is correct,
please reply with T, otherwise reply with F.

A. The assistant executed **all of** what's necessary to complete the objective. The
task is fully accomplished.
B. The assistant executed **most of** what's necessary to complete the objective. The
task is partially accomplished.
C. The assistant executed **part of** what's necessary to complete the objective. The
task is partially accomplished.
```

---

```
D. The assistant executed **mostly incorrect** steps. The task is not accomplished, and
 major revisions are needed.
E. It is not clear if the assistant executed what's necessary to complete the objective
. The task can be fully accomplished, partially accomplished, or not accomplished.

Provide your response as a list of letters for each statement, as follows:
T or F, T or F, T or F, T or F, T or F

<<Option-based, roman, five-way>>
Please evaluate the assistant's work by selecting the option that best represents your
level of confidence.
What best describes your confidence in the assistant's work?
I. You are very confident that the assistant's work is completely incorrect for the
task.
II. You are fairly confident that the assistant's work is incorrect, but there is a
small chance it could be partially correct.
III. You are uncertain about the correctness of the assistant's work; it could be right
 or wrong.
IV. You are fairly confident that the assistant's work is correct, but there is a small
 chance it could be partially incorrect.
V. You are very confident that the assistant's work is completely correct for the task.
```

## C  QUALITATIVE ANALYSIS

### C.1  AGREEMENT BIAS

Fig. 7 illustrates the trajectory for a failed task in VisualWebArena, along with evaluations produced by three verifier variants. Two key flaws are evident in the trajectory: (i) the agent does not perform any price comparisons, and (ii) it stops after adding an item to the cart, without proceeding to checkout. The CoT verifiers validate the execution, providing reasoning traces that justify it as correct despite the evident shortcomings. In contrast, the SGV verifier flags the absence of price comparison, and accurately declares the execution as unsuccessful.

### C.2  ONLINE VERIFICATION AND FEEDBACK

As discussed in section 5, our method enables an MLLM verifier to monitor agent behavior and provide natural language feedback in real time, guiding the agent toward successful task completion. We show there that this mechanism has a positive overall impact on performance, resulting in a 5 percentage point increase in success rate on VisualWebArena (approximately 10% relative improvement) and a 2 percentage point gain in OSWorld (approximately 9% relative), compared to a no-verifier baseline.

The success of such a mechanism relies on three factors: (i) the accuracy of the verifier's judgment, (ii) the quality of its feedback, and (iii) the ability of the agent to interpret and act upon that feedback, including the capacity to revise or backtrack when necessary.

In this section, we explore these factors through representative cases that offer insight into both the strengths and failure modes of our method — and of MLLM verifiers more broadly. The cases are as follows:

- **Accurate verification with helpful feedback:** The verifier correctly identifies flaws and provides actionable feedback that improves task execution.

- **Accurate verification with helpful feedback, limited by agent ability:** The verifier offers correct judgments and useful feedback, but the agent fails to act on it effectively.

- **Strict verification, harmless:** False negatives from the verifier that do not affect the agent's final outcome and often elicit more robust behavior.

- **Strict verification, harmful:** False negatives that lead to degraded task performance or failed executions.

- **Overly lenient verification:** False positives where the verifier fails to detect clear mistakes, and contributes no useful signal. This stems from agreement bias and is the most prominent failure case for conventional MLLM verifiers.

**Accurate Verification with Helpful Feedback** Fig. 13 shows an example where the verifier successfully steers the ReAct agent to a correct execution on a challenging VisualWebArena task. The agent performs a sequence of reasonable steps and finds a product closely aligned with the user's request, but not the cheapest one. The verifier detects this oversight and provides feedback that prompts the agent to backtrack, sort the results correctly, and select the appropriate item.

This is a particularly challenging case that is highly susceptible to agreement bias, as the initial trajectory comprises reasonable steps and ends with a product that closely aligns with the user's query.

**Accurate Verification with Helpful Feedback, Limited by Agent Ability** Fig. 14 presents a failure case where, despite the verifier correctly identifying the issue and providing appropriate feedback, the agent is unable to recover. The verifier advises the agent to calculate shipping costs before leaving a review. However, the agent instead performs a sequence of poor actions, from which it never recovers. This example demonstrates a central limitation of our agent: the performance is ultimately constrained by the capabilities of the generator, regardless of the quality of the verifier.

**Overly Strict Verification, Harmless** Fig. 15 illustrates this dynamic in a borderline case. The agent clicks on the first visible option without searching. Although this behavior is marked as correct by automated scripts, the verifier rejects it, requiring the agent to explicitly search for the product. After performing the search, the agent provides the same (correct) response, but is again rejected for not clicking on the item to inspect its details. In this task, that step is arguably unnecessary, as the required information is already visible. Still, the agent proceeds to click and confirm the details, ultimately converging on the same answer a third time.

This example reveals two insights: (i) Even though the verifier is demanding, its feedback aligns well with the user's original intent, and does not prevent task completion. (ii) In fact, the strict verifier elicits robust behavior from the agent. The benchmark task is relatively simple and susceptible to shortcuts, but the verifier enforces a robust and generalizable strategy.

**Overly Strict Verification, Harmful** Fig. 16 presents an example where a correct initial execution is degraded into a failure. The user's query accepts either "Ohio" or "Pennsylvania" as valid answers, but the verifier demands the cheapest option across both. The agent, however, is unable to navigate back to Ohio to perform a valid comparison and runs out of steps. This example also illustrates the difficulty of interpreting real-world user intents, which are often vague and under-specified.

**Overly Lenient Verification** Our verifier remains susceptible to false positives, primarily due to limitations in the underlying MLLM's vision-language perception and reasoning capabilities.

Fig. 11 shows a failure on a relatively simple task in VisualWebArena that exposes this issue: both the agent and the verifier fail to count the number of elements in an image correctly, resulting in a false positive. This is a known weakness of current MLLMs Rahmanzadehgervi et al. (2025) that becomes even more problematic in our setting, where perception errors can compound across multi-step trajectories.

A more complex failure is illustrated in Fig. 12, which juxtaposes the verifier's strengths and weaknesses. The verifier successfully detects that the agent has terminated the task prematurely, failing to submit a required comment. However, it misses two critical issues: (i) that the agent has incorrectly counted the number of red keys in the image, and (ii) that it has posted the comment in the wrong location. Accurately verifying these mistakes would require the detection of fine-grained visual cues, knowing which small button the agent clicked on, for example.

As perceptual abilities continue to improve in future MLLMs, we anticipate a corresponding increase in the effectiveness of SGV, as its benefits are largely orthogonal to these capabilities. Alternatively, an interesting direction for future work is to pursue modular approaches that combine models with complementary, orthogonal strengths–for example, integrating specialist models for fine-grained perception tasks such as object counting or UI element recognition, thereby improving the verification accuracy in visually demanding scenarios.

# D MAIN RESULTS - DETAILED BREAKDOWNS

## D.1 OFFLINE EVALUATION OF AGENT TRAJECTORIES - BREAKDOWNS PER ENVIRONMENT

Table 6: Performance of MLLM verifiers on VisualWebArena Trajectories.

| Model | (a) No SGV | | | | | (b) SGV | | | | | (b) - (a) | | |
|---|---|---|---|---|---|---|---|---|---|---|---|---|---|
| | Acc. | TPR | TNR | Bias | dSkew | Acc. | TPR | TNR | Bias | dSkew | Acc. | Bias | dSkew |
| Gemini 2.0 | 56 | 93 | 28 | 38 | 35 | 65 | 90 | 42 | 28 | 23 | +9 | -10 | -12 |
| Gemini 2.5 Lite | 56 | 91 | 26 | 36 | 32 | 62 | 84 | 42 | 28 | 21 | +5 | -8 | -10 |
| Gemini 2.5 | 65 | 90 | 47 | 26 | 21 | 76 | 84 | 71 | 11 | 5 | +11 | -15 | -16 |
| Gemini-2.5 (T) | 70 | 90 | 55 | 23 | 18 | 77 | 86 | 70 | 12 | 6 | +7 | -11 | -12 |
| GPT-4.1 Mini | 59 | 94 | 31 | 35 | 31 | 72 | 84 | 62 | 11 | 4 | +13 | -24 | -27 |
| GPT-4.1 | 67 | 88 | 52 | 22 | 17 | 77 | 86 | 71 | 12 | 6 | +10 | -11 | -11 |
| GPT-o1 (T) | 70 | 80 | 62 | 22 | 16 | 78 | 83 | 73 | 13 | 7 | +7 | -9 | -8 |
| o4-Mini | 73 | 89 | 59 | 17 | 11 | 79 | 86 | 73 | 8 | 3 | +8 | -9 | -8 |
| GPT-5 Nano | 67 | 78 | 58 | 13 | 6 | 70 | 76 | 66 | 6 | 2 | +4 | -7 | -4 |
| GPT-5 (T) | 77 | 83 | 71 | 8 | 3 | 81 | 81 | 81 | 1 | 0 | +5 | -7 | -3 |
| Qwen3-32b | 67 | 88 | 49 | 22 | 16 | 74 | 82 | 68 | 9 | 3 | +7 | -13 | -13 |
| Qwen3-235b-a22b | 65 | 86 | 47 | 23 | 16 | 73 | 80 | 68 | 8 | 3 | +8 | -14 | -13 |
| Qwen3-235b-a22b (T) | 66 | 85 | 50 | 20 | 13 | 75 | 83 | 69 | 9 | 3 | +9 | -11 | -10 |
| Llama-4-Maverick-17B-128E | 62 | 85 | 42 | 24 | 17 | 67 | 80 | 56 | 16 | 9 | +5 | -9 | -9 |

*(T) = Thinking enabled with the maximum thinking budget for the corresponding model.

Table 7: Performance of MLLM verifiers on evaluation of OSWorld trajectories.

| Model | (a) No SGV | | | | | (b) SGV | | | | | (b) - (a) | | |
|---|---|---|---|---|---|---|---|---|---|---|---|---|---|
| | Acc. | TPR | TNR | Bias | dSkew | Acc. | TPR | TNR | Bias | dSkew | Acc. | Bias | dSkew |
| Gemini 2.0 | 66 | 99 | 56 | 33 | 33 | 74 | 97 | 67 | 25 | 24 | +8 | -9 | -9 |
| Gemini 2.5 Lite | 55 | 100 | 42 | 45 | 45 | 68 | 96 | 60 | 31 | 29 | +13 | -15 | -16 |
| Gemini 2.5 | 71 | 97 | 63 | 28 | 27 | 83 | 92 | 81 | 13 | 11 | +13 | -15 | -17 |
| Gemini 2.5 (T) | 78 | 95 | 73 | 20 | 18 | 87 | 91 | 86 | 8 | 5 | +9 | -12 | -14 |
| GPT-4.1 Mini | 60 | 99 | 49 | 40 | 39 | 75 | 99 | 69 | 24 | 24 | +16 | -16 | -16 |
| GPT-4.1 | 80 | 93 | 76 | 16 | 14 | 85 | 88 | 84 | 9 | 6 | +5 | -7 | -8 |
| GPT-o4-Mini | 84 | 87 | 83 | 6 | 3 | 90 | 85 | 91 | 4 | 2 | +6 | -2 | -1 |
| GPT-5-Nano (T) | 76 | 90 | 72 | 19 | 16 | 81 | 90 | 80 | 14 | 10 | +5 | -5 | -6 |
| GPT-5 (T) | 87 | 90 | 86 | 9 | 6 | 92 | 90 | 92 | 4 | 2 | +6 | -5 | -4 |
| Qwen3-32b | 70 | 95 | 64 | 27 | 26 | 78 | 99 | 74 | 20 | 18 | +8 | -7 | -8 |
| Qwen3-235b-a22b | 66 | 100 | 56 | 34 | 34 | 78 | 99 | 73 | 21 | 21 | +13 | -13 | -14 |
| Qwen3-235b-a22b (T) | 66 | 100 | 56 | 34 | 34 | 78 | 99 | 73 | 21 | 21 | +13 | -13 | -14 |
| Llama-4-Maverick-17B-128E | 58 | 100 | 46 | 41 | 41 | 63 | 99 | 53 | 35 | 35 | +5 | -6 | -6 |

*(T) = Thinking enabled with the maximum thinking budget for the corresponding model.

## D.2 ONLINE SUPERVISION - BREAKDOWNS FOR OSWORLD

Tab. 8 shows the performance of the base agent with and without verifier intervention across each of the domains included in OSWorld.

| | Base Agent | + SGV, outcome verifier | + SGV, verify every 5 steps |
|---|---|---|---|
| All | 22 | 25 | 27 |
| Chrome | 29 | 30 | 35 |
| GIMP | 24 | 28 | 31 |
| LO Calc | 15 | 15 | 16 |
| LO Impress | 25 | 35 | 29 |
| LO Writer | 30 | 44 | 41 |
| Multi Apps | 8 | 9 | 13 |
| OS | 36 | 39 | 42 |
| Thunderbird | 33 | 36 | 47 |
| VLC | 8 | 8 | 8 |
| VS Code | 57 | 62 | 57 |

Table 8: UI-Tars-1.5-7b performance across OSWorld domains with and without SGV supervision. All numbers are averages across three runs. More frequent interventions help prevent the agent from derailing into hard-to-recover states, thereby delivering superior performance.

### D.3 ONLINE SUPERVISION - PERFORMANCE BREAKDOWN PER CONFUSION METRICS

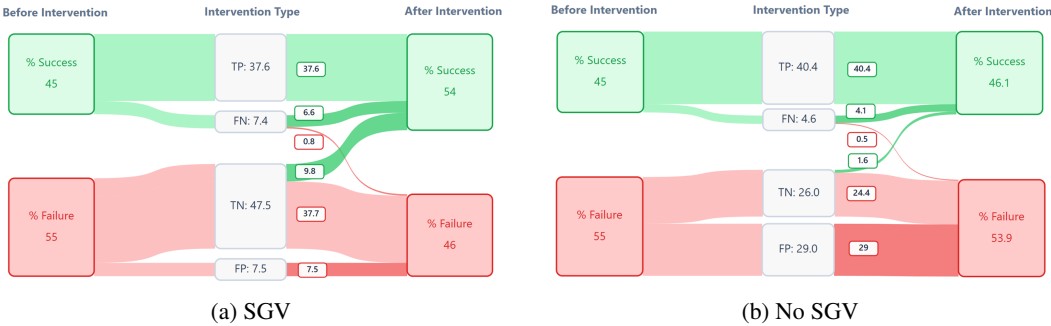

(a) SGV              (b) No SGV

Figure 3: Online performance transitions after verifier interventions. The boxes on the left and right show the base agent's performance before and after the MLLM Verifier intervention. The boxes in the middle categorize the interventions as false/true positives, false/true negatives. Numbers indicate % tasks in the benchmark.

Fig. 3 presents a breakdown of the performance gains discussed in Sec. 5 with and without SGV. The flow (Failure→TN→Success) shows the percentage of tasks that were initially unsuccessful, were correctly identified as such by the verifier, and subsequently became successful due to the verifier's feedback. The flow (0→FP→0) shows the percentage of tasks that were initially unsuccessful, were incorrectly identified as successful by the verifier, and therefore remained unsuccessful. The flow (1→FN→1) indicates the percentage of tasks that were initially successful, were classified as a failure by the verifier, but remained successful after its feedback.

As discussed in Sec. 5, most of the SGV interventions are non-disruptive: (1→FN→1) shows that the majority of "FN" evaluations in SGV remain successful, while 14% of the tasks that were initially unsuccessful become successful after corrective feedback from the verifier (0→TN→1). In contrast, in the No-SGV case, 35% of the unsuccessful tasks remain unsuccessful due to the lack of a corrective signal (0→FP→0). See Sec. C.2 for examples, discussion, and a categorization of these patterns.

# E  ABLATIONS AND ADDITIONAL RESULTS

## E.1  EXTERNAL VALIDATION: AGENTREWARDBENCH RESULTS

Tab. 9 shows the performance of the updated oracle evaluators in AgentRewardBench Men et al. (2025). The original oracles in VisualWebArena exhibit relatively low alignment with human annotations. After applying our fixes, the oracles reach near-perfect agreement with humans.

Tab. 10 shows that the **baseline MLLM Verifier** used in our analysis of agreement bias is relatively strong, **setting a state-of-the-art on AgentRewardBench**, while **SGV further improves** upon this, surpassing even the original oracles included in VisualWebArena.

Taken together, these results help validate our experimental design. First, they show that our observations on agreement bias are not artifacts of weak evaluators or flawed implementations. Second, the strong performance of our baseline implementation reinforces the motivations discussed in Sec. 3.2 for introducing stronger agents and environment corrections. Specifically, the high performance in AgentRewardBench is partly because its trajectories are generated by weaker agents and include bugs in the BrowserGym suite de Chezelles et al. (2025). This results in data points where agents fail due to looping indefinitely or encountering errors (e.g., "Page not found" or non-working `click` actions on `select` elements, see Sec. F.2), which are comparatively easy for MLLM verifiers to detect.

Table 9: Alignment of the original and revised VisualWebArena oracles with human annotations from AgentRewardBench. Our revised oracles reach near-perfect agreement with human labels.

|           | Original | Improved Oracle |
|-----------|----------|-----------------|
| Precision | 85.2     | 100             |
| TPR       | 58.2     | 92              |
| TNR       | 95.9     | 100             |
| Acc       | 85.1     | 98              |

Table 10: Comparison of our verifier to the best performing judges in AgentRewardBench for the VisualWebArena (VWA) environment. Our *baseline* MLLM Verifiers already achieve state-of-the-art performance. SGV further improves and surpass even the original VWA oracles.

| Method                                    | Precision |
|-------------------------------------------|-----------|
| Rule-based (VWA Oracle, original)         | 85        |
| **Rule-based (VWA Oracle, ours)**         | **100**   |
| No-SGV Baseline (Gemini 2.5, no Thinking) | 73        |
| WebJudge (GPT-o4, SOTA)                    | 75        |
| SGV (Gemini 2.5, no Thinking)             | 80        |
| **No-SGV Baseline (GPT-o4)**              | **80**    |
| **SGV (GPT-o4)**                          | **86**    |

Note: AgentRewardBench's primary leaderboard metric is precision (P), which is directly proportional to metrics used in our analysis: $P = \frac{TPR \cdot s}{TPR \cdot s + (1\text{-}TNR) \cdot (1-s)}$, where $s$ = agent success rate measured by humans.

## E.2  DISTRIBUTION OF MLLM RESPONSES ACROSS EVALUATION TEMPLATES

In Fig. 4, we show the distribution of MLLM Verifier responses across 28 scoring templates covering both Likert, score-based, and other evaluation criteria (see Sec. B.9 for examples and our codebase for all variations). We also include interventions aimed at mitigating biases in LLM evaluators, such as rephrasing, criteria order shuffling and reversal, and syntactic structure changes Chen et al. (2025a); Ye et al. (2024). The following summarizes the main findings.

(1) MLLMs tend to concentrate their evaluations at the high end of the scale, largely independent of interventions and how the evaluation is framed.

(2) MLLMs rarely express uncertainty, even on ambiguous cases, when the evaluation template explicitly includes an "uncertain" option, and reserves the highest score for high-confidence success.

(3) Likert scales tend to produce slightly more balanced distributions compared to numeric-based scales.

(4) Binary scales can exacerbate bias toward favorable evaluations. Including a third option to allow the model to "offload" evaluations into "partial success" helps mitigate this effect. Three-way scales seems sufficient to achieve this effect, with no meaningful differences observed for more granular scales.

(5) SGV leads to a more balanced distribution of scores for all templates.

(6) These observations align with discussions in Sec. 3.2, suggesting root causes stemming from inherent limitations of RLHF that conflates human rater satisfaction with truthfulness Sharma et al. (2025) and knowledge extraction bottlenecks depending on how the information is presented Allen-Zhu and Li (2024a;b), with SGV mitigating these issues effectively.

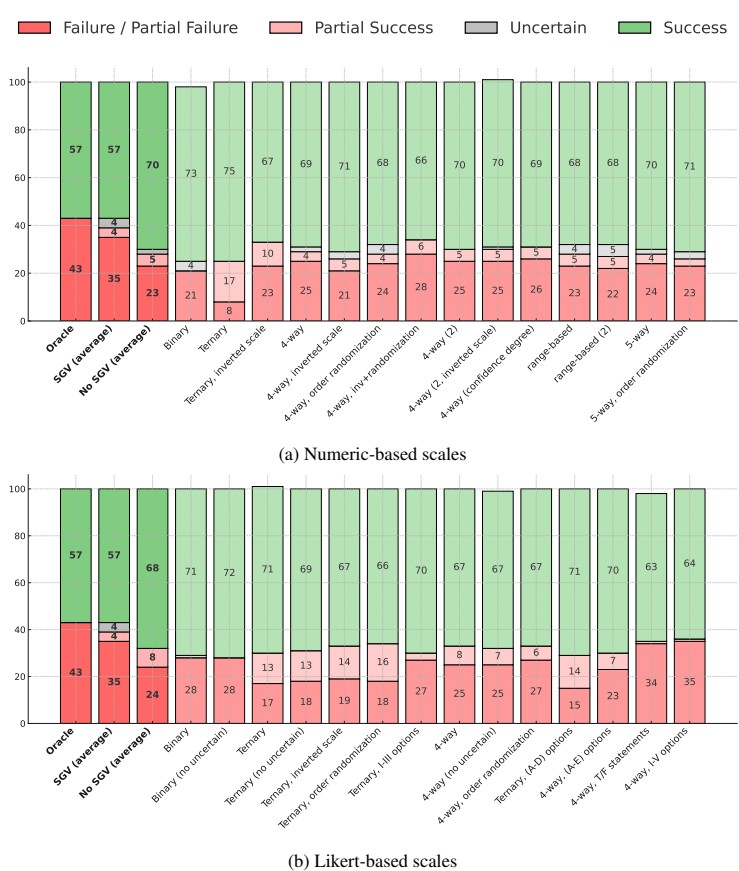

(a) Numeric-based scales

(b) Likert-based scales

Figure 4: Distribution of MLLM verifier responses across prompt/scoring templates and interventions to mitigate biases in LLM judgments.

### E.3 CALIBRATION OF MLLM VERIFIER DISTRIBUTIONS: LIMITATIONS AND OPPORTUNITIES

This section extends the observations in the main text regarding agreement bias and the tendency of MLLMs to assign disproportionately high confidence scores. Specifically, we examine whether post-processing calibration techniques can address the skewed distributions exhibited by MLLM verifiersSec. E.2, and provide a proof-of-concept experiment on the potentials for sampling-based techniques that take into account the skew created by agreement bias.

### E.3.1 Platt Calibration of MLLM-Verifier Outputs

We first evaluate Platt and Isotonic calibration of confidence scores generated by MLLMs under «Range-based Scale, implicit» template illustrated in Sec. B.9.[1] We focus on VisualWebArena, where the distribution of agent successes and failures is relatively balanced (47% success, 53% failure). We use our representative subset(Sec. F.1) for testing and train the calibrator on the remaining trajectories ( 605 examples). The threshold separating success from failure is optimized using Youden's J statistic on the training set, and we additionally report the Area Under the Curve (AUC) statistic for completeness.

Tab. 11 reports evaluation metrics before and after calibration, compared to SGV under the same template. In summary:

- We do not observe meaningful gains from applying Platt scaling to the generated confidences.
- This is explained by the skew in MLLM score distributions discussed in Sec. E.2, and shown in Tab. 12: MLLMs overwhelmingly assign labels at the upper end of the scale, leaving Platt scaling with no granularity to exploit and causing large clusters of identical scores to map to the same calibrated probability. The same issue arises with other methods, such as an isotonic transform.
- In line with Sec. E.2 results, SGV leads to a substantially more balanced distribution of scores, with corresponding gains in evaluation metrics.
- We highlight these findings seems not an artifact of prompt design/template: as SGV results show, it is possible to obtain more balanced distributions under this evaluation design.

Table 11: Calibration of MLLM verifier confidence scores

| Metric | Before Calibration | Platt | SGV |
|--------|-------------------|-------|-----|
| ACC | 63 | 65 | 72 |
| TPR | 92 | 92 | 82 |
| TNR | 37 | 40 | 64 |
| Bias | 29 | 28 | 10 |
| dSkew | 25 | 23 | 4 |
| AUC | 65 | 65 | 74 |

Table 12: Platt scaling vs. SGV – distribution of confidence scores.

| Score bucket | Before Calib. | Platt | Isotonic | SGV |
|--------------|---------------|-------|----------|-----|
| [0, 0.1) | 10% | 11% | 16% | 6% |
| [0.1, 0.2) | 2% | 10% | 0% | 8% |
| [0.2, 0.3) | 5% | 3% | 4% | 13% |
| [0.3, 0.4) | 4% | 1% | 4% | 5% |
| [0.4, 0.5) | 2% | 1% | 1% | 1% |
| [0.5, 0.6) | 2% | 74% | 74% | 3% |
| [0.6, 0.7) | 1% | 0% | 0% | 4% |
| [0.7, 0.8) | 1% | 0% | 0% | 4% |
| [0.8, 0.9) | 0% | 0% | 0% | 3% |
| [0.9, 1.0] | 74% | 0% | 0% | 55% |

### E.3.2 Proof-of-Concept: Calibration of Sampling-Derived Probabilities

We further explore whether calibration is more effective when applied to probabilities estimated via sampling. Specifically, for each of the 910 trajectories in VisualWebArena, we sample 8 completions

---

[1]We observe that letting the MLLM freely generate a confidence provides inferior results compared to providing some grounding on the confidence scale, as in the utilized template. Therefore, for confidence generation we adopted the latter variation, mapping 1-10 scores to 0-1 during calibration.

and estimate empirical probabilities by measuring the proportion of times the MLLM labels a trajectory as a success or failure under the best-performing template across the 28 tested—the ternary Likert scale used in our baselines. To test the Platt calibrator, we utilize our 1/3-sized representative subset that provides the same distribution as the full benchmark, and we train it on the remainder of the data (605 datapoints). In both cases, the data is roughly balanced, with 47% success and 53% failed trajectories.

Before discussing results, we highlight this experiment is a *proof-of-concept* under relatively ideal and computationally expensive conditions. Particularly:

- The calibrator is trained and tested on datasets with roughly equal numbers of success and failures.
- It assumes access to the high-quality labels provided by our improved VisualWebArena oracles ( Sec. 3.3)—a limitation in real-world settings that motivates the need for MLLM verifiers in the first place.
- Training and test domains match; i.e., we do not consider out-of-distribution generalization.
- To estimate probabilities, we sample a relatively large number of completions (8 per trajectory, totaling over 7,200 generations).
- The thresholds used to separate classes—which, as shown below, can be a sensible choice—are optimized under these same ideal conditions.
- At test time, we still generate 8 completions per new trajectory for probability estimation, and apply the calibrator to them.

Results are shown in table Tab. 13. In summary:

- Vanilla majority voting fails to deliver improvements for not taking into account the skew in MLLM label distributions toward positive evaluations.
- Leveraging this fact to apply a calibration to MLLM implicit probabilities leads to more balanced distributions and improvements in evaluation metrics.
- Most notably, SGV lead to distributions superior than the calibrated ones obtained in ideal scenarios through a single generation. Importantly, this arises organically by leveraging the model's own sampling mechanisms, without the need for ground-truth labels, additional training, or calibration steps. Moreover, these gains naturally extend to other domains, as demonstrated by SGV's improvements across benchmarks.
- SGV also avoids complicated hyperparameter tuning such as calibration thresholds. As shown in columns 'p15' and '0.5', setting this threshold to either the 15th percentile (bootstrap-estimated) or naively to 0.5 can eliminate the gains from Platt scaling.

Taken together, these results point to some important implications. First, they reinforce that SGV's effectiveness stems from mitigating the intrinsic bias present in MLLM response distributions. Secondly, they suggest a potential for both training-time and test-time approaches that explicitly account for the asymmetries introduced by agreement bias—e.g., loss functions that penalize such imbalance, or adjustments to sampling mechanisms for methods that rely on sampling (e.g., GRPO). Since SGV itself induces more balanced output distributions, incorporating SGV into larger pipelines may also be fruitful—e.g., as a replacement for oracle labels in methods like the one above.

We hope that our identification of agreement bias, our demonstration of the risks it poses for applications relying on MLLM verifiers, our evidence on the benefits of mitigating this bias to downstream applications, and the insights provided help motivate further research in this direction.

Table 13: Platt scaling vs. SGV using sampling-derived probabilities under a ternary Likert template.

| Metric | $\text{NoSGV}_{temp=0}$ | $\text{NoSGV}_{\textbf{maj}}$ | **Platt** | $\textbf{Platt}_{p15}$ | $\textbf{Platt}_{0.5}$ | $\textbf{SGV}_{temp=0}$ | $\textbf{SGV}_{\textbf{maj}}$ | $\textbf{SGV}_{\textbf{Platt}}$ |
|---|---|---|---|---|---|---|---|---|
| ACC | 65 | 65 | **72** | 69 | 71 | **76** | 75 | 78 |
| TPR | 92 | 93 | **77** | 85 | 86 | **84** | 85 | 79 |
| TNR | 41 | 40 | **68** | 56 | 57 | **71** | 67 | 78 |
| Bias | 27 | 28 | **6** | 20 | 16 | **11** | 11 | 1 |
| dSkew | 23 | 24 | **2** | 15 | 9 | **5** | 5 | 0 |
| AUC | 67 | 66 | **73** | 75 | 75 | **76** | 76 | 83 |

### E.4 OUTCOME VS PROCESS-BASED VERIFICATION

For some domains, such as OSWorld, certain actions can be more destructive, pushing the agent into states that are difficult or impossible to recover from—for example, deleting a file. This naturally raises an important question: *when* should a verifier be invoked? While we leave a full exploration to future work, Tab. 14 provides initial evidence for the promise of this direction, particularly when equipped with verifiers that know when to intervene in the first place. We examine two configurations: one in which the MLLM verifier intervenes only after STOP actions (the "Outcome-based" setting), and another in which it evaluates trajectories periodically every 5 steps. As observed, the qualitative conclusions mirror those in the main text: a baseline MLLM verifier yields only modest improvements, whereas SGV consistently produces substantial gains in both configurations, with slightly better results when verification is more frequent. More generally, we view the study of process and outcome-based verification (Ren et al., 2023; Lightman et al., 2023; Shao et al., 2024b; Setlur et al., 2025; Swamy et al., 2025) in digital environments as an exciting direction for future work, particularly due to their often discrete state-space representation that makes process reward modeling more amenable.

Table 14: Agent Performance across OSWorld and VisualWebArena under periodic and outcome-based verification.

| Verification Mode | OSWorld | VisualWebArena |
|---|---|---|
| Baseline Agent | 21.7 | 45.0 |
| **No SGV** | | |
| + Outcome-based | 23.4 | 46.1 |
| + Every 5 Steps | 24.3 | 46.9 |
| **SGV** | | |
| + Outcome-based | 24.9 | 54.0 |
| + Every 5 Steps | 26.5 | 54.9 |

### E.5 IMPACT OF TRAJECTORY LENGTH ON VERIFIER PERFORMANCE

Some tasks in the digital world can require long trajectories, which raises the question of how verifier performance is affected. In Tables 15 and 16 we examine MLLM verifier performance stratified by trajectory length in OSWorld and in an extended 60-step configuration of VisualWebArena. To increase the number of samples per bucket and improve statistical stability, we average results across three comparable MLLMs (GPT-4.1, Qwen3-VL, and Gemini 2.5). Across both benchmarks, several consistent patterns emerge. First, SGV continues to reduce bias and skewness and improves TNR and accuracy for all trajectory lengths, reinforcing the robustness of the method. Second, verifier performance tends to **increase** with trajectory length. As reflected by the low success rates in longer trajectories, this pattern arises largely because longer trajectories typically indicate that agents have drifted further from the task objective, making failures easier for verifiers to detect. For example, UI-TARS repeatedly opening irrelevant applications, or performing destructive operations like file deletions in OSWorld.

This observation reinforces an important methodological point raised in Sec. 3.3: to evaluate verifiers meaningfully—and to surface limitations such as agreement bias—it is essential to use strong agent policies (like our VisualWebArena agents) that produce high-quality trajectories.

While our results show no degradation in performance at longer trajectory lengths, we acknowledge that the combination of strong agents *and* long trajectories may introduce new challenges, including amplified agreement bias or yet-unidentified failure modes. However, a full examination of these factors under the current state of benchmarks and agents is challenging: unless tasks inherently require long trajectories and agents can reliably produce them, disentangling these factors without introducing confounders is non-trivial. In Sec. E.6 we partially address this limitation, varying agent quality in VisualWebArena while keeping trajectory length fixed. Experiments reveal that (i) verifier performance degrades when the verifier is weaker than the agent, specifically due to an increase in agreement bias, and (ii) SGV consistently improves outcomes across all settings.

Table 15: MLLM verification performance by trajectory length in VisualWebArena.

| Metric | All | 1–5 | 6–19 | >20 |
|---|---|---|---|---|
| % Success | 47 | 57 | 36 | 17 |
| N Samples | 2730 | 1587 | 1218 | 198 |
| % Total. | 100 | 58 | 45 | 7 |
| **No-SGV** | | | | |
| ACC | 67 | 68 | 61 | 65 |
| TPR | 90 | 91 | 88 | 64 |
| TNR | 48 | 40 | 46 | 65 |
| Bias | 24 | 22 | 30 | 22 |
| dSkew | 19 | 17 | 26 | 16 |
| **SGV** | | | | |
| ACC | 74 | 73 | 72 | 82 |
| TPR | 82 | 84 | 81 | 66 |
| TNR | 67 | 60 | 67 | 85 |
| Bias | 10 | 9 | 14 | 6 |
| dSkew | 5 | 4 | 8 | 2 |
| **Δ (SGV – No-SGV)** | | | | |
| ACC | +7 | +5 | +11 | +17 |
| Bias | -14 | -13 | -16 | -16 |
| dSkew | -14 | -13 | -18 | -14 |

Table 16: MLLM verification performance by trajectory length in OSWorld.

| Metric | All | 1–10 | 11–20 | 21–30 | >30 |
|---|---|---|---|---|---|
| % Success | 22 | 54 | 20 | 16 | 2 |
| N Samples | 1044 | 300 | 198 | 111 | 435 |
| % Total. | 100 | 29 | 19 | 11 | 42 |
| **No-SGV** | | | | | |
| ACC | 76 | 71 | 59 | 67 | 89 |
| TPR | 96 | 97 | 97 | 89 | 67 |
| TNR | 70 | 41 | 49 | 63 | 89 |
| Bias | 22 | 26 | 40 | 29 | 10 |
| dSkew | 21 | 24 | 39 | 27 | 9 |
| **SGV** | | | | | |
| ACC | 83 | 77 | 75 | 81 | 94 |
| TPR | 91 | 90 | 97 | 88 | 67 |
| TNR | 81 | 61 | 70 | 80 | 94 |
| Bias | 12 | 13 | 24 | 14 | 5 |
| dSkew | 10 | 8 | 23 | 14 | 5 |
| **Δ (SGV – No-SGV)** | | | | | |
| ACC | +7 | +6 | +16 | +14 | +6 |
| Bias | -10 | -13 | -13 | -15 | -5 |
| dSkew | -11 | -16 | -13 | -13 | -4 |

### E.6 CROSS-MODEL VARIATIONS AND VERIFICATION DIFFICULTY

In Sec. 4 we discuss how verification is affected by verification difficulty when comparing performance of a weaker agent in OSWorld and VisualWebArena. The results in Tab. 17 complement those findings by exploring variations in model strength between the models used to build the agents (or "generator") and the verifiers while keeping the environment constant (VisualWebArena). In summary, results show that:

(i) Agreement bias occurs both when the verifier is stronger or weaker than the generator, and when agents and verifiers are built from models of distinct families and size.

(ii) Verification performance is worse when the verifier is weaker than the generator (e.g., Gemini 2.0 verifies Gemini 2.5-based agent), particularly **due to an increase in agreement bias** (note TPRs actually increase).

(iii) SGV remains effective in all configurations, providing substantial improvements over the no-SGV baseline.

Table 17: (No SGV, SGV) performance across verifier-generator configurations in VisualWebArena. G-2.5 and G-2.0 refer to Gemini-2.5-Flash and Gemini-2.0-Flash, respectively.

| **Verifier** | G-2.5 | G-2.5 | G-2.5 | G-2.0 | G-2.0 |
| **Agent** | G-2.5 | G-2.0 | GPT4o | G-2.5 | GPT4o |
| --- | --- | --- | --- | --- | --- |
| Acc | (65, 76) | (70, 78) | (68, 77) | (56, 65) | (55, 65) |
| TPR | (90, 84) | (84, 77) | (80, 76) | (93, 90) | (85, 74) |
| TNR | (47, 71) | (63, 79) | (63, 77) | (28, 42) | (42, 61) |
| Bias | (26, 11) | (20, 7) | (20, 8) | (38, 28) | (35, 19) |
| dSkew | (21, 5) | (14, 2) | (13, 3) | (35, 23) | (30, 12) |
| Agent Success Rate | 47 | 33 | 31 | 47 | 31 |
| $\Delta$ Acc | 11 | 8 | 8 | 9 | 9 |
| $\Delta$ Bias | -15 | -13 | -12 | -10 | -16 |
| $\Delta$ dSkew | -16 | -12 | -11 | -12 | -18 |

### E.7 ABLATIONS TO SGV EXTRACTION AND VERIFICATION STEPS

Tab. 18 presents ablations for results in Sec. 4 where we (1) start with a conventional MLLM verifier, (2) add a prior generation step, (3) decouple prior generation and verification steps without modifying any of the prompts, and (4) add a separate prompt for the prior generation step. In all cases, we include a CoT instruction. Prompt templates are provided in Sec. B.8.

Table 18: Ablation results for SGV's two-step mechanism.

| # | Method | Accuracy (%) | TPR (%) | TNR (%) | Bias (%) | dSkew (%) |
| --- | --- | --- | --- | --- | --- | --- |
| 1 | Base MLLM Verifier | 65 | 90 | 47 | 26 | 21 |
| 2 | + First Step | 66 | 90 | 49 | 25 | 20 |
| 3 | SGV, Unified | 74 | 85 | 67 | 14 | 8 |
| 4 | SGV | 76 | 84 | 71 | 11 | 5 |

The conventional MLLM-based verification yields suboptimal performance, with slight improvement when a prior generation step is added (rows 1–2). One reason for this behavior is that when steps are combined, models tend to generate priors that align to information in their context window—a manifestation of agreement bias—reducing their effectiveness in grounding the verification. Decoupling the steps improves performance (rows 3–4), with SGV achieving similar results with or without specialized prompts. This suggests that SGV's effectiveness does not stem from prompt specialization and that it can be integrated easily with minimal modifications to existing codebases.

### E.8 ADDITIONAL ABLATIONS: PRIOR DIVERSITY, GROUNDING TOOLS, AND QUALITY OF FIRST-STEP GENERATION

In this section, we examine the effect of prior diversity on SGV performance and analyze the robustness of SGV under various prior-generation strategies. Across both sets of experiments, we observe that increasing the diversity of priors can enhance verification performance, while repeated

sampling or increased temperature within a single model family yields diminishing returns. We further demonstrate that alternative grounding strategies, such as providing web-search tools, fail to meaningfully mitigate agreement bias, and that SGV remains effective when prior generation is noisy or produced by weaker models, highlighting the stability of the mechanism.

**Effect of Prior Diversity**. To study how the diversity of first-step priors affects verification performance, table Tab. 19 reports results for four configurations: (1) a single prior generated from the same verifier model at high temperature; (2) a single prior from a model of a different family; (3) a concatenation of three priors sampled from the same model; and (4) a concatenation of three priors produced by three distinct model families. For reference, we include the baseline SGV and No-SGV settings. In all cases, the verifier model is Gemini-2.5-Flash with 0 temperature.

In summary, results shows that:

- **Diverse priors improve performance.** Incorporating priors from multiple model families (Column 5) yields the strongest results, suggesting that epistemic diversity provides additional useful signal for verification.
- **Increasing temperature or sampling within a model offers limited benefit.** Columns 1 and 4 demonstrate that once the first-step prior has been generated, additional variability from the *same* model yields marginal gains, indicating that most of the benefit is captured in the initial SGV step.

Table 19: Ablations on prior diversity.

| Setting | No SGV Baseline | SGV Baseline | (1) | (2) | (3) | (4) | (5) |
|---|---|---|---|---|---|---|---|
| First Step | – | $G_{(t=0)}$ | $G_{(t=2)}$ | Q | GP | G (k=3) | G+Q+GP (k=3) |
| Verify | G | G | G | G | G | G | G |
| Acc | 65 | 76 | 77 | 76 | 76 | 78 | 79 |
| TPR | 90 | 84 | 84 | 84 | 84 | 87 | 82 |
| TNR | 47 | 71 | 71 | 70 | 70 | 70 | 76 |
| Bias | 26 | 11 | 11 | 11 | 11 | 12 | 6 |
| dSkew | 21 | 5 | 5 | 5 | 5 | 7 | 2 |

**Legend:** G = Gemini 2.5    Q = Qwen3-VL    GP = GPT-5    t = temperature

**Effects of Grounding Tools, Noisy Priors, and First-Step Model Strength**. To further investigate agreement bias, and the effectiveness and robustness of the SGV mechanism, Tab. 20 compare MLLM verification performance in the following settings:

(1): No first-step prior generation, but provide the verifier a Web Search tool to ground the verification (Gou et al.);

(2): SGV, where the prior generation is supported by a web search tool;

(3) SGV, where prior generation is produced by a weaker variant of the same model;

(4) SGV, where we inject into the generated priors a random noiseShao et al. (2025)

(5): No SGV for a thinking variant of Gemini, but provide the verifier a Web Search tool to ground the verification

(6): SGV with thinking enabled, where priors are generated with thinking disabled.

In summary, results indicate that:

- In contrast to SGV, providing grounding tools such as web search to the verifier alone does not yield meaningful mitigation of agreement bias;
- Extracting knowledge from the own model seems sufficient, with the addition of web search not providing meaningful improvements;
- Weaker and non-thinking models can produce priors that are sufficiently informative for effective verification for stronger or thinking models;

- The similar performance of SGV with and without Web Search, the near-identical performance of Gemini-Thinking when priors are generated with or without thinking (6 vs. SGV baseline), and the comparable performance when thinking is fully disabled versus fully enabled all suggest that SGV's first-step mechanism accounts for the vast majority of the gains, and largely exhausts them.
- SGV remains effective even when the first-step generation is noisy, indicating that the method is robust to some degree of noise in the first-step prior generation.

Table 20: Effects of Grounding Tools, Noisy Priors, Priors with weaker and stronger models.

| Setting | NoSGV Base | (1) | SGV Base | (2) | (3) | (4) | NoSGV Base (T) | (5) | SGV Base (T) | (6) |
|---|---|---|---|---|---|---|---|---|---|---|
| First Step | – | – | G | G + Web | G-Lite | G + Noise | – | G (T) + Web | G (T) | G |
| Verify | G | G + Web | G | G | G | G | G (T) | G (T) | G (T) | G (T) |
| Acc | 65 | 64 | 76 | 74 | 74 | 73 | 69 | 66 | 77 | 76 |
| TPR | 90 | 92 | 84 | 84 | 82 | 81 | 90 | 73 | 86 | 86 |
| TNR | 47 | 44 | 71 | 65 | 69 | 68 | 55 | 61 | 70 | 68 |
| Bias | 26 | 29 | 11 | 17 | 11 | 14 | 23 | 10 | 12 | 13 |
| dSkew | 21 | 26 | 5 | 10 | 5 | 8 | 18 | 4 | 6 | 7 |

**Legend:** Base = baseline; G = Gemini-2.5-Flash; G-Lite = Gemini-2.5-Flash-Lite (T) = Thinking enable with max thinking budget; Web = Web Search Tool provided.

### E.9 ONLINE SUPERVISION AND SGV WITH WEAKER MODELS

Tab. 21 shows the performance of online verification with and without SGV on the full VisualWebArena benchmark utilizing `gemini-2.0-flash`, a weaker model than the one used in Sec. 5. Consistent with findings in Sec. 5, the no-SGV verifier fails to improve performance and even results in degradation in certain domains. In contrast, SGV consistently improves results across all domains, achieving a 3 percentage point gain overall (9% relative improvement).

Table 21: Task success rates (%) on VisualWebArena with and without SGV using `gemini-2.0-flash` as the base MLLM.

| Method | All VWA | Shopping | Reddit | Classifieds |
|---|---|---|---|---|
| Base Agent | 36 | 38 | 27 | 41 |
| + Verifier, no SGV | 36 | 39 | 27 | 40 |
| + Verifier, SGV | 40 | 41 | 29 | 46 |

## F ADDITIONAL DETAILS AND RESULTS FOR VISUALWEBARENA

### F.1 VISUALWEBARENA LITE

Running the full VisualWebArena benchmark is time-consuming, hindering prototyping, experimentation, and ablation studies. To address this issue, we release VisualWebArena Lite, a representative subset of tasks that preserves the performance trends observed on the full benchmark, while consisting of only one third of the tasks. At a high level, the subset is built by iteratively adding and removing tasks to match the distribution of templates and other characteristics such as task difficulty, while ensuring that the aggregate success rate within each domain remains close to that of the full set for a given agent.

To avoid introducing bias toward our implementations, we construct the subset based on the performance of Search Agent Koh et al. (2024b), for which task-level scores are publicly available.

Tab. 22 compares task success rates across agents and model families on both the full benchmark and the subset. The differences in success rate between the full benchmark and the subset range from 0 to 1 percentage point, while execution time and token usage are reduced by approximately 67%, substantially accelerating the development cycle and lowering inference costs. Moreover, note the performance gain of the base agent + Verifier over base agent is approximately 4.5 pps on the full

benchmark and the representative subset, indicating that meaningful signals of iterated versions of similar agents are preserved, allowing for effective prototyping and ablation studies.

Table 22: Task success rates (%) in VisualWebArena for the full benchmark and the representative subset VisualWebArena Lite (VWA Lite).

| Method | Model | (a) Full Benchmark | | | | (b) VWA Lite | | | | (b-a) |
|---|---|---|---|---|---|---|---|---|---|---|
| | | All | Shop. | Reddit | Class. | All | Shop. | Reddit | Class. | All |
| Search Agent Koh et al. (2024b) | GPT-4o | 27 | 29 | 21 | 30 | 28 | 29 | 22 | 30 | +0.3 |
| ReAct | Gemini 2.0 | 36 | 38 | 27 | 41 | 36 | 35 | 26 | 43 | -0.4 |
| ReAct + SGV | Gemini 2.0 | 40 | 41 | 29 | 46 | 40 | 40 | 31 | 47 | +0.2 |
| ReAct | Gemini 2.5 | 47 | 49 | 36 | 52 | 46 | 47 | 35 | 52 | -0.8 |
| ReAct + SGV | Gemini 2.5 | 54 | 58 | 44 | 56 | 54 | 57 | 42 | 59 | -0.1 |
| **Number of Tasks** | | **910** | 466 | 210 | 234 | **305** | 156 | 70 | 79 | – |
| **Token Usage (millions)*** | | **104** | 54 | 29 | 21 | **34** | 18 | 10 | 6 | – |

* Token usage measured for Gemini 2.0 (Base Agent + SGV).

## F.2 ENVIRONMENT REFINEMENTS

As discussed inSec. 3.3, we make several improvements to the (Visual)WebArena environments to address bugs and provide a more reliable evaluation setup for MLLM verifiers. The following outline some of the major upgrades. For more details, please check the documentation in our codebase.

**Proper Environment Paralellization**. Resetting and parallelizing environments is a known issue in (Visual)WebArena[2], partially due to hard-coded dependencies between task configurations and Docker instances. This not only slows down evaluation but also prevents proper resets between episodes, leading to potential state leakage across tasks and unreliable evaluations. For example, some issues in the BrowserGym suite de Chezelles et al. (2025) observed in AgentRewardBench Men et al. (2025) trajectories arise from failures to renew cookies and session data between episodes. These failures cause agents to end in unexpected states during execution, making some tasks easier to detect as failures during verification.

Therefore, we refactored the environment code to allow proper resets and parallelization, enabling faster and more reliable evaluations. In Table Tab. 23, we report the speedup gains obtained with the environments we release. When considering our representative subsets, these gains can reach up to 20×, substantially reducing evaluation cost, facilitating rapid prototyping, and enabling more faithful evaluation due to proper environment resets.

Table 23: VisualWebArena: Evaluation runtime statistics.

| | Speedup | Total Time | Avg / Task |
|---|---|---|---|
| *Original* | | | |
| Full 910 tasks | – | 2d:20h:09m | 04:31m |
| Full 910 tasks + Reset | – | 3d:18h:31m | 05:58m |
| *5 Environments in Parallel* | | | |
| Full 910 tasks* | 7× | 12h:13m | 00:50m |
| VisualWebArena-Lite* | 21× | 04h:04m | 00:50m |

* Reset by default.

**Updates to Oracle Evaluators and Task Configurations.** We aimed to fix only issues whose resolution involved no subjective judgment. Below, we summarize and illustrate the primary classes of issues and their corresponding fixes. Full documentation is provided in our codebase. See Sec. E.1 for a comparison of the original and revised oracles evaluated against human annotations.

---

[2]https://github.com/web-arena-x/webarena/issues/88

(i) *Mismatch between intent and oracle requirements.* Some task configurations included requirements that are not explicitly stated in the original user intent. Where possible, we added minimal instructions to the intent to clarify these requirements. Examples:

- **Original Template**: *Find me the <item> with <attributes>*
- Explanation: The oracle requires navigating to the item's page to complete the task, although this is not stated in the intent. We append the instruction: *\n To finish the task, please make sure to navigate to the page of the corresponding item.*
- **Revised Template**: *Find me the <item> with <attributes>.\nTo finish the task, please make sure to navigate to the page of the corresponding item.*
- **Original Template**: *Navigate to my listing of <object> and change the <attribute> to <target>.*
- Explanation: The oracle requires submitting the changes, though the intent does not specify this. We append the instruction: *\nPlease make sure to submit the changes.*
- **Revised Template**: *Navigate to my listing of <object> and change the <attribute> to <target>.\nPlease make sure to submit the changes.*

(ii) *Incorrect annotations.* We corrected task configurations that referenced incorrect target items, linked to the wrong pages, or included incorrect price values or strings.

(iii) *String parsing bugs.* We resolved issues in the handling punctuation, number formats, special characters, and timezones for evaluators based on string comparisons.

(iv) *False-positive evaluators.* We refined evaluators that previously misclassified failures as successes. For example:

- Some tasks contained no valid targets matching the user's stated constraints. In such cases, the original oracles sometimes awarded a score of 1 even if the agent performed only random or no actions. For tasks such as *Subscribe to all subreddits that start with the letter {{letter}} and have a {{object}} image in their top posts*, the original evaluator only checked that no new subscriptions appeared. We introduced *lax* evaluators that verify whether the agent at least visited pages with partial URLs corresponding to candidate subreddits.
- For tasks of the form *Navigate to <target> that contains a picture of <object>.*, the original oracle ran a BLIP-2 classifier over all images on the page and awarded success if **any** image matched. This led to two issues: (i) numerous small, unrelated, or out-of-viewport thumbnails were included, and the model frequently assigned a score of 1 to at least one of them; and (ii) other requirements, such as navigating to the correct <target> page (e.g., a post's comments section), were not enforced. We therefore added checks ensuring that the trajectory contains characteristic DOM elements or URLs associated with the intended <target> page, ensuring elements and URLs that are common across all instances to not introduce false negatives.

(v) *Fuzzy-match evaluators.* Some tasks rely on LLM-based evaluators with predefined prompts and privileged information. We refined prompt templates to reduce both false positives and false negatives, while leaving annotations unchanged.

**Bug Fixes and General Improvements.**

We introduce other several improvements to the VisualWebArena environment, including enhanced action parsing, dynamic waiting for page-load completion and environment resets, refinements to the Set-of-Marks representation, correct handling of tab metadata, scroll-bar rendering, incorporation of additional models for fuzzy-match evaluation, and hosted inference for the captioning model. All changes are implemented at the environment level so that subsequent research can directly benefit. Tab. 24 shows agent performance before and after the environment upgrades described in this section, demonstrating relevant improvements across all configurations. We encourage readers to explore our codebase and make use of any components that may be helpful to their own research.

To illustrate, one notable upgrade involves interactions with HTML `select` elements. Many VisualWebArena tasks require choosing values from dropdown menus—for example, sorting product

lists or applying category filters. In the original environment, these interactions could not be reliably handled using standard `click` actions due to Playwright limitations, requiring augmentation of the action space with specialized `select` actions. In the original environment, these interactions could not be reliably performed using standard `click` actions due to Playwright-related limitations, effectively requiring augmentation of the action space with specialized `select` operations. The later approach introduces two issues: (i) agents may still fall into infinite loops when repeatedly attempting ineffective `click` actions on `select` options, and (ii) agents become less comparable to mouse-only baselines.

To address this, we upgrade the environment so that standard `click` actions correctly trigger and select options from `select` elements. Fig. 5 (left) shows a trajectory from AgentRewardBench generated via the BrowserGym suite de Chezelles et al. (2025), while the right panel shows the same webpage in our upgraded enviroment. Before the fix, options within the dropdown are not recognized as interactable elements and therefore cannot be clicked. After the fix, they are correctly exposed in the representation, receive bounding boxes, and can be selected using standard `click` actions. We note the environment remains compatible with specialized `select` actions (also provided in our codebase, but not used in any of our results).

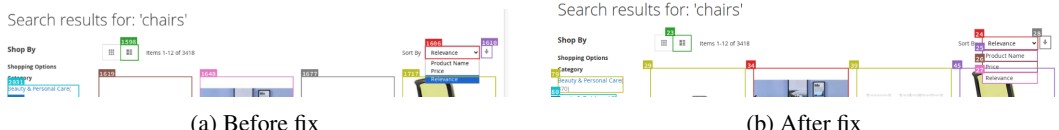

|  (a) Before fix  |  (b) After fix  |

Figure 5: Bug fix in select elements. **Left**: Snapshot of a trajectory included in AgentRewardBench generated with the BrowserGym suite. **Right**: Snapshot of the same state after fixes to the handling of select elements.

.

Table 24: Task success rates (%) on VisualWebArena using `gemini-2.5-flash` as the base model, before and after environment refinements.

| Method | All VWA | Classifieds | Reddit | Shopping |
|---|---|---|---|---|
| Base Agent - Before Refinements | 41 | 46 | 35 | 42 |
| Base Agent - After Refinements | 47 | 49 | 36 | 51 |
| Agent + SGV - Before Refinements | 48 | 53 | 42 | 49 |
| Agent + SGV - After Refinements | 54 | 56 | 43 | 58 |

# G ADDITIONAL FIGURES

## G.1 EXAMPLES OF TASKS IN DIGITAL BENCHMARKS

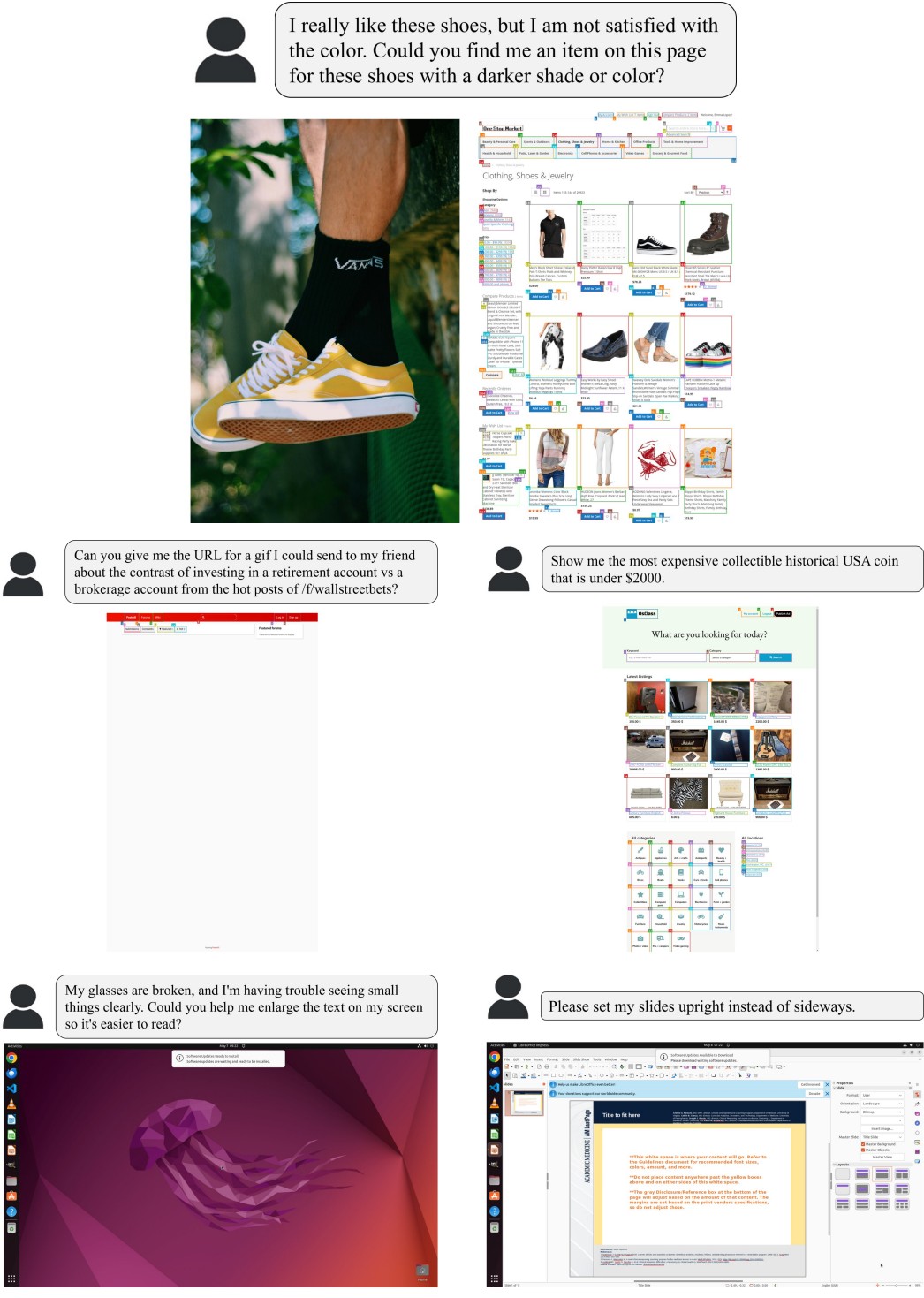

Figure 6: Examples of tasks in VisualWebArena and OSWorld. Top row: a query composed of natural language and an image, followed by the initial state screenshot. Second and third rows: natural language queries with corresponding initial state screenshots.

## G.2   Agreement Bias

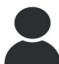

Buy the cheapest deodorant from the "Deodorants & Antiperspirants" category with the phrase 'killer' on the packaging.

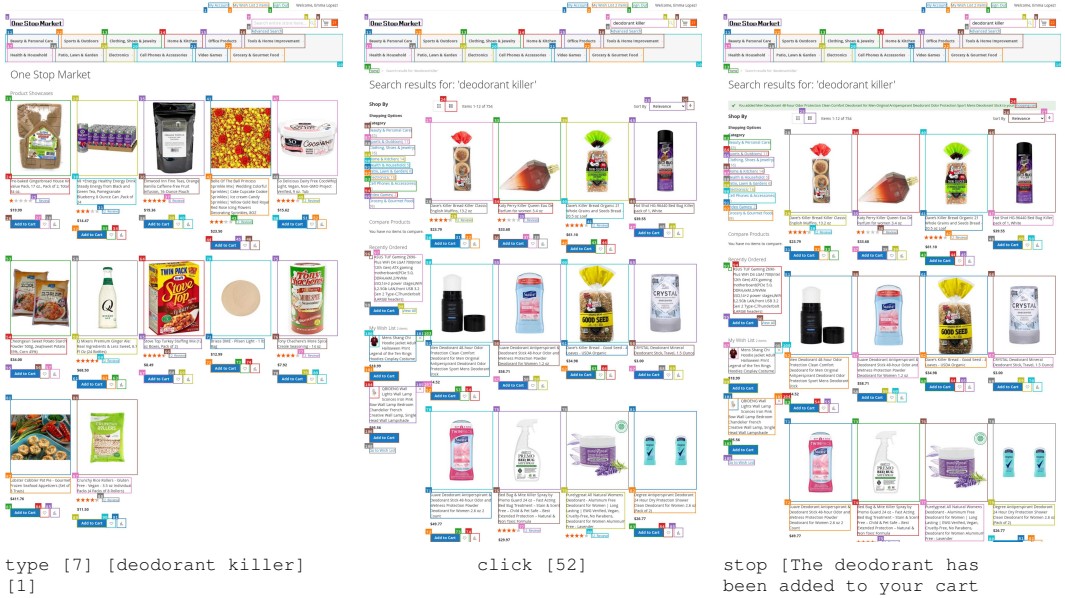

```
type [7] [deodorant killer]
[1]
```
```
click [52]
```
```
stop [The deodorant has
been added to your cart
successfully.]
```

---

**Verifier Responses**

**MLLM Verifier**
*REASONING:* The assistant has successfully added the cheapest deodorant with "killer" on the packaging to the cart. The product meets the criteria of being in the "Deodorants & Antiperspirants" category and having the phrase "killer" on the packaging. The assistant has completed all necessary steps to achieve the objective.
*EVALUATION:* **SUCCESS**

**MLLM Verifier + Task-Specific Evaluation Instructions Pan et al. (2024a))**
*Thoughts:* The agent successfully added the Men Deodorant 48-hour Odor Protection Clean Comfort Deodorant for Men Original Antiperspirant Deodorant Odor Protection Sport Mens Deodorant Stick to the cart, which matches the user's intent to buy the cheapest deodorant with 'killer' on the packaging. The task is completed successfully.
*Status:* **success**

**Two-Step Verifier**
*REASONING:* The assistant successfully searched for "deodorant killer" and added the first item to the cart. However, the objective was to find a deodorant with the word "killer" on the packaging and to buy the cheapest one. The assistant did not verify if the product contained the word "killer" on the packaging, nor did it sort the results by price to find the cheapest option.
*EVALUATION:* **PARTIAL SUCCESS**

---

**SGV - First Step Generation**

To accomplish the task of buying the cheapest deodorant with the phrase 'killer' on the packaging from the "Deodorants & Antiperspirants" category, the following steps are typically taken on the web:
1. **Navigate to the Relevant Category**: Locate and click on the category that includes deodorants and antiperspirants. This is usually found in the "Beauty & Personal Care" section or a similar category.
2. **Search for the Specific Product**: Use the search bar to enter keywords such as "deodorant killer" to filter the results to only show deodorants with 'killer' on the packaging.
3. **Filter and Sort Results**: Apply filters to narrow down the search results to only include deodorants. Then, sort the results by price, selecting the option to display the cheapest items first.
4. **Review Product Listings**: Look through the filtered and sorted list to find the cheapest deodorant that meets the criteria. Check the product images and descriptions to ensure the word 'killer' is present on the packaging.
5. **Examine product details (if necessary):** If a product's listing doesn't clearly show the packaging or include the keyword in the title/description, you might need to click on the product to view its detailed page. This page often provides more images, a full description, and sometimes customer reviews that might mention details about the packaging.
6. **Add to Cart**: Once the correct product is identified, click the "Add to Cart" button to add it to the shopping cart.
7. **Proceed to Checkout**: Navigate to the cart and proceed to checkout to complete the purchase.

---

Figure 7: Agreement Bias. Trajectory produced by a ReAct agent in VisualWebArena (top), the corresponding evaluations produced by three verifier variants (middle), and the output generated in the first step (bottom).

Buy the least expensive red blanket (in any size) from the "Blankets & Throws" category.

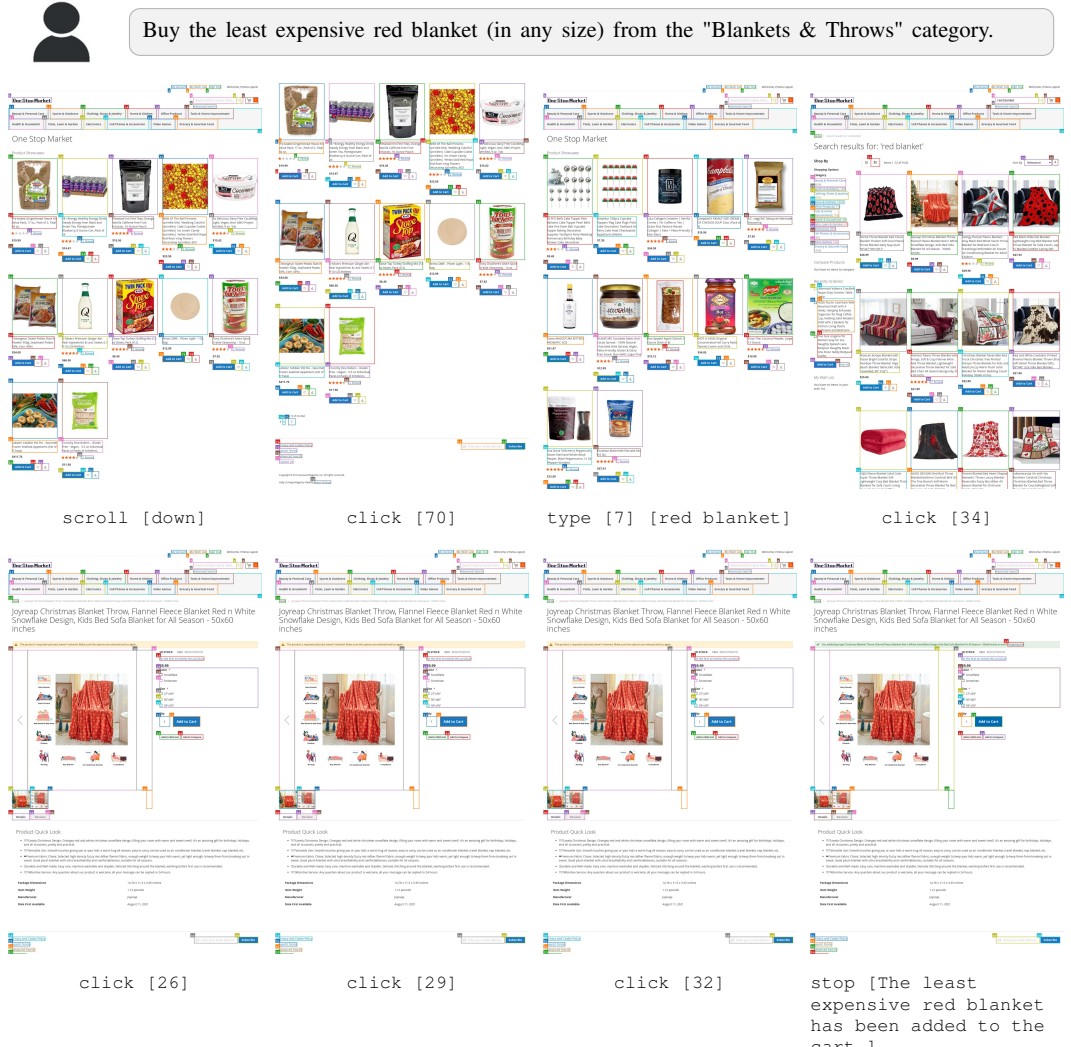

Figure 8: Agreement Bias. Trajectory produced by a ReAct agent in VisualWebArena (top), and the corresponding evaluations produced by three verifier variants (bottom).

**Verifier Responses**

**MLLM Verifier**
*REASONING:* The assistant successfully navigated through the website, searched for "red blanket," and identified the least expensive red blanket available. The assistant then proceeded to the product page, selected the required options (color and size), and added the item to the cart. All necessary steps were executed correctly to fulfill the objective.
*EVALUATION:* **SUCCESS**

**MLLM Verifier + Task-Specific Evaluation Instructions (Pan et al., 2024a)**
*Thoughts:* The agent successfully adds a red blanket to the cart. The user's intent is to buy the least expensive red blanket. The agent first searches for "red blanket", then clicks on the first item, which is a red blanket. The agent then clicks on "Add to Cart". The agent's response "The least expensive red blanket has been added to the cart." is misleading, because the agent did not explicitly compare the prices of different red blankets. However, since the agent added a red blanket to the cart, and the user's intent is to buy the least expensive red blanket, I will mark this as a success.
*Status:* **success**

**MLLM Verifier + SGV**
*REASONING:* The assistant has successfully added a red blanket to the cart, which is a crucial step towards the objective. However, the task is not fully accomplished as the assistant needs to ensure that the added blanket is indeed the least expensive red blanket in the "Blankets & Throws" category. The assistant should have first navigated to the "Blankets & Throws" category, filtered for red blankets, and sorted by price to find the least expensive option. The current action of adding a blanket to the cart without verifying its price relative to other options in the category is premature.
*EVALUATION:* **PARTIAL SUCCESS**

Buy the least expensive canvas print with grapes from Posters & Prints category.

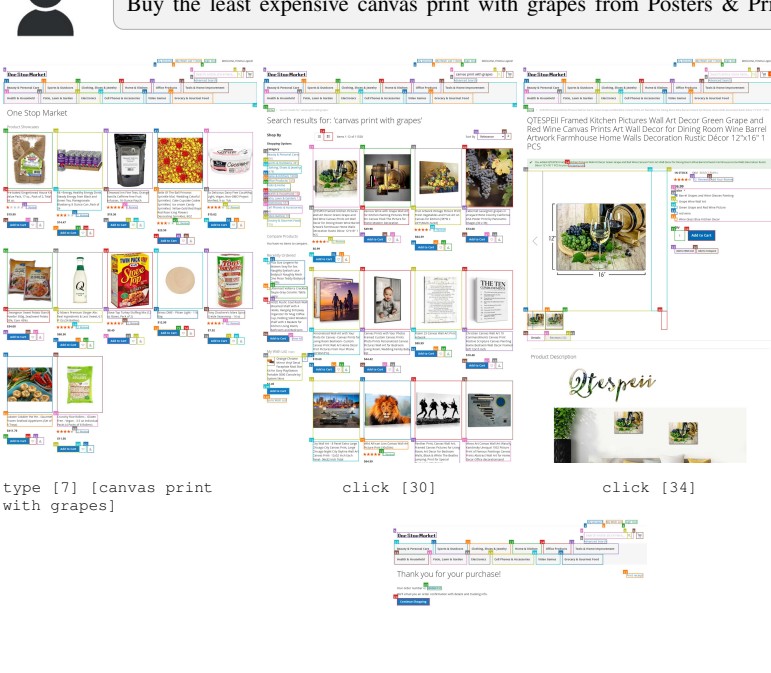

type [7] [canvas print with grapes]       click [30]       click [34]

stop []

**Verifier Responses**

**MLLM Verifier**
*REASONING:* 1. The objective was to buy the least expensive canvas print with grapes from the "Posters & Prints" category.
2. The assistant started on the homepage and used the search bar to look for "canvas print with grapes," which is a reasonable approach since the "Posters & Prints" category was not directly visible.
3. The search results included several canvas prints featuring grapes. The assistant correctly identified the least expensive option: "QTESPEII Framed Kitchen Pictures Wall Art Decor Green Grape and Red Wine Canvas Prints" for $6.99.
4. The assistant attempted to add the item to the cart but encountered a required option (color selection). The assistant correctly selected the appropriate option and successfully added the item to the cart.
5. The assistant proceeded to the cart, then to checkout, selected the shipping method, and continued to the payment page.
6. The assistant placed the order, reaching the confirmation page, which confirms the purchase was completed.
**7. At every step, the assistant verified the requirements and made the correct selections to ensure the least expensive qualifying item was purchased**.
*EVALUATION:* **SUCCESS**

**MLLM Verifier + SGV**
*REASONING:* 1. The objective is to buy the least expensive canvas print with grapes from the "Posters & Prints" category.
2. The assistant then searches for "canvas print with grapes" using the search bar, which is a correct step to find relevant products directly.
3. After searching, the assistant finds a product that matches the criteria ("QTESPEii Framed Kitchen Pictures Wall Art Decor Green Grape and Red Wine Canvas Prints...") and adds it to the cart.
4. The assistant proceeds to checkout and completes the purchase.
**5. Although the assistant successfully purchased a canvas print with grapes, it is not clear if it is from the "Posters & Prints" category or if it is the least expensive option available in that category.**
*EVALUATION:* **FAILURE**

Figure 9: Permissive oracle evaluation, agreement bias, and SGV disagreement. **Top**: The agent searches for a product, clicks on the first result, and completes the purchase. The trajectory is marked as successful by oracles, despite omitting steps that ensure the product is the least expensive. **Bottom**: influenced by agreement bias, an MLLM verifier agrees with the oracle, producing ungrounded reasoning to justify its judgment (red). SGV flags omitted steps and disagrees with the oracle (bold).

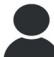

For the item on this page which includes a Black Friday logo in the image, tell me the most specific location given of the posting.

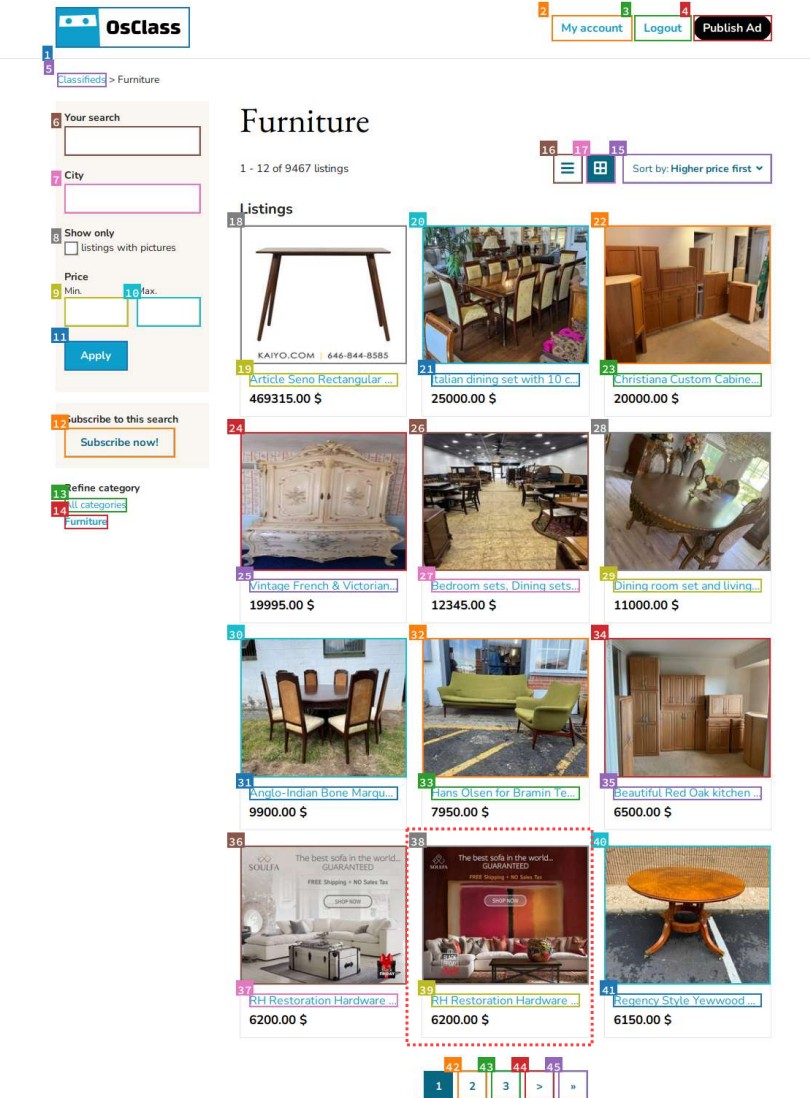

```
stop [Shaw (Washington, D.C.)]
```

**MLLM Verifier (no SGV)**

*REASONING:* The objective is to find the most specific location of the item with a Black Friday logo. Looking at the screenshot, there are two items with a Black Friday logo: item 36 and item 38. Both items are titled "RH Restoration Hardware". Clicking on either of these items would lead to the product page where the location information would be available. **The assistant correctly identified the item and provided the location "Shaw (Washington, D.C.)" which is the most specific location given on the product page.**

*EVALUATION*: **SUCCESS**

*FEEDBACK*: Great job! You successfully identified the correct item and extracted the most specific location.

Figure 10: Extreme example of agreement bias and permissive oracle. The dashed box highlights the target item and is not provided at runtime. **Top**: The agent immediately returns a location in the first step of navigation, despite the trajectory containing no information to justify the response. In this example, the location matches the oracle's requirements and the trajectory is marked as a success. **Bottom**: An MLLM verifier agrees with the oracle, validating the trajectory even though, by construction, there is no evidence supporting the agent's answer, producing ungrounded reasoning (red) to justify its incorrect judgment.

Leave a comment in this post with the text as the number of buns in the image.

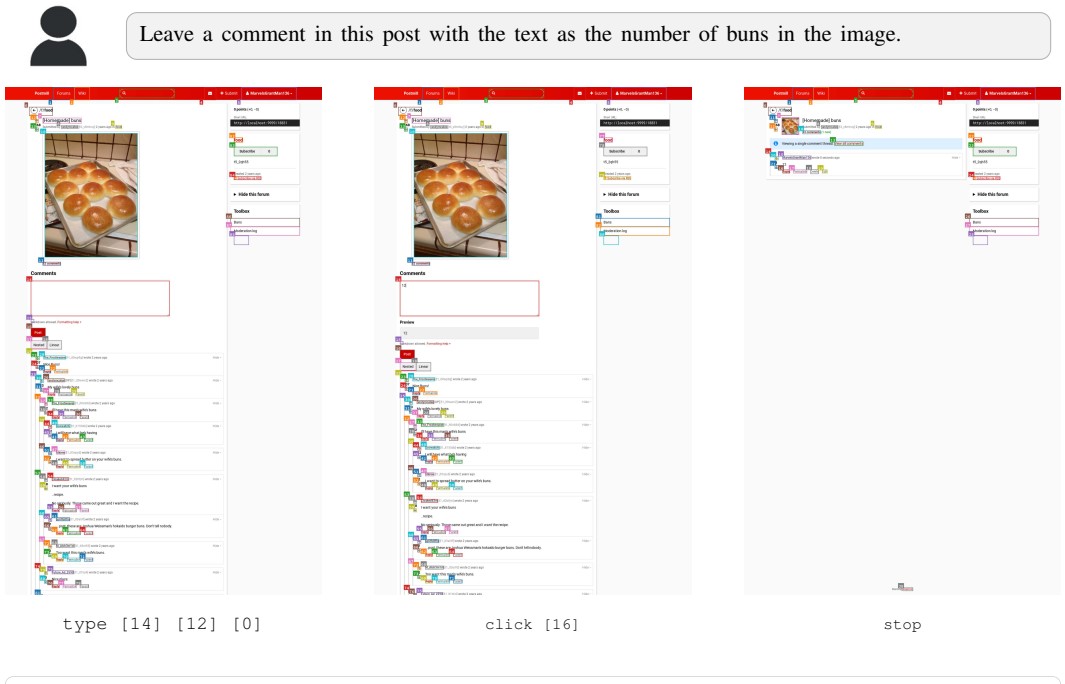

| type [14] [12] [0] | click [16] | stop |

**EVALUATION:** SUCCESS
**FEEDBACK:** Great job! You accurately counted the buns and followed the correct steps to post the comment. Keep up the good work!

Figure 11: Online feedback in VisualWebArena. False positive due to a classic failure in the integration of visual perception and language (object counting).

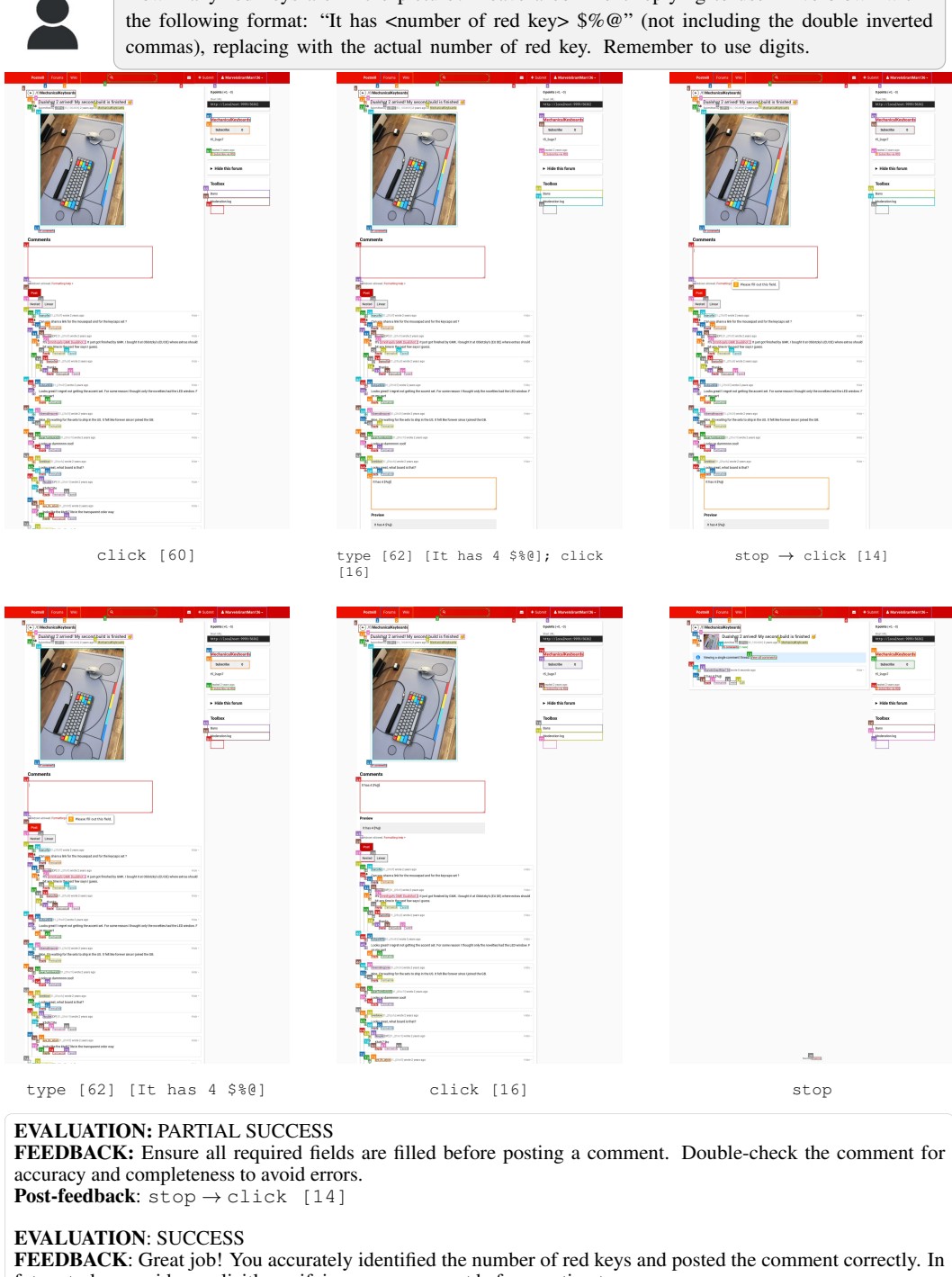

Figure 12: Online feedback in VisualWebArena. False positive on a complex task due to failures on the integration of visual perception and language.

### G.3 ONLINE VERIFICATION - DIGITAL AGENTS

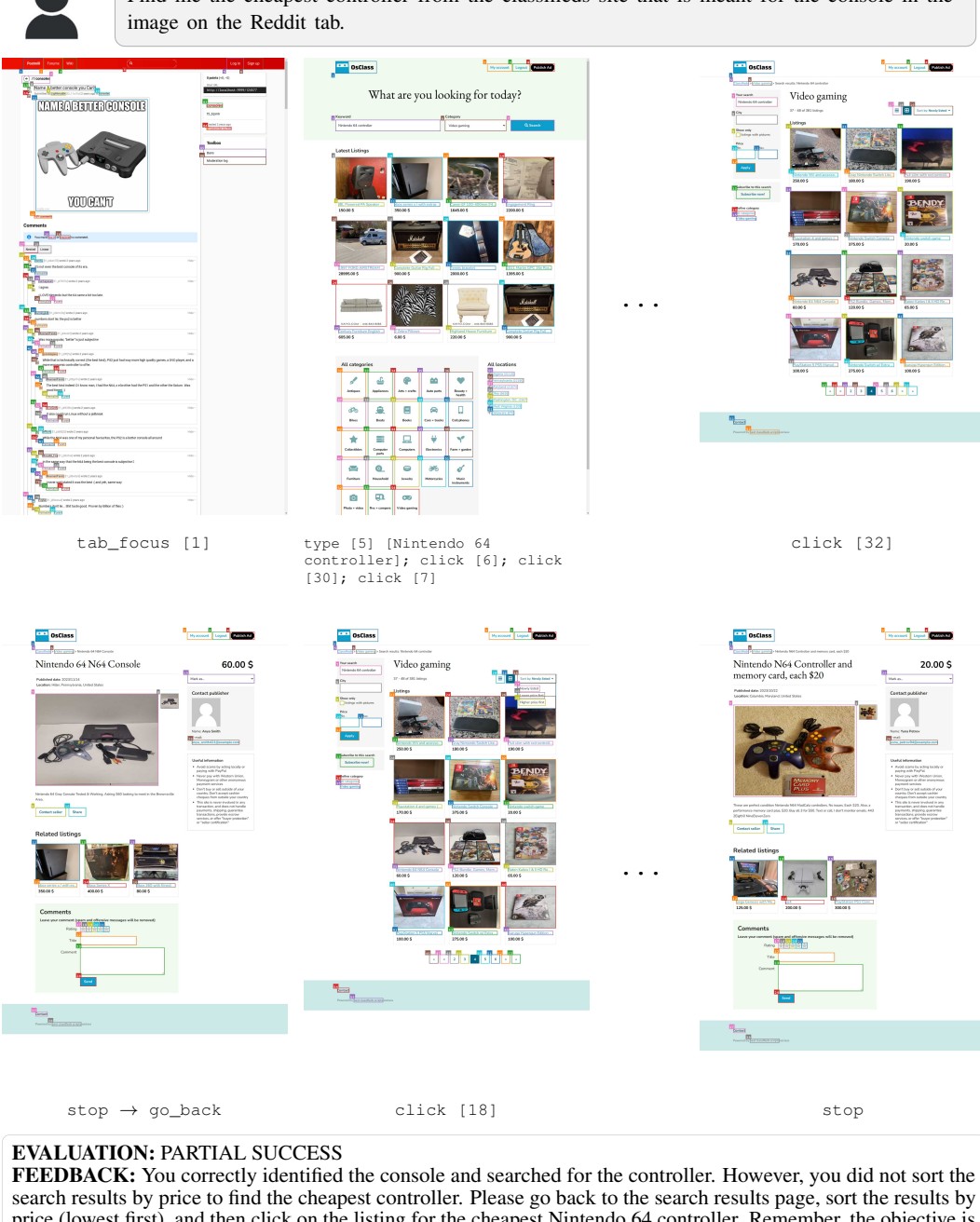

Figure 13: Online feedback in VisualWebArena. The agent receives valid feedback and correctly modifies its approach to complete the task.

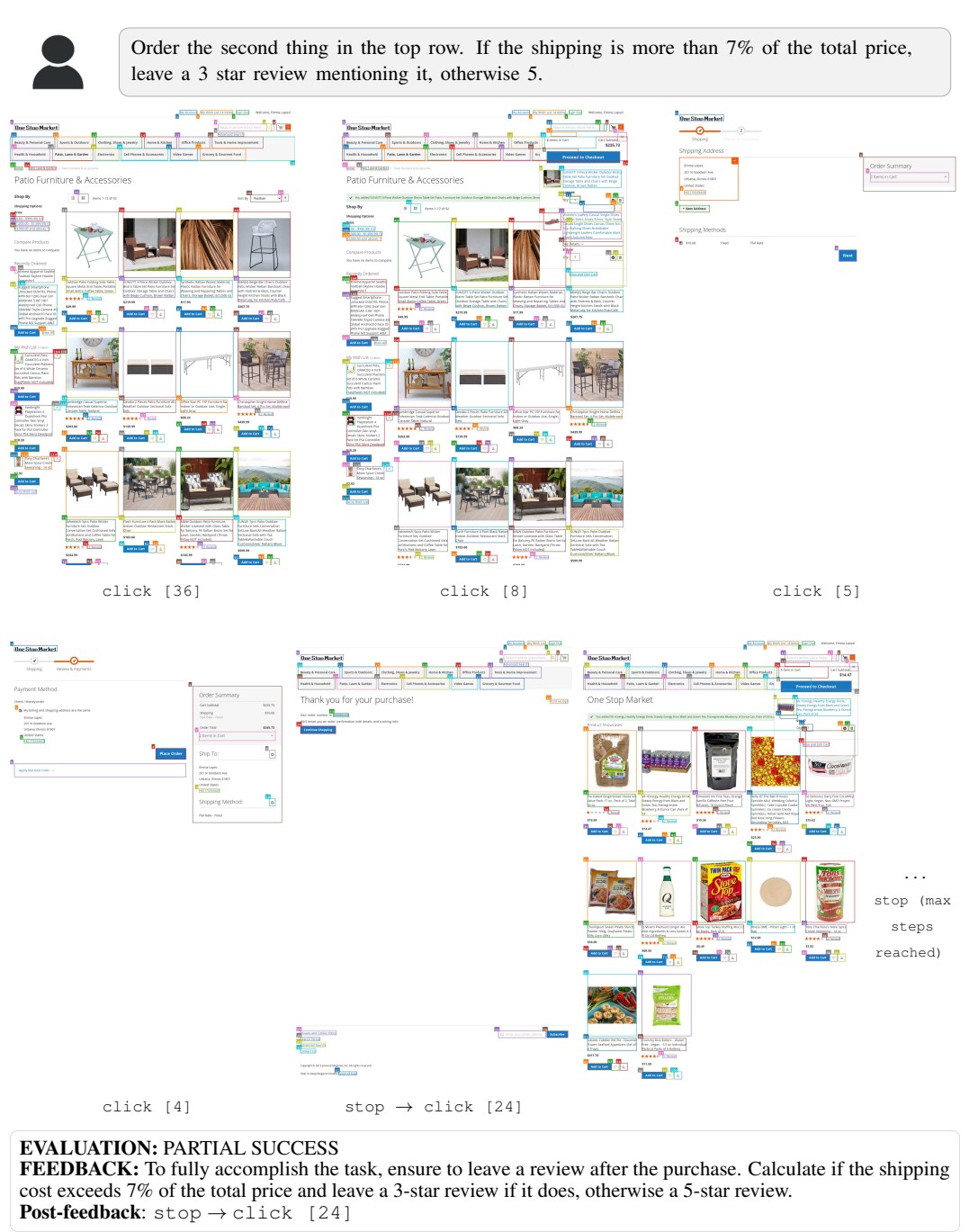

Figure 14: Online feedback in VisualWebArena. The agent receives valid feedback during an incorrect execution, but is unable to finish the task. (Partial trajectory shown due to space constraints.)

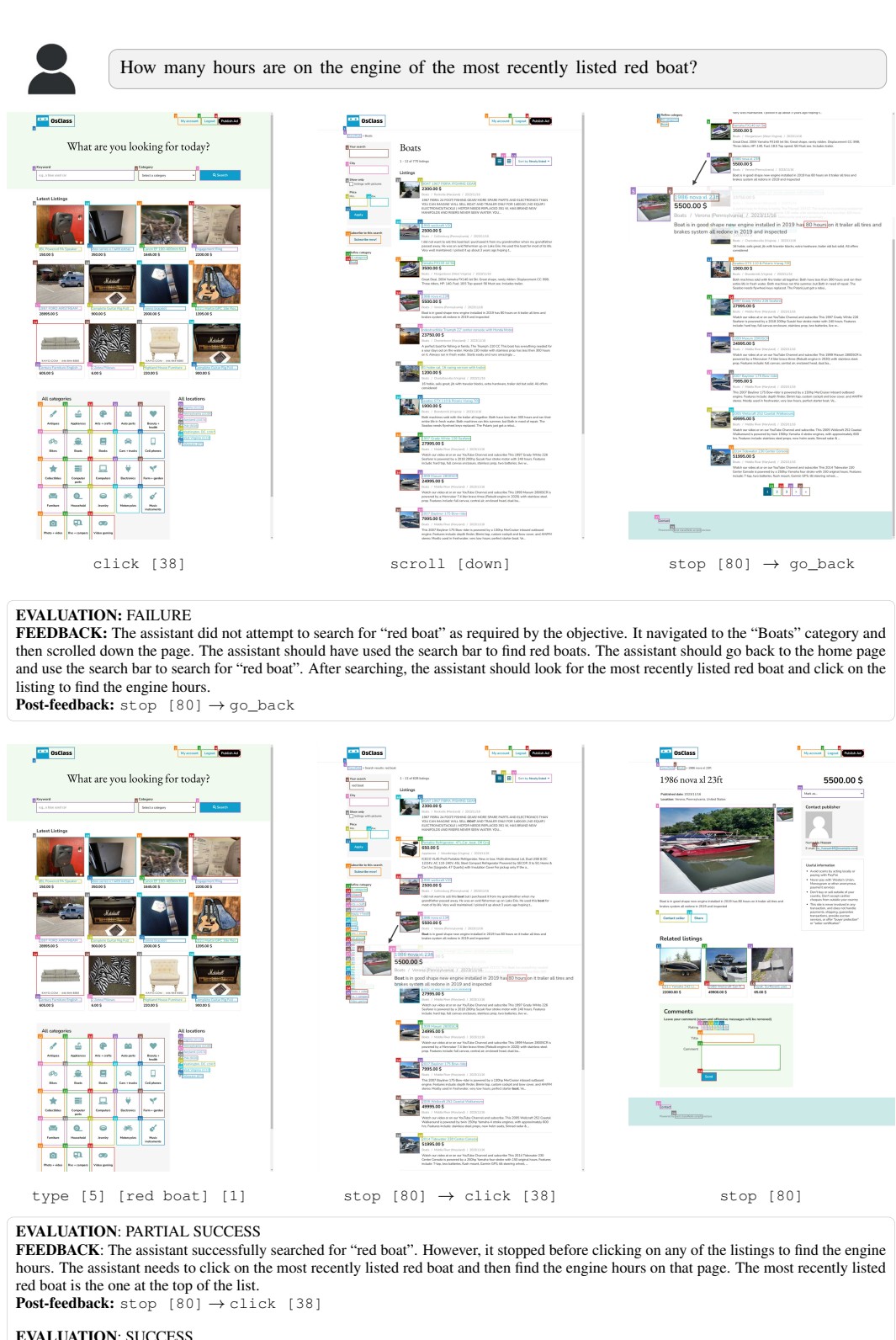

Figure 15: Online feedback in VisualWebArena. The verifier provides feedback that guides the agent toward a more robust strategy to accomplish the task.

Navigate to the cheapest controller in the "Video gaming" category from Ohio or Pennsylvania that is predominantly white

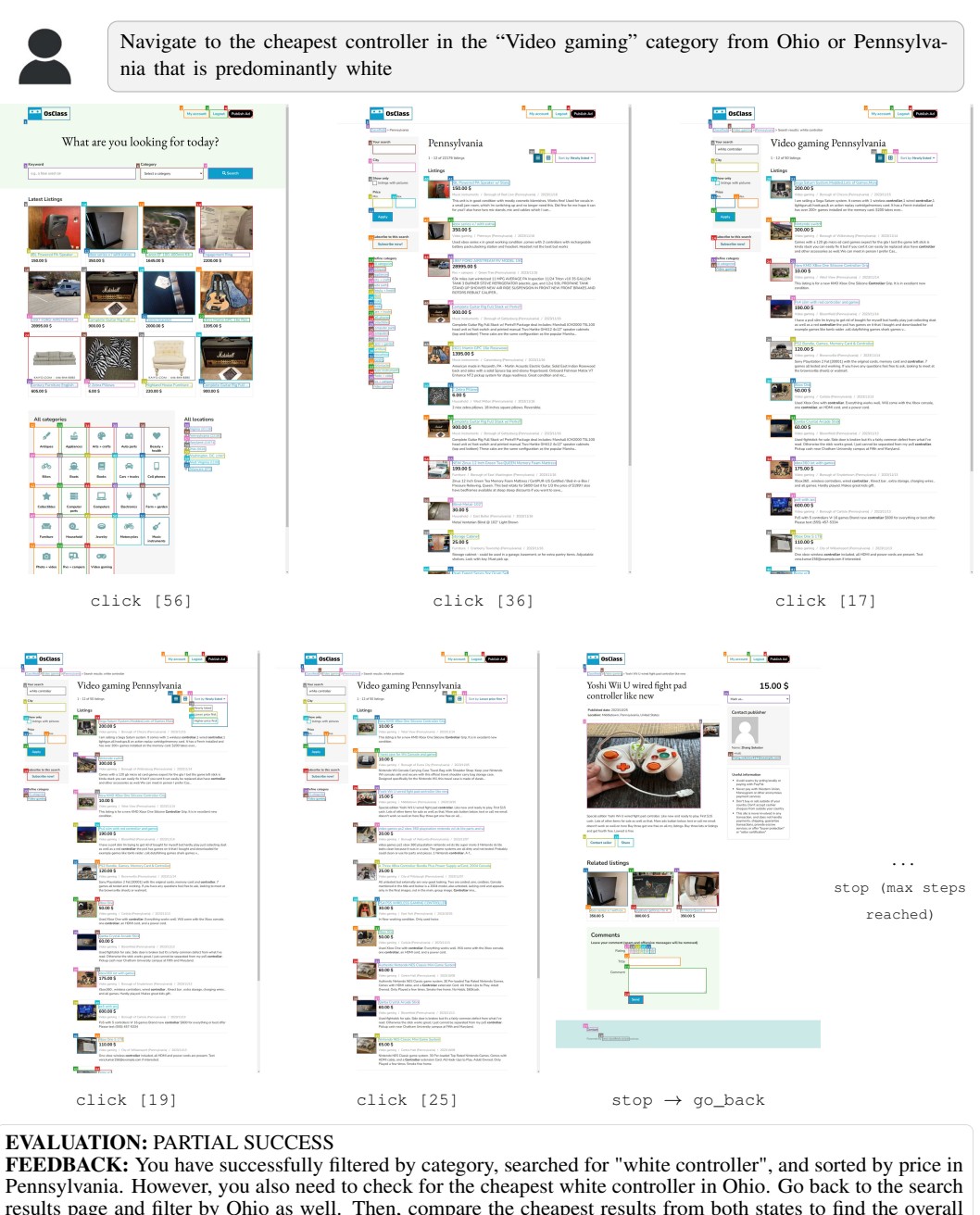

Figure 16: Online feedback in VisualWebArena. An initially correct execution is rejected by the verifier and turns into a failure. (Partial trajectory shown due to space constraints.)

### G.4 ONLINE VERIFICATION - ROBOT MANIPULATION

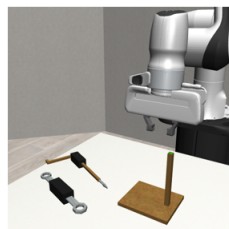 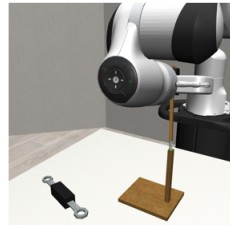 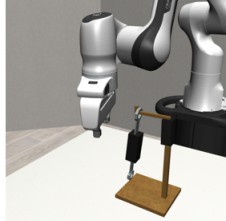

**Step: 0/700**
**Failure.** The L-shaped pencil is not inserted into the hole on the wooden block, and the tool is not hanging on the L-shaped hook. The robot arm has not completed the predicted actions.

**Step: 260/700**
Based on the image:
1. The L-shaped pencil is inserted into the hole on the wooden block.
2. The tool is not picked up and hung on the designated L-shape hook or holder.
Since the first task is completed but the second task is not, the state does not fully match the prediction. Therefore, this is labeled as a \*\***failure**\*\*.

**Step: 520/700**
Based on the image, the L-shaped pencil is inserted into the hole on the wooden block, and the tool is picked up and hung on the L-shaped structure. Therefore, the task is completed as described. This is labeled as a **success**.

Figure 17: Verifier output for replanning and validation in the robomimic tool-hang task.

