# OpenReview forum: "Let's Think in Two Steps: Mitigating Agreement Bias in MLLMs with Self-Grounded Verification"
_ICLR.cc/2026/Conference — ICLR 2026 Poster_

### Official Review · Reviewer_Uw7g · 2025-10-30

**Soundness:** 3
**Presentation:** 3
**Contribution:** 3
**Rating:** 6
**Confidence:** 4

**Summary:**

This work investigates the issue of agreement bias in vLLMs when used as evaluators. Specifically, vLLMs are employed to assess intermediate steps in multimodal agent tasks, serving as a form of reward or signal that can facilitate test-time scaling. However, the authors find that vLLMs may generate chains of thought to rationalize potentially flawed actions, even when their evaluative knowledge is aligned with human judgment. To mitigate this issue, the authors propose having the vLLM first generate a plausible prediction of the next state before conducting evaluation, which effectively reduces agreement bias. The problem is studied across domains such as web agents, GUI agents, and robotics.

**Strengths:**

The research question is both interesting and important, though the phenomenon of agreement bias and the proposed solution might appear somewhat intuitive or straightforward. The experiments are comprehensive and the results are solid. The paper is also very well written — I appreciate that the abstract clearly conveys the main takeaways.

**Weaknesses:**

A critical problem remains: I’m particularly curious whether you have any further experiments and analysis on the “generating chains of thought to rationalize flawed behavior” aspect. You also claim that MLLMs exhibit strong, human-aligned priors on desired behavior — so at which reasoning step exactly does the failure occur? Is this bias intrinsic to the model itself, and beyond prompting strategies, are there training or sampling techniques that could help mitigate this issue?

**Questions:**

Please check the weaknesses section.

---

> ### Author Response · Authors · 2025-11-27
> **Author Response - Part 1**
>
> Thank you for the thoughtful and positive evaluation of our work. We are encouraged that the reviewer found our study of agreement bias in MLLMs both interesting and important, and we appreciate the recognition of the strength of our results, the breadth of our experimental analysis, and the clarity of our presentation. We are also grateful for the reviewer's insightful questions, which motivated valuable improvements to our manuscript.
>
> All feedback has been incorporated into our revised manuscript. We hope that the additional results and clarifications help address the reviewer's concerns and can enable an even more positive evaluation of our work. We remain open to providing any further clarifications or analyses the reviewer may find helpful.
>
> ----
>
> To facilitate discussion, we broke down W1 into two parts, each covered in separate comments below.
>
> > "Is this bias intrinsic to the model itself?" + "Are there training or sampling techniques that could help mitigate this issue?"
>
> Thank you for raising this interesting discussion! We think the answer to both questions is yes, as our results show that (i) agreement bias manifests as a systematic skew in MLLM verifiers' output distributions toward positive evaluations, and (ii) can be leveraged to inspire other techniques aimed at improving MLLM verifiers. We elaborate below and in the following comment.
>
> First, we note this question relates to our results on the distribution of MLLM responses (Figure 3 (Appendix F)) and discussion to address R3 concerns. Particular points of interest:
> - Across 28 prompt/evaluation templates, we observe that MLLMs consistently concentrate evaluations at the high end of the scales. This occurs despite prior explored to reduce bias in LLM evaluators, such as criteria order shuffling and reversal, and syntactic structure changes.
> - SGV produces much more balanced score distributions across all templates
> - Applyig a Platt scaling calibration to MLLM confidence outputs does not mitigate agreement bias due to the highly skewed distributions
>
> For convenience, the table of average performance across templates is available at this [link](https://anonymous.4open.science/r/abias-sgv-iclr-26-3DD2/R2..1.md), the per‑template score distributions at this [link](https://anonymous.4open.science/r/abias-sgv-iclr-26-3DD2/R2..2.md), and the responses after platt scaling calibration at this link at this [link](https://anonymous.4open.science/r/abias-sgv-iclr-26-3DD2/Table-r2.3.md).

---

> ### Author Response · Authors · 2025-11-27
> **Response Part 1 (continued)**
>
> **Platt Scaling on Sampling-derived Probabilities**
>
> Inspired by the above results and R4's inquiry on potential sampling or training strategies to mitigate agreement bias in MLLM verifiers, we conducted a proof-of-concept experiment exploring calibration via Platt scaling on implicit probabilities derived from MLLMs during sampling generation.
>
> We focus this analysis on VisualWebArena, where our agents produce a roughly balanced set of successes and failures. Specifically, for each of the 910 trajectories, we sample 8 completions and estimate empirical probabilities by measuring the proportion of times the MLLM labels a trajectory as a success or failure under the best-performing template across the 28 tested templates---the ternary Likert scale used in our baselines. To test the Platt calibrator, we utilize our 1/3-sized representative subset that provides the same distribution as the full benchmark (Appendix G.1), and we train it on the remainder of the data (~605 datapoints). In both cases, the data is roughly balanced, with 47% success and 53% failed trajectories.
>
> Before discussing results, we emphasize that this is a **proof of concept** experiment under relatively *ideal* and *expensive* conditions. Particularly:
>
> - The calibrator is trained and tested on balanced sets with roughly balanced numbers of successes and failures.
> - It assumes access to the high-quality labels provided by our improved VisualWebArena oracles (lines 190)---a limitation in real-world settings that motivates the need for MLLM verifiers in the first place.
> - Training and test domains match; i.e., we do not consider out-of-distribution generalization.
> - To estimate probabilities, we sample a relatively large number of completions (8 per trajectory, totaling over 7,200 generations).
> - The thresholds used to separate classes—which, as we show below, can be a sensible choice—are optimized under these same ideal conditions.
> - At test time, we still generate 8 completions per new trajectory for probability estimation and apply the calibrator to them.
>
> Results are shown in the table below. In summary:
>
> - Vanilla majority voting fails to deliver improvements by not taking into account the skew in MLLM label distributions toward positive evaluations.
> - Leveraging this fact to apply a calibration to MLLM implicit probabilities leads to more balanced distributions and improvements in evaluation metrics.
> - Most notably, SGV leads to distributions superior to the calibrated ones obtained in ideal scenarios through a single generation.
> - Importantly, gains arise organically by leveraging the model's own sampling mechanisms, without the need for ground-truth labels, additional training, or calibration steps. Moreover, these gains naturally extend to other domains, as demonstrated by SGV's improvements across benchmarks.
> - SGV also avoids complicated hyperparameter tuning, such as calibration thresholds. As shown in columns 'p15' and '0.5', setting this threshold to either the 15th percentile (bootstrap-estimated) or naïvely to 0.5 mostly eliminates the gains from Platt scaling.
>
> **Table R4.1: Platt Scaling vs SGV, template = Ternary Likert, Sampling Generation**
> |Metric|No SGV, temp=0|No SGV, majority|**Platt**|Platt, p15|Platt, 0.5|**SGV (temp=0)**|SGV, Majority|SGV, Platt|
> |-|-|-|-|-|-|-|-|-|
> |ACC|65|65|**72**|69|71|**76**|75|78|
> |TPR|92|93|**77**|85|86|**84**|85|79|
> |TNR|41|40|**68**|56|57|**71**|67|78|
> |Bias|27|28|**6**|20|16|**11**|11|1|
> |dSkew|23|24|**2**|15|9|**5**|5|0|
> |AUC|67|66|**73**|75|75|**76**|76|83|
>
> We hope these results reinforce the effectiveness of SGV in mitigating a bias that stems from intrinsic MLLM response distributions. We believe these results suggest that there is potential for both training-time and test-time approaches that explicitly account for the asymmetries introduced by agreement bias---e.g., loss functions that penalize such imbalance, or adjust sampling mechanisms for methods that rely on sampling (e.g., GRPO). Since SGV itself induces substantially more balanced output distributions, incorporating SGV into larger pipelines may also be fruitful---e.g., as a replacement for oracle labels in methods like the one above.
>
> We hope that our identification of agreement bias, our demonstration of the risks it poses for applications relying on MLLM verifiers, our evidence on the benefits of mitigating this bias to downstream applications, and the insights provided by our work collectively help motivate further research in this direction.
>
> We thank R2 and R4 for raising this discussion and prompting us to better leverage these results. We have included this discussion in our revised version and leave further exploration to future work.

---

> ### Author Response · Authors · 2025-11-27
> **Author Response - Part 2**
>
> > Is this bias intrinsic to the model itself + experiments on chain-of-thought
>
> **Quantitatively** the main ablations we conduct in regards to CoT are provided in Table 2, included below with all evaluation metrics. The table shows results considering the Gemini 2.5 family, but we note that the same CoT setting is applied to all models in the paper.
>
> We observe limited mitigation of agreement bias either (i) when MLLMs are prompted to generate chains of thought to conduct the verification (2 vs 1), and (ii) for models that are trained to produce reasoning traces autonomously (LRMs) (5 vs 1). That is, in both cases, MLLMs fail to produce completions $k$ to condition themselves that mitigates agreement bias and promotes effective verification. This is particularly striking for LRMs, given that they are trained to produce whichever completions they need to maximize performance.
>
> **Table R4.2: Impact of Test-Time-Scaling Techniques**
> |#|Method|Acc (VWA)|TPR|TNR|Bias|dSkew|Acc (OSW)|TPR|TNR|Bias|dSkew|
> |-|-|-|-|-|-|-|-|-|-|-|-|
> |1|No CoT|64|91|44|28|24|68|97|60|30|30|
> |2|CoT|65|90|47|26|21|71|97|63|28|27|
> |3|CoT (M)|67|92|48|27|23|69|97|62|30|29|
> |4|Pan et al.|66|83|53|21|14|–|–|–|–|–|
> |5|Thinking|70|90|55|23|18|78|95|73|20|18|
> |6|Thinking (M)|70|91|55|24|20|76|95|71|22|20|
> |7|SGV|76|84|71|11|5|83|92|81|13|11|
> |8|SGV (T)|77|86|70|12|6|87|91|86|8|5|
>
> The table below further elucidates from the perspective of the MLLM output distributions. It reports how often MLLM verification matches the oracles over 10,000+ samples stratified by success and failure subsets, measuring the probability of sampling a correct verification from the MLLM output distribution. In line with part 1:
> - Over-validation is reflected in the model's internal distribution, with the probability of sampling a correct response from the MLLM on failure subsets reaching values at or below chance (e.g., 48% in VWA)
> - CoT is limited in mitigating this effect, including when models are allowed to produce reasoning traces autonomously
> - Under SGV, the distribution gets substantially more balanced
> - This explains the ineffectiveness of methods like majority voting, and suggests an imbalance to be accounted for in test-time and training methods that rely on sampling
>
> **Table R4.3: Agreement Rates in Sampling Generation**
> |Method|VWA|| |OSW|||
> |-|-|-|-|-|-|-|
> | |All|F|S|All|F|S|
> |No SGV|65|48|91|71|56|93|
> |No SGV (T)|68|55|90|79|65|93|
> |SGV|75|70|86|85|77|89|
> |# Samples|7280|-|-|2784|-|-|
>
> **Qualitatively**, we observe that, in several failure cases, models generate chains of thought that rationalize incorrect behavior. Besides the examples in Figs. 5 and 6, this [additional example](https://anonymous.4open.science/r/abias-sgv-iclr-26-3DD2/abias-extreme-example.png) (included in our revised paper) illustrates an extreme case: the agent gives an answer not supported by any information in the trajectory, and an MLLM generates reasoning to validate the behavior, even though, **by construction**, no evidence exists to justify that judgment.
>
> Although we do not examine all reasoning traces (which for only one setup surpass 1,300 trajectories), we note that agreement bias is a failure mode where MLLMs are biased toward **positive** labels. Here, this means that models generate chains of thought that lead to the **SUCCESS** keyword too often. Because we use very strong models, their reasoning traces are typically coherent; therefore, instances where the model constructs narratives, like in the examples shown, are likely to be common.

---

> ### Author Response · Authors · 2025-11-27
> **Response Part 2 (Continued)**
>
> > You also claim that MLLMs exhibit strong, human-aligned priors on desired behavior--so at which reasoning step exactly does the failure occur?
>
> The fact that a simple strategy as SGV, where in the first step models generate completions $k$ that in the second step produce verification substantially more aligned to human/oracle evaluations across models--and strikingly, for LRMs--*suggests* that: (i) models seem to have the necessary knowledge to verify about the tasks, however (ii) they fail to utilize it effectively when acting as a verifiers, with (iii) a systematic tendency to make **false-positive errors**, rather than errors in both directions. Given that this bias is pervasive across models, appears in the MLLM implicit distributions, and is robust to several prompt design and other interventions, we *conjecture* that it stems from limitations inherited from more fundamental stages of the MLLM training. In this regard, we find two observations from LLM literature insightful: (i) that the way a transformer's knowledge is embedded within its parameters can hinder extraction depending on how the information is presented [1,2] and (ii) that inherent limitations from RLHF can lead models to conflate human rater satisfaction with truthfulness [3]. In combination, these limitations (and potentially others) *may* give rise to the bias toward favorable evaluations we identify.
>
> However, while we draw inspiration from works in LLM interpretability, determining precisely at which step of text (or embedding space) this bias arises is, we believe, a substantial research direction on its own. We hope our work demonstrates that this is a relevant problem to the community and provides insights that motivate future research in this direction.
>
> [1] Zeyuan Allen-Zhu and Yuanzhi Li. Physics of language models: Part 3.1, knowledge storage and extraction, 2024a.
>
> [2] Zeyuan Allen-Zhu and Yuanzhi Li. Physics of language models: Part 3.2, knowledge manipulation,
> 2024b.
>
> [3] Mrinank Sharma et al. Towards understanding sycophancy in language models, 2025.

---

### Official Review · Reviewer_twch · 2025-11-01

**Soundness:** 3
**Presentation:** 3
**Contribution:** 3
**Rating:** 4
**Confidence:** 3

**Summary:**

This paper 1) identifies "agreement bias", a tendency for MLLMs to inappropriately favor agent trajectories in their context window, as a critical limitation for MLLM-based verifiers, and 2) proposes Self-Grounded Verification (SGV), a two-step method that retrieves task priors first then evaluates trajectories against them, achieving up to 20pp gains in verification accuracy and setting a new SOTA on VisualWebArena.

**Strengths:**

1. Originality: Framing "agreement bias" as a distinct limitation from self-bias and targeting it via self-generated priors is a novel angle for MLLM verifiers.
2. Quality: Experiments use diverse benchmarks (1,200+ tasks) and models, ensuring results are generalizable rather than model-specific.
3. Clarity: The SGV method is described simply, with step-by-step breakdowns and concrete examples (e.g., Figure 5) making it easy to follow.
4. Significance: Improving MLLM verifier reliability directly benefits downstream tasks like agent training, data filtering, and real-time supervision, a key for deploying AI agents safely.

**Weaknesses:**

1. SGV does not address underlying vision-language flaws (e.g., Figure 7’s counting error), and the paper lacks discussion on combining SGV with specialist models for fine-grained perception.
2. Current studies primarily focus on moderate-length trajectories (e.g., "We set the maximum number of steps to 30"). However, the scalability of SGV to extremely long sequences remains unclear. Such sequences are common in computer usage scenarios, and the context window pressure under extremely long sequences may cause biases to reemerge.
3. The ablation on SGV’s prompt design (Appendix B.6) is limited. more tests on prior generation diversity (e.g., multiple priors vs. single) would strengthen claims about SGV’s mechanism.

**Questions:**

Address the weaknesses.

---

> ### Author Response · Authors · 2025-11-27
> **Author Response - Part 1**
>
> Thank you for your thoughtful and constructive feedback. We are grateful that the reviewer found our framing of agreement bias to be original and important, and we appreciate the positive assessment of the breadth of our experimental evaluation, the clarity of our methodological presentation, and the significance of our contributions for improving the reliability of MLLM-based verifiers and downstream agent systems. We are also glad that the reviewer found SGV intuitive to follow and recognized the generality of our empirical findings across diverse tasks, models, and environments.
>
>
> Below, we respond to specific questions and concerns the reviewer raises. All feedback has been incorporated into our revised manuscript. We hope that the additional results and clarifications help address the reviewer's concerns and enable a re-evaluation of our work. We remain open to providing any further clarifications or analyses the reviewer may find helpful.
>
> ---
>
>
> > W1: SGV does not address underlying vision-language flaws (e.g., Figure 7’s counting error), and the paper lacks discussion on combining SGV with specialist models for fine-grained perception.
>
> Thank you for raising this discussion. As noted in our limitations section, some remaining failures indeed seem to stem from deficiencies in the base models' integration of visual perception and language. We agree that verification performance could be further improved by combining SGV with specialist models--for example, models dedicated to counting, or object detection and fine-grained image annotations--which can yield gains complementary and orthogonal to those provided by SGV.
>
> More generally, as mentioned in our conclusion, we are optimistic about modular approaches [1] that can leverage MLLMs flexibility while introducing orthogonal sources of knowledge and capabilities. This includes symbolic methods, as well as generative or specialist models whose output distributions tend to be less correlated with the base MLLMs and can therefore provide complementary signals (which, as suggested by W2 ablations on prior diversity, can help performance). However, we were not able to produce reliable experiments in the current timeframe, and need to leave this exploration to future work. We added more discussions in this regard to our revised manuscript.
>
> [1] Subbarao Kambhampati et al. LLMs Can't Plan, But Can Help Planning in LLM-Modulo Frameworks, 2024.

---

> ### Author Response · Authors · 2025-11-27
> **Autor Response - Part 2**
>
> ### W2: scalability of SGV to long sequences
>
> Thank you for raising this interesting discussion!
> We wish first to clarify that we use 30 steps in VisualWebArena and 50 steps in OSWorld because these are commonly adopted in the benchmarks.
>
> In the table below, we report MLLM verification performance stratified by trajectory length in OSWorld and a new 60-step setting in VisualWebArena. To increase the number of samples per bucket and provide more statistical significance, we average results across three comparable MLLMs (GPT-4.1, Qwen3-VL, and Gemini 2.5).
>
> As observed, SGV reduces bias, skewness and improves TNR and accuracy across lengths. Additionally, we note that MLLM verifier performance tends to be **higher** on longer trajectories. This pattern arises because longer trajectories often indicate that agents have deviated from the task objective and contain more opportunities for undesired actions--e.g, UI-Tars deleting files in OSWorld or looping over ineffective sequences--which make failures easier to detect. This reinforces a point raised in Section 3.2 of our manuscript: to properly assess verifier performance--and reveal limitations like agreement bias that could otherwise be obscured--it is crucial to consider relatively strong agents (as our Agents in VisualWebArena).
>
> Finally, although we do not observe degradation in performance with longer trajectories, we acknowledge that the combination of stronger agents *and* longer trajectories may introduce additional challenges--whether agreement bias or yet unidentified limitations. However, analyzing this dimension is difficult with the current state of agents and benchmarks: disentangling quality from trajectory length without introducing arbitrariness is non-trivial, unless tasks inherently require long trajectories and agents can produce them reliably. In the additional results below, we partially address this by varying the strength of our VisualWebArena agents while keeping a fixed trajectory length, finding that (i) performance degrades when verifiers are weaker than agents, specifically due to an increase in agreement bias and (ii) SGV remains effective in all configurations.
>
> We added this discussion to our revised manuscript, and hope it alleviates R2 concerns. We remain open to further experiments R2 may find helpful.
>
> **Table R3.1 (a) - MLLM Verification Performance by Trajectory Length - VisualWebArena**
> ||Metric|All|0–5|6–19|≥20|
> |--|-|-|-|-|-|
> ||% Success|47|57|36|17|
> ||N samples|2,730|1,587|1,218|198|
> ||% Traj.|100|58|45|7|
> |No-SGV|ACC|67|68|61|65|
> ||TPR|90|91|88|64|
> ||TNR|48|40|46|65|
> ||bias|24|22|30|22|
> ||dSkew|19|17|26|16|
> |SGV|ACC|74|73|72|82|
> ||TPR|82|84|81|66|
> ||TNR|67|60|67|85|
> ||Bias|10|9|14|6|
> ||dSkew|5|4|8|2|
> |Δ (SGV − No-SGV)|ACC|+7|+5|+11|+17|
> ||Bias|−14|−13|−16|−16|
> ||dSkew|−14|−13|−18|−14|
>
>
> **Table R3.1 (b) - MLLM Verification Performance by Trajectory Length - OSWorld**
> ||Metric|All|0–10|11–20|21–30|≥30|
> |--|-|--|--|-|-|-|
> ||% Success|22|54|20|16|2|
> ||n Samples|1044|300|198|111|435|
> ||% Traj|100|29|19|11|42|
> |No-SGV|ACC|76|71|59|67|89|
> ||TPR|96|97|97|89|67|
> ||TNR|70|41|49|63|89|
> ||bias|22|26|40|29|10|
> ||dSkew|21|24|39|27|9|
> |SGV|ACC|83|77|75|81|94|
> ||TPR|91|90|97|88|67|
> ||TNR|81|61|70|80|94|
> ||bias|12|13|24|14|5|
> ||dSkew|10|8|23|14|5|
> |Δ (SGV − No-SGV)|ACC|+7|+6|+16|+14|+6|
> ||Bias|−10|−13|−13|−15|−5|
> ||dSkew|−11|−16|−13|−13|−4|
>
> ---
>
> **Additional Discussion: Verification Difficulty**
>
> R3 implicitly raises a more fundamental question concerning how verification performance depends on the difficulty of the verification task relative to model capability.
> To examine this dimension, we vary the strength of models to build the agent and verifiers under constant environment (VisualWebArena) and trajectory budget (30 steps).
>
> In summary, results show that:
> - Agreement bias occurs both when the verifier is built with stronger or weaker models than agents, and when they are built from models of distinct families and sizes.
> - Verification performance is worse when the verifier is weaker than the generator (e.g., Gemini 2.0 verifies Gemini 2.5-based agent), particularly due to an **increase** in agreement bias (**note TPRs actually increase**).
> - SGV remains effective in all configurations.
>
> **Table R3.2 - Verification Difficulty and Cross-Model Variations**
> |Verifier/Agent|G 2.5 / G 2.5|G 2.5 / G-2.0|G 2.5 / GPT4o|G 2.0 / G 2.5|G-2.0 / GPT4o|
> |--|-|-|-|-|-|
> |Acc|(65, 76)|(70, 78)|(68, 77)|(56, 65)|(55, 65)|
> |TPR|(90, 84)|(84, 77)|(80, 76)|(93, 90)|(85, 74)|
> |TNR|(47, 71)|(63, 79)|(63, 77)|(28, 42)|(42, 61)|
> |Bias|(26, 11)|(20, 7)|(20, 8)|(38, 28)|(35, 19)|
> |dSkew|(21, 5)|(14, 2)|(13, 3)|(35, 23)|(30, 12)|
> |**Success Rate**|47|33|31|47|31|
> |Δ Acc|11|8|8|9|9|
> |Δ Bias|-15|-13|-12|-10|-16|
> |Δ dSkew|-16|-12|-11|-12|-18|
>
> *Note: G 2.x = Gemini-Flash 2.x

---

> ### Author Response · Authors · 2025-11-27
> **Author Response - Part 3**
>
> ### W3: The ablation on SGV's prompt design is limited. More tests on prior generation diversity (e.g., multiple priors vs. single) would strengthen claims about SGV’s mechanism.
>
> Thank you for this suggestion!
>
> First of all, we would like to highlight results in Appendix F, Figure 3, further discussed in the comments addressing R2 concerns. These results show that SGV yields more balanced MLLM verification and improves performance across a diverse set of over 28 prompt/evaluation templates. For convenience, the table of average performance across templates is available at this [link](https://anonymous.4open.science/r/abias-sgv-iclr-26-3DD2/R2..1.md) and the per‑template score distributions at this [link](https://anonymous.4open.science/r/abias-sgv-iclr-26-3DD2/R2..2.md).
>
> As for the effect of prior diversity, in the table below we report results where the verifier is given (1) a single prior generated from the same model with a high temperature, (2) a single prior generated from models of distinct families, (3) a concatenation of K=3 priors from the same model and (4) K=3 priors from models of distinct families. We also include the baseline SGV and No SGV results for ease of comparison.
>
> In summary:
> - We observe performance can be improved when multiple priors come from models of distinct families, indicating that diversity in the source of priors can be beneficial-though in this case, at a bit higher computational cost.
> - However, we do not observe substantial gains by increasing temperature or the number of generations within the same model, suggesting that after the first step generation, the gains are very much exhausted (more on this below).
>
> **Table R3.3 - Ablations on multiple priors and prior diversity**
> ||||(1)|(2)|(3)|(4)|(5)|
> |-|-|-|-|-|-|-|-|
> |First Step|-|Gemini 2.5, temp=0 (G)|Gemini 2.5, temp=2|Qwen3-VL (Q)|GPT-5 (GP)|Gemini 2.5, k=3|(G+Q+GP), k=3|
> |Verify|Gemini 2.5 (no SGV baseline)|Gemini 2.5 (SGV baseline)|Gemini 2.5|Gemini 2.5|Gemini 2.5|Gemini 2.5|Gemini 2.5|
> |Acc|65|76|77|76|76|78|79|
> |TPR|90|84|84|84|84|87|82|
> |TNR|47|71|71|70|70|70|76|
> |Bias|26|11|11|11|11|12|6|
> |dSkew|21|5|5|5|5|7|2|
>
> ----
>
> **Additional Results**
>
> To add more clarity on the effectiveness and robustness of the SGV mechanism, below we provide additional ablations to its prompt design and prior generation. Specifically, in the table below, we compare MLLM verification performance in the following settings:
> - (1): No prior generation, but provide the verifier a Web Search tool to ground the verification [1]
> - (2): SGV, where the prior generation is supported by a web search tool;
> - (3) SGV, where prior generation is produced by a weaker variant of the same model;
> - (4) SGV, where we inject into the generated priors a random noise la Rulin Shao [2].
> - (5): No SGV for a thinking variant of Gemini, but provide the verifier to a Web Search tool to ground the verification
> - (6): SGV with thinking enabled, where priors are generated with thinking disabled.
>
> In summary, results suggest that:
> - (i) In contrast to SGV, providing grounding tools such as web search to the verifier alone does not yield meaningful mitigation of agreement bias;
> - (ii) Extracting knowledge from the own model seems sufficient, with the addition of web search not providing meaningful improvements;
> - (iii) Weaker and non-thinking models can produce priors that are sufficiently informative for effective verification for stronger or thinking models;
> - (iv) The similar performance of SGV with and without Web Search, the near-identical performance of Gemini-Thinking when priors are generated with or without thinking (6 vs. SGV baseline), and the comparable performance when thinking is fully disabled versus fully enabled all suggest that SGV’s first-step mechanism accounts for the vast majority of the gains, and largely exhausts them.
> - (v) SGV remains effective even when the first-step generation is noisy, indicating that the method is robust to some degree of noise in prior generation;
>
>
> **Table R3.4 - Further Ablations to Prior Generation**
>
> ||No SGV Baseline|(1)|SGV Baseline|(2)|(3)|(4)|
> |-|-|-|-|-|-|-|
> |First Step|-|Web|Gemini 2.5|Gemini 2.5 + Web|Gemini 2.5-Lite|Gemini 2.5 + Random Noise|
> |Verify|Gemini 2.5 (no SGV baseline)|Gemini 2.5|Gemini 2.5|Gemini 2.5|Gemini 2.5|Gemini 2.5|
> |Acc|65|64|76|74|74|73|
> |TPR|90|92|84|84|82|81|
> |TNR|47|44|71|65|69|68|
> |Bias|26|29|11|17|11|14|
> |dSkew|21|26|5|10|5|8|
>
> |-|No SGV Baseline|(5)|SGV Baseline|(6)|
> |-|--|-|-|-|
> |First step|-|Web|Gemini 2.5-Thinking|Gemini 2.5|
> |Verify|Gemini 2.5-Thinking|Gemini 2.5-Thinking|Gemini 2.5-Thinking|Gemini 2.5-Thinking|
> |Acc|69|66|77|76|
> |TPR|90|73|86|86|
> |TNR|55|61|70|68|
> |Bias|23|10|12|13|
> |dSkew|18|4|6|7|
>
> ---
>
> [1] Gout et al., CRITIC: Large Language Models Can Self-Correct with Tool-Interactive Critiquing. ICLR 2024.
>
> [2] Rulin Shao et al., Spurious Rewards: Rethinking Training Signals in RLVR.

---

### Official Review · Reviewer_dsJd · 2025-11-01

**Soundness:** 3
**Presentation:** 3
**Contribution:** 3
**Rating:** 8
**Confidence:** 3

**Summary:**

This paper identifies “agreement bias,” i.e., a preference of verifiers to return positive results, as a key issue for agent tasks. They then propose a simple method in which a verifier first proposes a task-specific rubric and scores trajectories according to this rubric in order to mitigate agreement bias. Finally, the paper shows that these improved verifiers can be used to improve model performance on VisualWebArena, OSWorld, and the robotic manipulation task robomimic.

**Strengths:**

1. This paper proposes a straightforward extension of the idea in Pan, et al. (2024) which shows that automatic evaluators can be used to improve the performance of web navigation and device control agents at training or inference time

2. The paper makes a compelling case that models exhibit agreement bias when evaluating agent trajectories, i.e., a bias toward positive labels (Table 1a). The paper also clearly shows that its method leads to a reduction in agreement bias (Table 1b), and that this method works across a wide range of verifier models

3. Most importantly, the paper shows that stronger verifiers are more useful, e.g., at improving agents at inference time using methods like Reflexion (Figure 2)

**Weaknesses:**

1. It would be nice to see some comparison of the proposed method with simpler strategies, e.g., different prompts to the verifier model, or prompting models to generate confidences and applying Platt scaling

2. The paper could benefit from an additional round of proofreading. For example:

   - Line 101: missing a period
   - Line 221: “Table Table 7” -> “Table 7”
   - Line 263: broken reference (“??”)
   - Line 1234: “AgentRewarBench” -> “AgentRewardBench”
   - \citet should be replaced with \citep in many places

**Questions:**

N/A

---

> ### Author Response · Authors · 2025-11-26
> **Author Response - Part 1**
>
> Thank you for the positive evaluation of our work! We are encouraged that the reviewer found our identification of agreement bias to be compelling and comprehensive, and that our method shows a clear reduction of agreement bias across a wide range of models. We also appreciate the reviewer's recognition of our results connecting verifier strength to performance in downstream applications--a central message of our work.
>
> Below, we provide answers to each of the questions and concerns raised by the reviewer. All feedback was incorporated in our revised manuscript.
>
> ----
>
> > W1: It would be nice to see some comparison of the proposed method with simpler strategies, e.g., **(A)** different prompts to the verifier model, or **(B)** prompting models to generate confidences and applying Platt scaling
>
>
> First of all, we would like to thank the reviewer for this suggestion. The comments from R2 and R4 both point to our results on the distribution of MLLM responses, where we show that agreement bias manifests as MLLM response distributions skewed toward positive labels. This is reflected in our metrics (high and positive bias and dSkew, low TNR), and is more explicitly illustrated in Figure 3 (Appendix F), where we evaluate MLLM verifiers across 28 prompt/evaluation templates. Thanks to this discussion, we recognize that these results could have been emphasized and leveraged more. Below, we address this point directly, and we have revised the manuscript accordingly to integrate the conclusions that follow.
>
> > W1-A: different prompts to the verifier model
>
> In section F we evaluate MLLM verifiers across 28 evaluation/prompt templates, finding that (i) MLLMs tend to concentrate their evaluations at the high end of the scale, largely independent of how the evaluation is framed, and (ii) SGV leads to a more balanced and human-aligned distribution across templates. We elaborate below.
>
>
> In the following, we clarify and expand on our results explored in section F, where we show MLLM verification performance across 28 evaluation/prompt templates, including: likert and numeric-based scales (including confidence generation), coupled with interventions explored in the literature to mitigate biases in LLM evaluators, such as criteria order shuffling and reversal, and syntactic structure changes [1], [2]. In all cases, the verifier prompt is the same as in section B.1, except for the section asking for the evaluation, which is substituted by the corresponding template. All templates are included in our code submission and at this [link](https://anonymous.4open.science/r/abias-sgv-iclr-26-3DD2/evaluation_templates_exampels.py).
>
> The table below shows the average performance across all template variations. The figures in this [link](https://anonymous.4open.science/r/abias-sgv-iclr-26-3DD2/Response-Distribution.png) show the distributions of MLLM responses across all templates.
>
>
> In summary, results show that:
> - **MLLMs tend to concentrate their evaluations at the high end of the scale, largely independent of interventions and how the evaluation is framed.**
> -  MLLMs rarely express uncertainty, even when the evaluation template explicitly includes an “uncertain” option, and reserves the highest score for high-confidence success.
> -  SGV leads to a more balanced distribution of scores for all templates.
> -  Binary scales (as adopted in Pan, et al. (2024)) can exacerbate bias toward favorable evaluations. A ternary scale (as in our baseline) seems sufficient to mitigate this effect, with no meaningful differences observed for more granular scales.
> -  Both Likert- and numeric-based templates can perform well, although we find the former to be slightly more effective.
>
> **Table R2.1 - Average MLLM Verification Performance Across 28 prompt templates**
> |Template Group||ACC|TPR|TNR|Bias|dSkew|
> |-|-|-|-|-|-|-|
> |**All**|no SGV|67|85|53|21|17|
> ||SGV|70|80|63|13|9|
> |**Numeric-based**|no SGV|63|92|42|30|26|
> ||SGV|73|74|71|6|2|
> |**Likert-based**|no SGV|65|90|47|27|22|
> ||SGV|74|76|72|7|2|
>
>
> ---
> [1] Chen et al. Unbiased evaluation of large language models from a causal perspective, 2025.
>
> [2] Ye et al. Justice or prejudice? quantifying biases in llm-as-a-judge, 2024.

---

> ### Author Response · Authors · 2025-11-26
> **Author Response - Part 2**
>
> > W1-B generates confidence and applies Platt scaling
>
> We do not observe meaningful gains from applying Platt scaling to the generated confidences, particularly in comparison to the gains provided by SGV. We elaborate below.
>
> First, we observed that letting the MLLM freely generate a confidence ((a) below) tends to provide inferior results compared to providing some grounding on the confidence scale. Therefore, for confidence generation, we adopted the latter variation, shown below (b).
>
> ```
> # (a) Free-form confidence generation examples, suboptimal:
> Provide a number between 0 and 1 reflecting your confidence that the assistant's work is correct for the task.
>
> Provide a number between 0 and 1 reflecting the likelihood that the assistant's work is correct for the task.
>
> # (b) Adopted template (obs.: 10 is mapped to 0-1 during calibration)
> Provide a single score, rating the assistant's work strictly on a scale of 1 to 10 based on the confidence of its correctness for the task.
>
> The score can be any scalar between 1 and 10, where:
>
> - A **low score** means you are very confident that the assistant's work is completely incorrect for the task.
>
> - A **high score** means you are very confident that the assistant's work is completely correct for the task.
>
> - A **mid-range score** means you are uncertain about the correctness of the assistant's work; it could be right or wrong.
> ```
>
> We focus this analysis on VisualWebArena, where our agents produce a roughly balanced ratio of success and failures.
> To test the Platt calibrator, we utilize our 1/3-sized representative subset (section G.1) that provides the same distribution as the full benchmark, and we train it on the remainder of the data (~605 datapoints). In both cases, the data is roughly balanced, with 47% success and 53% failed trajectories. In all cases, we choose the optimal threshold to define a trajectory as a success or failure by maximizing Youden's J statistic on the training set. We also report the Area Under the Curve (AUC) statistic for completeness. In the first table below, we compare our main metrics after Platt scaling and after SGV. In the second table, we report the distribution of MLLM generations before and after Platt scaling.
>
> In summary:
> - We do not observe meaningful gains from applying Platt scaling to the generated confidences.
> - This is explained by the skew in MLLM score distributions discussed above, and shown in Table R2.3: MLLMs overwhelmingly assign labels at the upper end of the scale, leaving Platt scaling with no granularity to exploit and causing large clusters of identical scores to map to the same calibrated probability. The same issue arises with other methods, such as an isotonic transform.
> - In line with W1(a) results, SGV leads to a substantially more balanced distribution of scores, with corresponding gains in evaluation metrics.
> - We highlight these findings do not constitute an artifact of prompt design/template: as SGV results show, **it is** possible to obtain more balanced distributions under this evaluation design.
>
> Additionally, we point R2 to additional discussion to address R4's inquiry on potential sampling or training strategies to mitigate agreement bias, where we conduct a proof of concept experiment exploring calibration via Platt scaling on implicit probabilities derived from MLLMs in sampling generation. We included the conclusions in the next comment for convenience.
>
>
> **Table R2.2: Platt Scaling vs SGV, template = Generate Confidence**
> |Metric|Before Calibration|After Platt|SGV|
> |-|-|-|-|
> |ACC|63|65|72|
> |TPR|92|92|82|
> |TNR|37|40|64|
> |Bias|29|28|10|
> |dSkew|25|23|4|
> |AUC|65|65|74|
>
> **Table R2.3: Platt Scaling vs SGV, template = Generate Confidence**
> | Score bucket | Before Calibration | Platt | Isotonic | SGV |
> | ------------ | -----------------: | ----: | -------: | --: |
> | [0–0.1)      |                10% |   11% |      16% |  6% |
> | [0.1–0.2)    |                 2% |   10% |       0% |  8% |
> | [0.2–0.3)    |                 5% |    3% |       4% | 13% |
> | [0.3–0.4)    |                 4% |    1% |       4% |  5% |
> | [0.4–0.5)    |                 2% |    1% |       1% |  1% |
> | [0.5–0.6)    |                 2% |   74% |      74% |  3% |
> | [0.6–0.7)    |                 1% |    0% |       0% |  4% |
> | [0.7–0.8)    |                 1% |    0% |       0% |  4% |
> | [0.8–0.9)    |                 0% |    0% |       0% |  3% |
> | [0.9–1.0]    |                74% |    0% |       0% | 55% |

---

> ### Author Response · Authors · 2025-11-26
> **Additional Analysis - Platt Scaling on Sampling-derived Probabilities**
>
> Inspired by R2 suggestions, the above results, and to R4's inquiry on potential sampling or training strategies to mitigate agreement bias in MLLM verifiers, we conducted a proof of concept experiment exploring calibration via Platt scaling on implicit probabilities derived from MLLMs during sampling generation.
>
> Specifically, for each of the 910 trajectories in VisualWebArena, we sample 8 completions and estimate empirical probabilities by measuring the proportion of times the MLLM labels a trajectory as a success or failure under the best-performing template across the 28 tested templates---the ternary Likert scale used in our baselines.
>
> Before discussing results, we note that this is a *proof of concept* experiment under relatively *ideal* and *expensive* conditions. Particularly:
> - The calibrator is trained and tested on balanced sets with roughly equal numbers of success and failures.
> - It assumes access to 605 datapoints with high-quality labels provided by our improved VisualWebArena oracles (lines 190)---a limitation in real-world settings that motivates the need for MLLM verifiers in the first place.
> - Training and test domains match; i.e., we do not consider out-of-distribution generalization.
> - To estimate probabilities, we sample a relatively large number of completions (8 per trajectory, totaling over 7,200 generations).
> - The thresholds used to separate classes—which, as we show below, can be a sensible choice—are optimized under these same ideal conditions.
> - At test time, we still generate 8 completions per new trajectory for probability estimation, and apply the calibrator to them.
>
>
> Results are shown in the table below. In summary:
> - Vanilla majority voting fails to deliver improvements for not taking into account the skew in MLLM label distributions toward positive evaluations.
> - Leveraging this fact to apply a calibration to MLLM implicit probabilities leads to more balanced distributions and improvements in evaluation metrics.
> - **Most notably,** SGV lead to distributions superior than the calibrated ones obtained in ideal scenarios through a **single** generation.
> - Importantly, this arises organically by leveraging the model's own sampling mechanisms, without the need for ground-truth labels, additional training, or calibration steps. Moreover, these gains naturally extend to other domains, as demonstrated by SGV's improvements across benchmarks.
> - SGV also avoids complicated hyperparameter tuning such as calibration thresholds. As shown in columns 'p15' and '0.5', setting this threshold to either the 15th percentile (bootstrap-estimated) or naïvely to 0.5 mostly eliminate the gains from Platt scaling.
>
>
> **Table R4.1: Platt Scaling vs SGV, template = Ternary Likert, Sampling Generation**
> |Metric|No SGV, temp=0|No SGV, majority|**Platt**|Platt, p15|Platt, 0.5|**SGV (temp=0)**|SGV, Majority|SGV, Platt|
> |-|-|-|-|-|-|-|-|-|
> |ACC|65|65|**72**|69|71|**76**|75|78|
> |TPR|92|93|**77**|85|86|**84**|85|79|
> |TNR|41|40|**68**|56|57|**71**|67|78|
> |Bias|27|28|**6**|20|16|**11**|11|1|
> |dSkew|23|24|**2**|15|9|**5**|5|0|
> |AUC|67|66|**73**|75|75|**76**|76|83|
>
> We hope these results reinforce the effectiveness of SGV in mitigating a bias that stems from intrinsic MLLM response distributions. We believe these results suggest that there is potential for both training-time and test-time approaches that explicitly account for the asymmetries introduced by agreement bias---e.g., loss functions that penalize such imbalance, or adjust sampling mechanisms for methods that rely on sampling (e.g., GRPO). Since SGV itself induces substantially more balanced output distributions, incorporating SGV into larger pipelines may also be fruitful---e.g., as a replacement for oracle labels in methods like the one above.
>
> We hope that our identification of agreement bias, our demonstration of the risks it poses for applications relying on MLLM verifiers, our evidence on the benefits of mitigating this bias to downstream applications, and the insights provided by our work collectively help motivate further research in this direction.
>
> We thank the R2 and R4 for raising this discussion and prompting us to better leverage these results. We have included this discussion in our revised version and leave further exploration to future work.

---

### Official Review · Reviewer_187k · 2025-11-02

**Soundness:** 2
**Presentation:** 3
**Contribution:** 2
**Rating:** 4
**Confidence:** 3

**Summary:**

This paper identifies "agreement bias" as a critical failure mode in Multimodal Large Language Models (MLLMs) when they are used as verifiers for agent trajectories. The authors find that MLLMs tend to favorably evaluate flawed agent behavior, even generating rationalizations for it, despite possessing strong, human-aligned priors on correct task execution. They attribute this to a retrieval bottleneck. To address this, the paper introduces Self-Grounded Verification (SGV), a two-step prompting method. SGV first elicits the MLLM's broad priors about task completion independent of the agent's trajectory, and then conditions the MLLM on these self-generated priors to evaluate the candidate trajectory. Experiments across web navigation, computer use, and robotics benchmarks show that SGV significantly improves failure detection and accuracy, boosting the performance of agents in online supervision settings.

**Strengths:**

1. This work identifies a significant and practical problem, the "agreement bias" of MLLM verifiers. This is an important contribution as these verifiers are increasingly proposed for data filtering, self-refinement, and online agent guidance.
2. The paper demonstrates strong empirical results, particularly in improving the True Negative Rate (failure detection). This is a crucial metric that is more informative than overall accuracy for this problem, as the primary goal of a verifier is to catch flawed behavior.
3. The method is validated across a diverse and challenging set of multimodal environments (VisualWebArena, OSWorld, robomimic) and application settings (offline evaluation, self-refinement, and online supervision), which strengthens the generality of the claims.

**Weaknesses:**

1. The core mechanism of SGV—generating "broad priors" in Step 1 independent of the agent's trajectory —may be a significant flaw. By being ungrounded from the specific context of the agent's current state, these priors may be overly generic or common sense hallucinations that are irrelevant to the task at hand. This could lead the verifier to be "overly strict," unfairly penalizing valid or creative solutions that deviate from the generic script, a failure mode the authors acknowledge. The paper lacks a sufficient analysis of when these broad priors are helpful versus when they are harmful.
2. The proposed solution is a heuristic-driven prompting technique. While it shows good results, it is not clear how the method will scale or interact with future model development. The paper claims the issue is a "retrieval bottleneck", but it is equally plausible that agreement bias is an artifact of current alignment techniques or context-window management. The paper does not provide evidence to suggest whether SGV is a durable solution or a temporary patch for a flaw that might be solved more fundamentally by future models, rendering the heuristic obsolete.
3. The evaluation of the online supervision setting seems arbitrary in its implementation. For instance, in OSWorld, the verifier is called "every 5 steps". There is no justification for this hyperparameter, and it glosses over the significant trade-off between verification frequency (and thus, token/compute cost) and the ability to catch errors in real-time.

**Questions:**

How does the performance benefit of SGV change with model scale and capability? The paper shows it helps both weaker and stronger "reasoning" models, but does the relative gain (SGV vs. baseline) shrink as models become more capable? This would provide insight into whether SGV is fixing a fundamental reasoning flaw or a specific weakness of current models.

---

> ### Author Response · Authors · 2025-11-26
> **Author Response - Part 1**
>
> Thank you for the thoughtful and constructive feedback. We are glad the reviewer found our identification of agreement bias of MLLM verifiers to be significant and practically important, especially given its consequences for downstream applications. We are also grateful for the positive assessment of our empirical results--particularly the improvements in TNR, which we agree is a crucial metric for verification--and encouraged that they found our evaluation across diverse environments and applications to strengthen the generality of our findings.
>
> Below, we respond to the reviewer's specific questions. We hope that the additional results and clarifications help address the reviewer's concerns and enable a re-evaluation of our work. All feedback has been incorporated into our manuscript, and we remain open to providing any further clarifications or analyses.
>
> ---
> > Q1: Does the relative gain (SGV vs. baseline) shrink as models become more capable?
>
> > W2:  The paper does not provide evidence to suggest whether SGV is a durable solution or a temporary patch for a flaw that might be solved more fundamentally by future models, rendering the heuristic obsolete.
>
> First, we wish to clarify that the results in the paper across model types and sizes suggest that this is not the case, and new experiments below further support that. Specifically, Tables 1 and 9 report gains across models of varying capability, including reasoning and non-reasoning variants.
>
> Nonetheless, we very much appreciate the suggestion and expanded our results to a broader set of models, which we hope further clarifies SGV's benefits and the pervasiveness of agreement bias. The table below reports the average performance of MLLM verifiers on 1,300 trajectories from OSWorld and VisualWebArena, including all evaluation metrics to provide a comprehensive characterization of SGV's relative gains
>
> As observed:
> - Agreement bias---the tendency to overly validate agent behavior---is pervasive across models, manifesting as distributions skewed toward positive evaluations (high bias and skewness) and, notably, a low TNR, with some models performing below chance at flagging failures.
> - Across all models, SGV improves TNR, Accuracy, and produces outputs more aligned with oracle judgments, as evidenced by the consistent reduction in bias and skewness.
> - **Regarding relative gains, we do not observe a uniform trend as a function of model capability.** For instance, within both the Gemini and Qwen families, SGV gains are comparable to---or slightly greater than--those observed for smaller or older variants. A similar pattern holds within the GPT family, with the exception of GPT-4.1 and GPT-4.1 Mini, where the latter exhibits the largest gains across all models.
>
> Additionally, we would like to note that:
> - Both agreement bias identification and SGV gains are based on **strong** baselines: our no-SGV verifier is already state of the art in the AgentRewardBench benchmark (Table 8 and comment G2 to all reviewers)
> - They remain stable across multiple ablations of the verifier design, as shown in Figure 3 of the manuscript and further discussed in our response to R3.
>
> **Table R1.1: MLLM Verifier Performance on Trajectories in OSWorld and VisualWebArena.**
> **\*(T) = Thinking enabled with the maximum thinking budget**
>  **\*\* Success rates are 47\% in VisualWebArena and 23\% in OSWorld.**
> | |No SGV|||||SGV|||||$\Delta$|||
> |-|-|-|-|-|-|-|-|-|-|-|-|-|-|
> |Model|Acc|TPR|TNR|Bias|dSkew|Acc|TPR|TNR|Bias|dSkew|$\Delta$ Acc|$\Delta$ Bias|$\Delta$ dSkew|
> |Gemini 2.0|61|96|42|36|34|69|94|55|27|23|9|-9|-11|
> |Gemini 2.5 Lite|55|96|34|41|39|65|90|51|29|25|9|-11|-13|
> |Gemini 2.5 Flash|68|94|55|27|24|80|88|76|12|8|12|-15|-16|
> |Gemini-2.5-Flash (T)|74|92|64|21|18|82|89|78|10|6|8|-11|-13|
> |Qwen3-32b|69|92|57|25|21|76|88|71|14|11|7|-11|-10|
> |Qwen3-235b-a22b|65|93|51|28|25|76|89|70|15|12|10|-14|-13|
> |Qwen3-235b-a22b (T)|66|92|53|27|24|77|91|71|15|12|11|-12|-12|
> |GPT-4.1 Mini|60|96|40|37|35|74|92|65|17|14|14|-20|-21|
> |GPT-4.1|74|90|64|19|15|81|87|78|10|6|7|-9|-9|
> |GPT-o1 (T)|70|80|62|22|16|78|83|73|13|7|7|-9|-8|
> |GPT-o4 (T)|78|88|71|11|7|84|86|82|6|2|6|-6|-5|
> |GPT-5-Nano (T)|72|84|65|16|11|76|82|73|10|6|4|-6|-5|
> |GPT-5 (T)|81|86|78|8|4|86|85|87|2|1|5|-6|-3|
> |Llama-4-Maverick-17B-128E|60|92|44|33|29|65|89|54|25|22|5|-7|-8|
>
> We hope these results help to alleviate concerns, especially given that they are based on the most recent and strongest models. That said, we prefer to be conservative about long-term claims. In our field, it is common for methods to be superseded by subsequent techniques that build on their underlying insights. Nevertheless, such works often play an important role in enabling scientific progress by (i) clearly articulating and quantifying important problems, and (ii) providing knowledge that is practically useful in their time and that helps inform future approaches. We hope that our contributions fulfill (i) and (ii) and can be viewed in this spirit.

---

> ### Author Response · Authors · 2025-11-26
> **Author Response - Part 2**
>
> > W1: The paper lacks a sufficient analysis of when these broad priors are helpful versus when they are harmful.
>
> Thank you for pointing this out. Although our manuscript has qualitative and quantitative parts dedicated to this discussion, we agree that further clarification and breakdown of results would be beneficial.
>
> **Clarification on 'Strictness' and Verifier Design**
>
> First, we note that, during verification, the MLLM **is** aware of the agent's state and can choose how much to rely on the Step-1 priors versus the observed trajectory. SGV does not force the model to ignore context; rather, it provides an additional signal that encourages more balanced verification (Fig. 3, R2, R4 discussion).
>
> As acknowledged by us, SGV can sometimes produce stricter verification (lower TPR). However, we highlight two observations from Section 4:
> - **This is most pronounced when oracles are lenient** -- for instance, labeling a trajectory as successful despite missing steps such as filtering/sorting products (Fig. X). In such cases, SGV tends to disagree with the oracle, whereas the baseline MLLM verifier, influenced by agreement bias, tends to agree, leading to a relative drop in TPR.
> - **Verification involves a natural strictness-leniency trade-off**, where leniency can allow brittle behavior to pass unnoticed, while strictness may reject valid solutions. This trade-off is intrinsic to verification, and no metric can perfectly capture its optimal balance.
>
> **Clarification on Quantitative Metrics and Qualitative Analysis**
>
> Given this leniency-strictness trade-off, we believe the most practical evaluation metric is the net impact of a verifier on downstream applications. As shown in our main results--and highlighted by R2, R3--SGV **consistently improves** task completion rates in both self-refinement and online supervision, whereas the baseline verifier does not. As discussed in sections 4 and 5, this is explained by:
> - SGV identifies suboptimal behavior, providing corrective feedback that enables the agent to improve,
> - While SGV can be stricter, interventions are largely non-disruptive (more below),
> - Influenced by agreement bias, the baseline verifier fails to provide feedback exactly when it is most needed---when agent behavior is flawed and requires improvement.
>
> All these cases are illustrated qualitatively in the corresponding passages. For instance, Fig. 9 shows that SGV rejects a greedy strategy that technically satisfies the benchmark and prompts the agent to search for and confirm user-requested attributes, ultimately leading to task completion via a more generalizable strategy.
>
> Finally, more discussion is provided in our **qualitative analysis Section C.2**, which includes a categorization of cases as "strict verification, harmful", "strict verification, harmless", "overly lenient verification", and so forth.
>
> **Additional Results and Analysis**
>
> To further elucidate this point, in the table below (better visualized via this [Sankey Diagram](https://anonymous.4open.science/r/abias-sgv-iclr-26-3DD2/SankeyDiagram.png), also included in our revised manuscript), we report VisualWebArena success rates before and after verifier interventions, binned by whether the verifier classifies the trajectory as a failure when it is not (False Negative, FN), a success when it is not (False Positive, FP), and so forth. The notation (B→A) indicates the status of the tasks (success=1, failure=0) before and after the verifier intervention. For example, (TN, 0→1) indicates the percentage of tasks that were initial failures (0), were identified as failures by the verifier (TN), and became successful (1) after its intervention. The entries of most interest are highlighted in bold.
>
> In summary:
> - (SGV, FN, 1->1) shows that the majority of "false-negative" evaluations by SGV (6.6% of 7.4%) **still lead to successful task completion**.
> - Importantly, 9.8% of initially unsuccessful tasks become successful after SGV's corrective feedback (SGV, TN, 0→1).
> - In contrast, (No SGV, FP, 0→0) shows that in the baseline scenario, 29% of unsuccessful tasks remain unsuccessful because, influenced by agreement bias, the verifier **fails to deliver corrective feedback** exactly when agent behavior is flawed.
> - As a result, SGV leads to an overall increase of about 10% in success rates, while the baseline verifier fails to provide meaningful gains.
>
> We hope this discussion alleviates R1 concerns, clarifies the complementarity of downstream metrics, and highlights our contributions about the risks imposed by agreement bias, as well as the benefits of its mitigation.
> We remain open to further analysis R1 deems necessary.
>
> **Table R1.2: Breakdown of Performance Before and After Verifier Interventions**
> |Case|SGV 0→0|SGV 0→1|SGV 1→0|SGV 1→1|No-SGV 0→0|No-SGV 0→1|No-SGV 1→0|No-SGV 1→1|
> |-|-|-|-|-|-|-|-|-|
> |FN|-|-|0.8|**6.6**|-|-|0.5|4.1|
> |TN|37.7|**9.8**|-|-|24.4|1.6|-|-|
> |FP|7.5|-|-|-|**29**|-|-|-|
> |TP|-|-|0|37.6|-|-|0|40.4|

---

> ### Author Response · Authors · 2025-11-26
> **Author Response - Part 3**
>
> > W3: The evaluation of the online supervision setting seems arbitrary in its implementation. For instance, in OSWorld, the verifier is called "every 5 steps".
>
> We implemented verification every 5 steps in OSWorld simply because actions in that benchmark can be more destructive (irreversible). However, as we show below, conclusions about baseline and SGV verifier performance are similar no matter which setting is used.
>
> Specifically, using the verifier for outcome supervision (at the end of an episode, not periodically) as in VisualWebArena has some advantages: it aligns with prior work demonstrating the effectiveness of outcome supervision [1],[2], and it more closely resembles settings such as reinforcement learning. However, in OSWorld UI-Tars can rapidly diverge from the objective and enter states that are difficult to recover from due to the more destructive nature of actions in this benchmark (e.g., deleting a file or closing a window required to define the task). For this reason, we opted for more frequent verifier interventions in OSWorld. The specific choice of interventions every 5 steps was purely based on budget constraints.
>
>
> In the table below, we show results for additional experiments under outcome and periodic interventions across both benchmarks. Consistent with our main findings, SGV provides larger gains than the baseline in all cases, with slightly greater improvements when verification is more frequent. We also note that the issue highlighted in W1--namely, that SGV can overly penalize the agent--is more likely to occur at higher intervention frequencies. Therefore, the improved performance under this setting can be viewed as an additional dimension of robustness in our results.
>
> We thank the reviewer for raising this point and allowing us to clarify our design choices. We have added a discussion of these aspects in our revised manuscript.
>
> **Table R1.3: Performance of Digital Agents Across Verifier Intervention Modes**
> **\*Numbers are averages across 3 runs.**
>
> || Verification Mode | OSW  | VisualWebArena |
> |-|-|-|-|
> | Baseline Agent | -                 | 21.7 | 45.0           |
> | No SGV         | + Outcome-based   | 23.4 | 46.1           |
> |                | + Every 5 Steps   | 24.3 | 46.9           |
> | SGV            | + Outcome-based   | 24.9 | 54.0           |
> |                | + Every 5 Steps   | 26.5 | 54.9           |
>
>
> As a final note, the reviewer implicitly touches on a more fundamental question of *when* to call a verifier. We agree that this is an interesting and largely open research direction that we do not cover, and we will acknowledge it in our limitations. While we leave this exploration for future work, progress on this question benefits from understanding how to design and evaluate verifiers that effectively intervene in the first place. We hope our work provides useful insight into the potential and limitations of these approaches.
>
> [1] Setlur, A., Rajaraman, N., Levine, S., & Kumar, A. (2025). Scaling Test-Time Compute Without Verification or RL is Suboptimal. arXiv:2502.12118
>
> [2] Swamy, G., Choudhury, S., Sun, W., Wu, Z. S., & Bagnell, J. A. (2025). All Roads Lead to Likelihood: The Value of Reinforcement Learning in Fine-Tuning. arXiv:2503.01067

---

### Author Response · Authors · 2025-11-27
**General Comment - Strength of MLLM verifier implementations, Metrics Evaluated**

We thank all reviewers for their time and thoughtful evaluation of our work. We are encouraged that reviewers found our identification of agreement bias---the tendency of MLLMs to overly validate agent behavior---to be compelling and an important, original contribution (R1, R2, R3, R4), especially given its implications for downstream applications that rely on MLLM-based verification (R1, R2, R3). We also appreciate the positive feedback on the quality and comprehensiveness of our experimental design (R3, R4), including the breadth of evaluated models (R2, R3), the diversity of environments (R1, R3), and the range of applications considered (R1), which together support the generality of our findings (R1, R4). Finally, we are grateful that reviewers found our results demonstrating the effectiveness of SGV in mitigating agreement bias to be strong (R1, R2, R4) and recognized the significance of these improvements for downstream applications such as self-refinement and online supervision across multiple benchmarks (R1, R2, R4). We also appreciate the reviewers' insightful questions and suggestions, which have motivated valuable improvements to our manuscript.

Besides the specific responses to each of the reviewers' concerns and questions, below we clarify and extend two points we believe are of general interest to all reviewers.

## Strength of our MLLM Verifier implementations
The table below extends Table 8 of our manuscript, where we evaluate our baselines on the human annotations provided by the concurrent work AgentRewardBench [1] for one of the benchmarks we study, VisualWebArena. As shown, **our *baseline* MLLM verifier is already relatively strong, setting a state-of-the-art on AgentRewardBench**, **SGV further improves upon it, surpassing even the oracles included in the original benchmark**. As discussed in section 3.2, this high performance stems from the fact that existing work utilizes (i) trajectories produced by weak agents, (ii) environments with bugs that make verification more trivial, and (iii) imprecise oracle evaluators---all factors we explicitly address for a reliable evaluation of MLLM Verifiers.

We hope this reassures reviewers that our analysis is based on a sound experimental design; that our identification of agreement bias is not an artifact of weak implementations; and that gains from SGV are measured against relatively strong baselines.

|Category|Method|Precision|
|-|-|-|
|**Rule-based verification**|VWA Oracle (original)|85|
||VWA Oracle (ours)|94|
|**Model-based verification**|No-SGV Baseline (Gemini 2.5, no thinking)|73|
||WebJudge (GPT-o4, current leaderboard SOTA)|75|
||No-SGV Baseline (GPT-o4)|80|
||SGV (GPT-o4)|86|

*Obs.: The primary performance metric in AGRB is precision, which is directly proportional to metrics we consider: $P = \frac{s\cdot TPR}{s\cdot TPR + (1-s)\cdot (1-TNR)}, s= SuccessRate$*

## Metrics Evaluated
To facilitate discussions, we restate the metrics utilized to evaluate MLLM Verifiers.

To evaluate performance and alignment in **offline verification of agent trajectories**, we consider:
- **Bias**: $bias = \tfrac{1}{n}\textstyle\sum_{i}\mathbb{E}[\hat{r}_i-r_i^{\ast}]$,
- **Distance Skewnes**: $dSkew = 1-\frac{\sum_{ij}\lVert(\hat r_i - r_i^{\ast})-(\hat r_j - r_j^{\ast})\rVert}{\sum_{ij}\lVert (\hat r_i - r_i^{\ast})+(\hat r_j - r_j^{\ast})\rVert}$,
- **True Positive and True Negative Rates**: $\text{T?R}(c) = \frac{\sum_{i}\mathbf{1}(\hat r_i=c \wedge r_i^{\ast}=c)}{\sum_{i}\mathbf{1}(r_i^{\ast}=c)}\approx \hat P(\hat r_i=c\mid r_i^{\ast}=c), c\in\\{0,1\\}$

Where $\hat{r}_{i}$ is the reward given by the MLLM verifier to an agent trajectory $i$, and $r_i^*$ is the reward given by humans or oracles.

Notes:
- $bias$ and $dSkew$ are summary statistics reflecting the *distribution* of MLLM responses, where positive values indicate over-validation of trajectories relative to humans or oracles, while values near zero reflect closer alignment.
- $TNR$ measures how reliably MLLMs identify trajectories that are true failures, and is an empirical estimate of the probability of identifying failures. It directly relates to false positives by $TNR=1-FalsePositiveRate$. (Analogous interpretation for TPR).
- **Accuracy** $ACC=(1-SuccessRate)\cdot TNR +SuccessRate\cdot TPR$  is used as an auxiliary metric to support comparisons, providing a summary of the tradeoff between TPR and TNR

A verifier should also be evaluated through its impact on downstream applications: if it is effective, its impact on downstream applications should yield a net benefit. Accordingly, **for downstream applications** (self-refinement via Reflexion and online supervision), we use the **task completion rates (SR)** of agents **with and without MLLM verifier interventions** as the primary evaluation metric.

[1] Koh et al. AgentRewardBench: Benchmarking Large Language Models as Reward Models for Autonomous Agents, 2025.

---

### Author Response · Authors · 2025-11-29
**Summary of Manuscript Revisions**

We thank all reviewers for their constructive comments and suggestions for new experiments, which we believe have substantially improved our manuscript.

We have color-coded all revisions in the updated manuscript, annotating them with searchable "Rₓ" tags corresponding to each reviewer and their questions, as follows:
- Blue, R1: reviewer 187k
- Green, R2: reviewer dsJd
- Red, R3: reviewer twch
- Purple, R4: reviewer Uw7g
- Gold: relevant to multiple reviewers

Below, we summarize relevant modifications.

## Relevant to All Reviewers
**MLLM Verifier distributions:** Following R2 and R4 suggestions, we leveraged and expanded results regarding MLLM output distributions.

- We show that agreement bias manifests as output distributions skewed toward positive labels across 28 evaluation/prompt templates, including common interventions to mitigate bias in LLM evaluators.
- Which in turn:
  - Explains the ineffectiveness of methods like majority voting
  - Provides guidance for future training/test-time methods that may exploit the skewed MLLM distributions induced by agreement bias, as demonstrated by our proof-of-concept Platt-scaling experiment using sampling-based probabilities (R4-W1)
  - Reinforce the effectiveness of SGV, which organically produces more human-aligned output distributions across all templates, while requiring lower compute cost, no ground-truth labels, and extending seamlessly across domains
- Previously, this analysis appeared mostly in the appendix. Part of the discussion was moved to the main text (Tab 3; lines 366–377), and Sec. E.3 was included to discuss the Platt scaling experiments.

**AgentRewardBench:** Updated results demonstrating the strength of our baseline MLLM verifier, which sets a SOTA on AgentRewardBench (Tab. 10).

**Verifier Applications**: Added more context on how existing methods rely directly or indirectly on MLLM Verifiers, and can thus be affected by agreement bias.

## Reviewer-specific
**187k (R1)**
- Q1 & W2: Replaced Tab 1 with Tab. R1.1 containing MLLM Verification results for the expanded set of models. Conclusions were not changed.
- W1: Added ([Fig. 3](https://anonymous.4open.science/r/abias-sgv-iclr-26-3DD2/SankeyDiagram.png)) quantitatively showing that SGV interventions are mostly non-disruptive and lead to higher task completion, in contrast to the baseline verifier.
- W3: Added dedicated Sec. E.4 on outcome vs periodic verification, and discussion about future work in our Conclusion.

**dsJd (R2)**
- W1:
  - Added Sec. E.3 including results for Platt calibration based on MLLM confidence scores, and sampling-derived probabilities.
  - Included results and discussion about MLLM output distributions to the main text (Tab. 3; and lines 366-377), showing that: (i) agreement bias manifests as distributions highly skewed towards positive labels across 28 prompt templates including interventions aimed at mitigating bias in LLM evaluators, and (ii) SGV leads to distributions more aligned to human and oracle judgments.
  - Replaced Fig. 3 by [Fig. 4](https://anonymous.4open.science/r/abias-sgv-iclr-26-3DD2/Response-Distribution.png) containing distributions across all 28 templates.
- W2: Fixed all presentation issues.

**twch (R3)**
- W1: Added discussion of possible orthogonal gains to SGV via integration with visual experts
- W2, effects of trajectory length and verification difficulty: added discussion to our main text (L394-402) and dedicated Sec.s E.5 and E.6
- W3: Added Sec. E.8 with all the additional ablations to SGV mechanism, including prior diversity, grounding tools, and noise in the first-step generation

**Uw7g (R4)**:
- W1: Added discussion in the main text (lines 366–377) and a dedicated Sec. E.3, including Platt calibration experiments using both MLLM confidence scores and model-intrinsic probabilities. Results illustrate the limitations and the potential for mitigating agreement bias through sampling/training techniques that account for skewness in MLLM distributions
- Added Tab. R2.3 to main text showing the effects of chain-of-thought (CoT) on MLLM output distributions and [additional examples](https://anonymous.4open.science/r/abias-sgv-iclr-26-3DD2/abias-extreme-example.png) of CoT rationalization of incorrect behavior
- Regarding the comment that "agreement bias might appear straightforward", the "Verifier Applications" discussion clarifies that several *existing* methods rely on MLLM verifiers whose implementations are similar or often weaker than our baselines, and are therefore vulnerable to agreement bias

## Other changes:
- To make room for additional discussions and results, we made a few presentational improvements in our introduction, Fig. 2 (Reflexion), and Tab. 4 (Online Supervision), and moved the method description to Sec. 3, leaving Secs. 4 and 5 entirely for the discussion of experiments.
- We made minor numerical updates reflecting the release of the stable version of Gemini 2.5, with no impact on our conclusions.

---

### Author Response · Authors · 2025-12-04
**General Comment to the AC - Rebuttal Summary (2/2)**

# Reviewer dsJd (R2)
> W1: Compare the proposed method with (a) different prompts to the verifier model, or (b) prompting models to generate confidences and applying Platt scaling

W1 is addressed by expanded discussions and additional results related to section F of our original manuscript, where we evaluate MLLM verifiers across 28 evaluation/prompt templates with existing methods aimed at mitigating biases in LLM evaluators. Results show that:
- (a) Across all designs, (i) MLLMs tend to concentrate evaluations at the high end of scales (e.g., high scores, or the most favorable Likert labels), and (ii) SGV leads to more balanced and human-aligned distributions.
- (b) Platt scaling of model-generated confidences provides no gains, as models tend to concentrate generations at the high end of the [0,1], rendering calibration corrections ineffective. In contrast, SGV promotes more balanced and human-aligned model-generated confidences, yielding significant gains in verification performance.

Furthermore, R4-W1 experiments demonstrate that SGV surpasses a highly optimistic, proof-of-concept Platt scaling of sampling-derived model probabilities.


# Reviewer twch (R3)
>W1: discussion combining SGV with specialist models for fine-grained perception

We expanded the discussion in our main text on remaining failures due to base models' limitations in vision-language capabilities, discussing how specialist models can complement SGV gains in this dimension.

>W2: effects of longer trajectories on SGV/verifier performance

We provide experiments decomposing performance by trajectory length in OSWorld (50-step) and a new 60-step setup in VisualWebArena, including ~3,744 samples generated across three MLLMs. Results show that:
- SGV gains hold across all trajectory lengths and benchmarks
- Verification performance is **higher** on longer trajectories, as they are dominated by failures and mistakes that are easier to catch--supporting our Sec. 3.2 discussion on the need for reasonably strong trajectories are needed for reliable assessment.
- We explain challenges in evaluating both long trajectories *and* strong agents in the current state of benchmarks/agents, and provide experiments on the latter, showing that agreement bias increases when agents are stronger, and that SGV remains effective in all settings.

>W3: limited ablations of SGV design; more tests would strengthen claims about SGV's mechanism.

Beyond SGV’s gains across 28 prompt/verification ablations (R2W1), we add 10 new ablations showing that:
- SGV can be improved with more diversity in the prior generation, particularly when generated by a model from different families.
- SGV outperforms grounding through web-search tools, and is robust to moderate noise in the first-step generation
- SGV's first step can be implemented with smaller models, offering a lower cost alternative, especially under multiple prior generations.

# Reviewer Uw7g (R4)
>W1: further experiments and analysis on CoT; Is this bias intrinsic to the model itself?; Are there training or sampling techniques that could help mitigate this issue?

- Besides results in our original manuscript showing the ineffectiveness of test-time scaling methods (Table 2), we provide additional experiments over 10k+ samples showing that CoT--whether prompt-based or generated by reasoning models--fails to reduce the model’s tendency to overly validate agent behavior.

- We include new examples illustrating rationalization of flawed trajectories, where an MLLM validates behavior that is, by construction, unsupported by the provided information.

- Altogether, results from the manuscript and this rebuttal (R1W1, R2W1, Table R4.3) consistently indicate that current techniques remain limited in mitigating the model’s inherent tendency to over-validate agent behavior.

- Nevertheless, we offer a new **proof-of-concept** experiment (R4, Response 2) informing on the potentials for test-time/training approaches explicitly aware of the positively skewed confidence distributions induced by agreement bias.

---

### Author Response · Authors · 2025-12-04
**General Comment to the AC - Rebuttal Summary (1/2)**

We thank the AC for handling our submission under the challenging circumstances of this review cycle. To help reduce the workload, below we summarize key points raised by the reviewers and how our rebuttal addresses their concerns. More details are provided in the specific responses to each reviewer.

All feedback was incorporated into our revised manuscript. The changes are described in the general comment 'Summary of Manuscript Revisions', and are annotated in the manuscript with the reviewer ID (Rx) and the concern/question addressed.

---
# Strengths
In terms of **strengths**, reviewers highlighted:
- Our identification of agreement bias--the tendency for MLLMs to overly validate agent behavior--as compelling, and a significant and original contribution (R1, R2, R3, R4), particularly due to its implications for methods relying on MLLM-based verification (R1, R2, R3).
- The comprehensiveness of our experiments, covering several MLLMs (R2, R3), diverse environments and applications (R1, R3), which together support the generality of our findings (R1, R4).
- The strong performance of SGV in improving MLLM verification across challenging benchmarks (R1, R2, R4), and the importance of these gains for downstream applications (R2, R4).

# Response to all reviewers
First, **we highlight additional results provided in the general comment** extending Table 8 of our manuscript, where we evaluate our verifier implementations on the human annotations from the concurrent work AgentRewardBench. Results show that our **baseline** MLLM Verifier already sets a **new state of the art on the benchmark**, and that SGV further improves upon this, surpassing the previous best by 11pps. We hope this confirms that both our identification of agreement bias and the gains of SGV rest on strong implementations and serve as support for the claims made throughout the rebuttal.

# Reviewer 187k (R1)
>Q1&W2: Does the relative gain of SGV shrink as models become more capable?

Our new experiments across 14 MLLMs--including the most recently released models such as Qwen3 and GPT5--shows that SGV benefits MLLMs of different sizes and families with **no uniform trend** as a function of capability, in line with Tables 1 and 9 of our original manuscript.

>W1: The paper lacks a sufficient analysis of when SGV broad priors are helpful vs when they are harmful.

W1 is addressed by:

(1) **New quantitative results** decomposing SGV's impact on downstream applications, showing that its interventions are mostly non-harmful

(2) Parts included the original manuscript:
- Verification involves an intrinsic leniency-strictness tradeoff whose balance is better measured through the impact of a verifier on downstream applications.
- As noted by R2 & R4, SGV meaningfully improves downstream performance, while the (strong) baseline verifier does not.
- As quantified in (1) and illustrated with examples (e.g., Fig 8), this occurs because SGV's 'strict' interventions are mostly harmless, while the reduced agreement bias enables corrective feedback precisely when the agent is wrong--unlike the baseline verifier
- Our qualitative analysis discusses and categorizes cases into strict/lenient and harmless/harmful verification.
- We expanded and clarified these points in the revised manuscript.

> W3: arbitrariness in the online supervision setting (e.g., periodical verification "every 5 steps" in OSWorld).
- We provide **new experiments** covering both periodic and outcome-based verification, showing conclusions remain the same: agreement bias limits MLLM verifier gains, in contrast to SGV.
- We clarify the benefits of periodic verification in settings where irreversible actions (e.g., file deletion in OSWorld) are more common, and add discussion of both designs in the main text and Section E.4.

We highlight **other parts of the rebuttal that address R1's concerns**:
- Expanded discussion and experiments on the positive skew in MLLM evaluations (R2W1, R4W1,) deepening our contributions on agreement bias--a dimension valued by R1--by providing further insights into its root causes and guidance for future work
- SGV promotion of more human-aligned distribution of MLLM evaluations across 28 verification templates (R2W1) and AgentRewardBench results.
- R3W3 ablations demonstrating SGV robustness across several dimensions.

>W2: lack of evidence to suggest whether SGV is a durable solution

While we prefer to remain conservative about long-term claims, we hope our additional clarifications and results can position our work as a relevant contribution, and a step forward in: (i) quantifying a significant problem to the community, and (ii) analysis, metrics, and a method (SGV) that effectively mitigates this problem and can inform future work.

---

### Meta-Review · Area_Chair_LanE · 2026-01-02

**Summary:**

The following reviewer comments informed my decision for this paper:

The reviewers (**187k**, **dsJd**, **twch**, **Uw7g**) comment on the *strength* being the paper’s case that existing MLLMs are biased verifiers towards positive evaluations. Another strength (**187k**, **twch**, **Uw7g**) is the evaluation which is performed in diverse set of multimodal environments (VisualWebArena, OSWorld, robomimic) and applications (offline evaluation, self-refinement, and online supervision)

**187k** states the main *concerns* are (1) the lack of specific context given to SVG can make the method overly strict, (2) the heuristic prompting approach that may not stand the test of time, (3) the online supervision evaluation rigor.

**dsJd** states the main *concerns* are (1) comparison to simpler strategies, (2) further proofreading is needed for the text.

**twch** states the main *concerns* are (1) how the method works with specialist models for fine-grained perception, (2) moderate-length trajectories vs. long trajectories, (3) more experiments on generation and on prompts needed.

**Uw7g** states the main *concerns* are about additional experiments on the reasoning (e.g., to study the “chains of thought to rationalize flawed behavior” and which reasoning steps are flawed).

Another useful reference the authors may find relevant is:
* [1] Mei, Zhiting, et al. "Reasoning about Uncertainty: Do Reasoning Models Know When They Don't Know?." arXiv preprint arXiv:2506.18183 (2025).

**Reviewer Concerns:**

**187k Concerns**
* (1) the lack of specific context given to SVG can make the method overly strict $\rightarrow$ **addressed (discussion and new results)**. Authors clarified that “MLLM _is_ aware of the agent's state” and they point out that “SGV can sometimes produce stricter verification (lower TPR)”.
* (2) the heuristic prompting approach that may not stand the test of time $\rightarrow$ **addressed (empirically added more experiments with more models)**
* (3) the online supervision evaluation rigor $\rightarrow$ **addressed (discussion, few more experiments)**

**dsJd Concerns**
* (1) comparison to simpler strategies $\rightarrow$  **addressed (discussion and more empirical results and a new proof of concept study on calibration)**
* (2) further proofreading $\rightarrow$  **addressed**

**twch Concerns**
* (1) how the method works with specialist models for fine-grained perception $\rightarrow$  **addressed (discussion; agreed with limitations)**
* (2) moderate-length trajectories vs. long trajectories $\rightarrow$  **addressed (empirical, doubled the trajectory horizon)**
* (3) more experiments on generation and on prompts needed $\rightarrow$  **addressed (empirical)**

**Uw7g Concerns**
* (1) additional experiments on the reasoning $\rightarrow$ **addressed**

**Reviewer Scores:**

* **187k** would have *maintained* a score of 4: marginally below the acceptance threshold.
* **dsJd** would have *maintained* a score of 8: accept, good paper (poster)
* **twch** would have *maintained* a score of 4: marginally below the acceptance threshold.
* **Uw7g** would have *maintained* a score of 6: marginally above the acceptance threshold.

---

### Decision · Program_Chairs · 2026-01-26

Accept (Poster)